# Learning Personalized Ad Impact via Contextual Reinforcement Learning under Delayed Rewards[*]

**Yuwei Cheng**
Department of Statistics
University of Chicago
Chicago, IL 60637
`yuweicheng@uchicago.edu`

**Zifeng Zhao**
Mendoza College of Business
University of Notre Dame
Notre Dame, IN 46556
`zzhao2@nd.edu`

**Haifeng Xu**
Department of Computer Science and Data Science
University of Chicago
Chicago, IL 60637
`haifengxu@uchicago.edu`

## Abstract

Online advertising platforms use automated auctions to connect advertisers with potential customers, requiring effective bidding strategies to maximize profits. Accurate ad impact estimation requires considering three key factors: delayed and long-term effects, cumulative ad impacts such as reinforcement or fatigue, and customer heterogeneity. However, these effects are often *not* jointly addressed in previous studies. To capture these factors, we model ad bidding as a Contextual Markov Decision Process (CMDP) with delayed Poisson rewards. For efficient estimation, we propose a two-stage maximum likelihood estimator combined with data-splitting strategies, ensuring controlled estimation error based on the first-stage estimator's (in)accuracy. Building on this, we design a reinforcement learning algorithm to derive efficient personalized bidding strategies. This approach achieves a near-optimal regret bound of $\tilde{\mathcal{O}}(dH^2\sqrt{T})$, where $d$ is the contextual dimension, $H$ is the number of rounds, and $T$ is the number of customers. Our theoretical findings are validated by simulation experiments.

## 1 Introduction

E-commerce is expanding rapidly worldwide, with online sales expected to constitute 23% of total retail by 2027, supported by a 14.4% annual growth [ITA, 2025]. This growth has made digital advertising inevitable, empowering tech giants like Google, Meta, Microsoft, and Amazon, which leverage automated auctions to connect advertisers with customers. Auto-bidding, where platforms handle bid placement on behalf of advertisers, has grown significantly because of its simplified interaction for the advertisers and improved performance thanks to real-time optimization [Aggarwal et al., 2024, Google Ads Support, 2025, Facebook Ads, 2025, Microsoft Ads, 2025, Amazon Ads, 2025]. Given its importance, it is crucial to develop effective ad bidding strategies.

Developing effective bidding strategies requires accurately understanding advertising impacts. Psychological studies have long shown that advertising has a delayed and long-term impact on consumer beliefs and attitudes, ultimately shaping purchasing behavior over time [Vakratsas and Ambler, 1999,

---

[*]This work is supported by the AI2050 program at Schmidt Sciences (Grant G-24-66104), Army Research Office Award W911NF-23-1-0030, and NSF Award CCF-2303372.

39th Conference on Neural Information Processing Systems (NeurIPS 2025).

Lewis and Wong, 2022, Sakalauskas and Kriksciuniene, 2024]. Advertising effectiveness varies significantly across individuals. Observational studies reveal substantial heterogeneity based on demographics, platform usage patterns, and census data [Liu et al., 2019, Gordon et al., 2019]. These insights recommend personalized e-commerce advertising, suggesting platforms should leverage customer data to target high-value users and optimize bidding strategies tailored to diverse behavioral profiles [Sakalauskas and Kriksciuniene, 2024]. Additionally, the impact of ads also depends on their cumulative effect: while repeated exposure can strengthen brand recognition, it may also lead to ad fatigue—highlighting a subtle trade-off that is often overlooked in "learning to bid" literature [Pechmann and Stewart, 1988, Lane, 2000, You et al., 2015, Bell et al., 2022, Guo and Jiang, 2024].

**Limitation in recent work.** However, these insights are not fully reflected in current algorithmic designs. The problem of "learning to bid" has been widely studied, with most prior work modeling it as a (contextual) bandit problem that assumes immediate rewards, such as click-through rates, which prioritizes short-term customer engagement [Weed et al., 2016, De Haan et al., 2016, Ren et al., 2017, Feng et al., 2018, 2023, Han et al., 2024, Zhang and Luo, 2024]. This approach, however, neglects the delayed and cumulative effects of advertising on consumption, potentially leading to incentive misalignment [Deng et al., 2022]. This misalignment is further exacerbated by the rise of auto-bidding systems, where platforms automatically manage bidding decisions on behalf of advertisers. While platforms typically optimize for engagement-based metrics, advertisers ultimately care about long-term revenue growth via production conversion [2]. Recent work has therefore begun to emphasize metrics like target return on ad spend as a practical alternative [Wang et al., 2019, Aggarwal et al., 2024] and design bidding strategies which account for delayed and cumulative effects of ads. Recently, Badanidiyuru et al. [2023] models the long-term causal impact of ad impressions using Markov Decision Process with mixed and delayed Poisson rewards. However, this approach assumes homogeneous treatment effects across users, overlooking the importance of personalization, a crucial factor in advertising effectiveness. To the best of our knowledge, no theoretical work jointly addresses all three aspects—delayed effects, cumulative impacts, and heterogeneity—in modeling ad effectiveness and designing bidding algorithms, potentially due to the modeling complexity and difficulty in estimation.

**Our contribution.** To address this gap, motivated by the initial proposal of Contextual Markov Decision Process (CMDP) [Hallak et al., 2015] to model customer behavior during website interactions, we model auto-bidding as CMDP with delayed Poisson rewards—using context to capture personalized ad impacts and states to capture ads cumulative effects. For effective estimation, rather than fitting all model parameters simultaneously using a single giant likelihood function, we introduce a novel data-splitting strategy and develop a two-stage maximum likelihood estimator that ensures controlled estimation error in the presence of delayed impacts. Based on this efficient online estimation oracle, we design a reinforcement learning algorithm to solve the CMDP with near-optimal regret of $\tilde{\mathcal{O}}(dH^2\sqrt{T})$, where $d$ is the contextual dimension, $H$ is the number of rounds, and $T$ is the number of customers. Finally, we perform simulation studies which validate our theoretical findings.

## 2 Modeling Personalized Ad Impact by CMDP with Delayed Rewards

**Notation.** For a positive integer $T$, we denote $[T] = \{1, 2, \ldots, T\}$. We use standard asymptotic notations, including $\mathcal{O}(\cdot)$, $\Omega(\cdot)$, and $\Theta(\cdot)$, as well as their counterparts $\tilde{\mathcal{O}}(\cdot)$, $\tilde{\Omega}(\cdot)$, and $\tilde{\Theta}(\cdot)$ to suppress logarithmic factors. The symbol $e$ represents the base of the natural logarithm. The Mahalanobis norm is defined as $\|\mathbf{x}\|_\Sigma = \sqrt{\mathbf{x}^\top \Sigma \mathbf{x}}$. $\|\cdot\|_2$ represents $L_2$ norm. For vectors $\mathbf{x}$ and $\mathbf{y}$, we use $\langle \mathbf{x}, \mathbf{y} \rangle$ and $\mathbf{x}^\top \mathbf{y}$ interchangeably to denote their inner product.

To address the complexities of advertisement impacts on product conversions, we model online ad bidding as a Contextual Markov Decision Process (CMDP). This framework captures the three key impacts discussed in Section 1 and allows transitions and rewards to depend on context $\mathbf{x}_t$, personalized information of customer $t$, supporting dynamic and personalized bidding. While incorporating $\mathbf{x}_t$ directly into the state is possible, it greatly enlarges the state space and complicates learning [Levy and Mansour, 2023]. Instead, we adopt a CMDP formulation—common in user-driven applications—that keeps the state compact and treats context as auxiliary information [Hallak et al., 2015], preserving both efficiency and personalization. Our analysis focuses on ad platforms that bid on behalf of advertisers. We use "ad platform" and "learner" interchangeably.

---

[2]We defer a more detailed discussion of the "learning-to-bid" literature to Appendix A.1

Mathematically, a CMDP is defined as the tuple $(\mathcal{X}, \mathcal{S}, \mathcal{A}, \mathcal{M})$, where $\mathcal{X} \subseteq \mathbb{R}^d$ represents the contextual feature space, $\mathcal{S}$ denotes the state space capturing customer ad exposure history, and $\mathcal{A}$ is the action space for ad bidding strategies. The mapping $\mathcal{M}$ assigns each context $\mathbf{x}$ to a Markov Decision Process (MDP), $\mathcal{M}(\mathbf{x}) = (\mathcal{S}, \mathcal{A}, \mathcal{P}^{\mathbf{x}}, \mathcal{R}^{\mathbf{x}}, S_1, H)$, where the state transition probability $\mathcal{P}^{\mathbf{x}}$ and reward function $\mathcal{R}^{\mathbf{x}}$ depend on $\mathbf{x}$. $S_1$ is the initial state. $H$ represents the maximum number of customer-learner interactions with details in the context of online bidding as below.

**Definition 2.1** (State). The state of customer $t$ at round $h$, denoted as $S_h^t = [S_{h,1}^t, S_{h,2}^t]$, encodes information about the two most recent ad exposures. Specifically, let $G_{h,1}^t$ represent the round of the most recent ad impression, we set $S_{h,1}^t = h - G_{h,1}^t$, which captures the time elapsed since that impression. Similarly, let $G_{h,2}^t$ denote the round of the second-to-last ad impression, we set $S_{h,2}^t = G_{h,1}^t - G_{h,2}^t$, representing the time interval between the two most recent impressions. $S_{h,1}^t \in \mathcal{H}_1 = \{\infty, 1, 2, \ldots, H-1\}$ and $S_{h,2}^t \in \mathcal{H}_2 = \{-\infty, 1, 2, \ldots, H-2\}$. The interpretation of $\infty$ and $-\infty$ is deferred to Remark 2.3.

**Assumption 2.2** (Observation). The expected product conversion rate $\mu_h^t$ at round $h$ for customer $t$ with context $\mathbf{x}_t$ follows

$$\mu_h^t = \begin{cases} d_{S_{h,1}^t} \mathbf{x}_t^\top \boldsymbol{\theta}_{S_{h,2}^t} & \text{if } \boldsymbol{o}_h^t = 0 \\ \mathbf{x}_t^\top \boldsymbol{\theta}_{S_{h,1}^t} & \text{otherwise.} \end{cases}$$

The observed product conversion $\mathbf{y}_h^t$ follows a Poisson distribution, i.e. $\mathbf{y}_h^t \sim \text{Poi}(\mu_h^t)$. $d_{S_{h,1}^t}$ represents the delayed advertisements impact, $S_{h,1}^t \in \mathcal{H}_1 = \{\infty, 1, 2, \ldots, H-1\}$. $\boldsymbol{o}_h^t$ indicates the bidding outcome for customer $t$ at round $h$. $\boldsymbol{o}_h^t = 1$ if the bid is won and $\boldsymbol{o}_h^t = 0$ otherwise.

*Remark* 2.3. If customer $t$ has no prior ad exposure, the state is $S_h^t = [\infty, -\infty]$. Specifically, when no advertisement has been successfully displayed, product conversion $\mathbf{y}_h^t \sim \text{Poi}(\mu_h^t)$, where $\mu_h^t = d_\infty \mathbf{x}_t^\top \boldsymbol{\theta}_{-\infty}$, Here, $\mathbf{x}_t^\top \boldsymbol{\theta}_{-\infty}$ represents the natural demand in the absence of ads, which may vary with context $\mathbf{x}_t$ to capture factors such as income, preferences, tastes, and seasonality [Manandhar, 2018]. We set $d_\infty = 1$ to avoid identifiability issue. If the learner wins the bid, $\mathbf{y}_h^t \sim \text{Poi}(\mu_h^t)$ with $\mu_h^t = \mathbf{x}_t^\top \boldsymbol{\theta}_\infty$, where $\boldsymbol{\theta}_\infty$ captures the effect of first-time ad exposure. It is possible to generalize the linear modeling of the expected product conversion rate to a more flexible form, i.e., $\mu_h^t = h_{S_{h,1}^t}(x_t)$. For example, $h_{S_{h,1}^t}$ may be modeled as a state-dependent neural network, though this would come at the cost of sacrificing theoretical guarantees. We refer readers to Appendix A.2 for a detailed discussion.

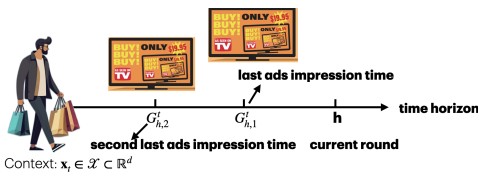

Figure 1: Illustration

As illustrated in Figure 1, if no new advertisement is displayed at round $h$, the impact of the previous ad shown at round $G_{h,1}^t$ carries over, with a delayed and long-term effect governed by $d_{h-G_{h,1}^t}$, depending on the time interval $h - G_{h,1}^t$. This delayed factor allows ad effects to span multiple rounds, enabling delayed conversion peaks, as $d_l$ is not restricted to be less than 1. This results in an expected product conversion of $d_{h-G_{h,1}^t} \mathbf{x}_t^\top \boldsymbol{\theta}_{G_{h,1}^t - G_{h,2}^t}$ (Assumption 2.2), where $\boldsymbol{\theta}_{G_{h,1}^t - G_{h,2}^t}$ captures the impact of the ad shown at $G_{h,1}^t$, which affected by its most recent predecessor at $G_{h,2}^t$. In contrast, if a new ad is successfully displayed at round $h$, its effect overrides the previous one, but itself is influenced by the most recent prior impression at round $G_{h,1}^t$. The resulting impact is modeled by $\boldsymbol{\theta}_{h-G_{h,1}^t}$, leading to an expected conversion of $\mathbf{x}_t^\top \boldsymbol{\theta}_{h-G_{h,1}^t}$.

A natural question arises: why does the cumulative advertising impact depend only on the most recent display, rather than the entire history of ad exposures? This modeling choice is motivated by the recency effect, a well-documented cognitive bias in psychology and marketing [Murphy et al., 2006, Chatfield, 2016, Phillips-Wren et al., 2019]. It suggests that individuals tend to place greater weight on recent experiences when making decisions. This phenomenon is especially relevant in advertising, where exposure timing significantly affects consumer behavior [Chatfield, 2016], and underlies practical approaches like Google's attribution models [Google Ads Attribution, 2025] which uses last-touch heuristics that prioritize recent ad exposures. In fact, our state formulation is very flexible and readily accommodates an enlarged state space. It supports a broad class of CMDP

formulations for $\boldsymbol{\theta}_S$, where $S$ encodes domain knowledge about ad effects, and remains compatible with our proposed learning algorithm. For example, if we believe that all past ad exposures contribute to customer behavior, we can easily extend the state to include $S_h^t = [S_{h,1}^t, S_{h,2}^t, n_h^t]$, where $n_h^t$ is the total number of successful ad displays in the past. We refer readers to Appendix A.3 for a detailed discussion of the flexibility of our state formulation.

The bidding outcome $\boldsymbol{o}_h^t$ depends on both learner's bid $\mathbf{a}_h^t$ and the competitors' bids. The bidding space is given by $\mathbf{a}_h^t \in [0, \boldsymbol{B}_{\mathcal{A}}]$, where $\boldsymbol{B}_{\mathcal{A}}$ is the maximum allowable bid. $\mathbf{a}_h^t = 0$ indicates that the learner opts out of the auction. At each round, the learner submits $\mathbf{a}_h^t$ and wins if its bid exceeds all competitors' bid. The probability of winning by $\mathbf{a}_h^t$ is shown below.

**Definition 2.4** (Transition). For a given bid amount $\mathbf{a}_h^t$, the probability of winning the auction, denoted as $\mathbb{P}(\boldsymbol{o}_h^t = 1)$, is modeled by $\mathcal{F}_h(\mathbf{a}_h^t, \mathbf{x}_t)$, where $\mathcal{F}_h : [0, \boldsymbol{B}_{\mathcal{A}}] \times \mathcal{X} \to [0, 1]$ represents the cumulative distribution function (CDF) of the Highest Other Bids (HOB).

*Remark* 2.5. Given the current state $S_h^t = [S_{h,1}^t, S_{h,2}^t]$ and bid $\mathbf{a}_h^t$, the next state transitions according to $\mathcal{F}_h(\mathbf{a}_h^t, \mathbf{x}_t)$. With probability $\mathcal{F}_h(\mathbf{a}_h^t, \mathbf{x}_t)$, the bid is successful, and the next state is $S_{h+1}^t = [1, S_{h,1}^t]$. Otherwise, the bid is lost, and the state transitions to $S_{h+1}^t = [S_{h,1}^t + 1, S_{h,2}^t]^3$.

This CDF, $\mathcal{F}_h(\mathbf{a}_h^t, \mathbf{x}_t)$, of HOB captures the competitive bidding environment by modeling both context $\mathbf{x}_t$ and time $h$, allowing for round-to-round variation from changing competitors and context-driven bid adjustments. In line with standard practice, we consider a full information feedback setting, where the realized HOB $m_h^t$ is observed by the learner regardless of the bidding outcome [Cesa-Bianchi et al., 2014, Feng et al., 2018, Zhang et al., 2022, Badanidiyuru et al., 2023]. This setting is practical since bidders can always access the "minimum-bid-to-win" feedback [Google Developers, 2024]. Also, ad platforms inherently know realized bids for their auto-bidding algorithms. We assume $m_h^t$ follows a lognormal distribution, a common modeling choice in the literature for advertisement auctions [Laffont et al., 1995, Wilson, 1998, Smith et al., 2003, Skitmore, 2008, Ballesteros-Pérez and Skitmore, 2017] (Assumption 2.6), resulting the probability of winning with a bid $\mathbf{a}_h^t$ as $\mathcal{F}_h(\mathbf{a}_h^t, \mathbf{x}_t) = \Phi((\log(\mathbf{a}_h^t) - \langle \mathbf{x}_t, \boldsymbol{\beta}_h \rangle)/\sigma_h)$, where $\Phi$ denotes the CDF of the standard normal distribution, $\boldsymbol{\beta}_h \in \mathbb{R}^d$ represents the highest willingness to pay for displaying ads from other competitors, capturing the competitiveness of the environment, while $\sigma_h$ reflects the associated variability.

**Assumption 2.6** (Lognormal Distribution of HOB). $\log(m_h^t) \sim \mathcal{N}(\langle \mathbf{x}_t, \boldsymbol{\beta}_h \rangle, \sigma_h^2)$.

In a second-price auction, the format we study, the winning bidder pays the second-highest bid, with payment given by $p_h(\mathbf{a}_h^t, \mathbf{x}_t) = \mathbf{a}_h^t - \frac{1}{\mathcal{F}_h(\mathbf{a}_h^t, \mathbf{x}_t)} \int_0^{\mathbf{a}_h^t} \mathcal{F}_h(v, \mathbf{x}_t) dv$. This format is standard in both industry and research [Cooper and Fang, 2008, Weed et al., 2016, Zhao and Chen, 2019], and our analysis extends to other single-item auctions without entry fees, such as first-price auctions. To bid effectively, the learner must balance the probability of winning, the incurred payment, and the expected product conversion. Without loss of generality, we normalize the value of each unit of product conversion to $\nu = 1$, leading to the following definition of the expected reward.

$$R_h^t \left( S_h^t, \mathbf{a}_h^t, \mathbf{x}_t \right) = d_{S_{h,1}^t} \langle \boldsymbol{\theta}_{S_{h,2}^t}, \mathbf{x}_t \rangle (1 - \mathcal{F}_h(\mathbf{a}_h^t, \mathbf{x}_t)) + (\langle \boldsymbol{\theta}_{S_{h,1}^t}, \mathbf{x}_t \rangle - p_h(\mathbf{a}_h^t, \mathbf{x}_t)) \mathcal{F}_h(\mathbf{a}_h^t, \mathbf{x}_t).$$

By these formulation, in the context of online bidding, the tuple $(\mathcal{X}, \mathcal{S}, \mathcal{A}, \mathcal{P}^{\mathbf{x}_t}, \{R_h^t(S_h^t, \mathbf{a}_h^t, \mathbf{x}_t)\}_{h=1}^H, S_1^t, H)$ is a CMDP (Fact 2.7, Appendix D.1). $\mathcal{P}^{\mathbf{x}_t}$ is the joint probability distribution of states $S_h^t$, induced by $\{\mathcal{F}_h(\cdot, \mathbf{x}_t)\}_{h \in [H]}$. We direct reader to Appendix A.4 for an illustration example for modeling online bidding as a CMDP.

*Fact* 2.7. $(\mathcal{X}, \mathcal{S}, \mathcal{A}, \mathcal{P}^{\mathbf{x}_t}, \{R_h^t(S_h^t, \mathbf{a}_h^t, \mathbf{x}_t)\}_{h=1}^H, S_1^t, H)$ is a CMDP.

### 2.1 Learning goal: regret minimization

The learner interacts with $T$ customers over $H$ rounds, receiving context information $\mathbf{x}_t$ for customer $t$. The learner begins without prior knowledge of the product conversion parameters $\Theta = \{\boldsymbol{\theta}_l\}_{l \in \mathcal{H}} \cup \{d_l\}_{l \in \mathcal{H}_1}^4$ nor the transition $\mathcal{F}^{\mathbf{x}_t} := \{\mathcal{F}_h(\cdot, \mathbf{x}_t)\}_{h=1}^H$. We assume $\Theta$ and transition parameters $\{\boldsymbol{\beta}_h\}_{h \in [H]} \cup \{\sigma_h\}_{h \in [H]}$ are bounded. The context space $\mathcal{X}$ is also bounded, with no distributional

---

$^3$Due to the special meaning of $\infty$ and $-\infty$, we define $h - \infty = \infty$, and $h - \infty - (-\infty) = \infty$.

$^4\mathcal{H} = \mathcal{H}_1 \cup \mathcal{H}_2$

assumptions on the arrival of $\mathbf{x}_t$. $b$ is a small positive constant, which ensures that displaying an ad always yields a non-zero (expected) purchase quantity (Assumption 2.8).

**Assumption 2.8.** There exists positive constants $b, B_x, B_\theta, B_d, B_\beta, \bar{\sigma}$ such that $\|\mathbf{x}_t\|_2 \leq B_x$, and $\theta_l^\top \mathbf{x}_t \geq b, \forall t \in [T], \forall l \in \mathcal{H}, \|\theta_l\|_2 \leq B_\theta, \forall l \in \mathcal{H}, d_l \in [0, B_d], \forall l \in \mathcal{H}_1$, and $\|\beta_h\|_2 \leq B_\beta, \sigma_h \leq \bar{\sigma}, \forall h \in [H]$.

The learner's objective is to minimize the cumulative regret over $T$ customers with regret defined as:

$$\text{Reg}_T := \sum_{t=1}^{T} \text{OPT}\left(\Theta, \mathbf{x}_t, \mathcal{F}^{\mathbf{x}_t}\right) - \mathbb{E}_{\pi_1,\ldots,\pi_T \sim \mathcal{G}}\left[\sum_{t=1}^{T} R\left(\pi_t; \mathbf{x}_t, \Theta, \mathcal{F}^{\mathbf{x}_t}\right)\right]. \tag{1}$$

The expectation is taken over the policy $\pi_t : \mathcal{S} \times \mathcal{X} \rightarrow [0, B_{\mathcal{A}}]$ generated by the algorithm $\mathcal{G}$ employed by the learner. We define $\text{OPT}(\Theta, \mathbf{x}_t, \mathcal{F}^{\mathbf{x}_t})$ as the optimal expected utility achievable for a customer with contextual features $\mathbf{x}_t$ over $H$ rounds, under the true parameters $\Theta$ and the distribution $\mathcal{F}^{\mathbf{x}_t}$. The reward collected by the policy $\pi_t$ over $H$ rounds for customer $t$, denoted as $R(\pi_t; \mathbf{x}_t, \Theta, \mathcal{F}^{\mathbf{x}_t})$, denoted by $R\left(\pi_t; \mathbf{x}_t, \Theta, \mathcal{F}^{\mathbf{x}_t}\right) = \mathbb{E}_{S_h^t \sim \mathcal{P}^{\mathbf{x}_t}}[\sum_{h=1}^{H} R_h^t\left(S_h^t, \pi_t(S_h^t, \mathbf{x}_t), \mathbf{x}_t\right)]$.

## 3  Algorithm Design

In this section, we introduce design principles for solving CMDPs with delayed Poisson rewards. The proposed algorithm (Algorithm 1) consists of three main stages: exploration, exploitation, and estimation, with the estimation stage playing a central role.

The estimation stage contains three key components. First, ridge regression [Abbasi-Yadkori et al., 2011] combined with two-stage variance estimators is used to estimate the transition dynamics $\mathcal{F}^{\mathbf{x}_t}$, handling potential adversarial arrivals of $\mathbf{x}_t$. Second, a variant of online Newton estimator [Xue et al., 2024] is employed to estimate the instant advertisement effects $\{\theta_l\}_{l\in\mathcal{H}}$. Third, a novel two-stage maximum likelihood estimator (TS-MLE) is developed to estimated the delayed impacts $\{d_l\}_{l\in\mathcal{H}_1}$.

The key challenge in online estimation is that a naive approach—simultaneously estimating and updating all parameters by maximizing a joint log-likelihood $\mathcal{L}(\Theta)$—is infeasible due to two main issues. First, the log-likelihood function $\mathcal{L}(\Theta)$ is non-concave in $\Theta$, making it difficult to ensure a unique solution. Second, the score equations $\nabla_\Theta \mathcal{L}(\Theta) = 0$ lack closed-form solutions, rendering direct analysis intractable. These challenges motivate the development of the estimator, TS-MLE, which efficiently estimates delayed effects while maintaining computational tractability.

The core idea of the TS-MLE is to divide the estimation into manageable steps. Intuitively, if the estimation error, $\|\hat{\theta}_l - \theta_l\|_2$, is small, $\hat{\theta}_l$ can then be treated as a close approximation of the true parameter $\theta_l$. Based on this approximation, a maximum likelihood estimator $\hat{d}_l$ can be constructed, ensuring that the estimation error for $d_l$ remains small. This approach is feasible because the conditional log-likelihood $\mathcal{L}(\{d_l\}_{l\in\mathcal{H}_1}|\{\hat{\theta}_l\}_{l\in\mathcal{H}})$ is concave in $\{d_l\}_{l\in\mathcal{H}_1}$ (Eqn. (2)), and both its gradient $\nabla_{\{d_l\}_{l\in\mathcal{H}_1}}\mathcal{L}$ (Eqn. (12)) and the solution to the corresponding score equation $\nabla_{\{d_l\}_{l\in\mathcal{H}_1}}\mathcal{L} = 0$ (Eqn. (3)) are straightforward to compute.

To control the estimation error of $\hat{d}_l$, it is crucial to ensure that this error depends only on the error in $\hat{\theta}_l$, not vice versa. This prevents feedback loops between the estimation errors of $\hat{d}_l$ and $\hat{\theta}_l$, which would otherwise complicate mathematical analysis and make the estimation process intractable. To achieve this, we employ a carefully designed data-splitting strategy, where separate data subsets are allocated exclusively for estimating $\theta_l$ and $d_l$, ensuring their errors remain independent.

To achieve this separation, we introduce two datasets, $\mathbf{W}_{t,l}$ and $\mathbf{D}_{t,l}$, which share a common structure but serve distinct purposes (Def. 3.1). Both datasets focus on round $h$ for customer $t$ where the most recent bid win occurred $l$ rounds before the current round ($S_{h,1}^t = l$). The key difference lies in the parameter being estimated. $\mathbf{W}_{t,l}$ includes rounds that observations $\{\mathbf{y}_h^t\}_{h\in\mathbf{W}_{t,l}}$ with mean $\langle\theta_l, \mathbf{x}_t\rangle$. Thus this datasets is used to estimate $\theta_l$, as they depend solely on $\theta_l$. $\mathbf{D}_{t,l}$ contains rounds that observations $\{\mathbf{y}_h^t\}_{h\in\mathbf{D}_{t,l}}$ with mean $d_l\langle\theta_{S_{h,2}^t}, \mathbf{x}_t\rangle$. This datasets is used to estimate $d_l$ because they depend solely on the unknown parameter $d_l$ when $\{\hat{\theta}_l\}_{l\in\mathcal{H}}$ are given. This ensures that the estimation error of $d_l$ depends on $\theta_l$, while the estimation error of $\theta_l$ remains independent of $d_l$.

**Definition 3.1.** For customer $t$, we define two datasets: $\mathbf{W}_{t,l} = \{h|S_{h,1}^t = l, \boldsymbol{o}_h^t = 1\}$ and $\mathbf{D}_{t,l} = \{h|S_{h,1}^t = l, \boldsymbol{o}_h^t = 0\}$, for $l \in \mathcal{H}\backslash[-\infty]$. We define $\mathbf{W}_{t,-\infty} = \{h|S_{t,1}^t = \infty, \boldsymbol{o}_h^t = 0\}$ for the estimation of $\theta_{-\infty}$. The collection of observations across the first $t$ customers are $\mathbf{W}_l^t = \{\mathbf{W}_{s,l}\}_{s=1}^t$ and $\mathbf{D}_l^t = \{\mathbf{D}_{s,l}\}_{s=1}^t$. The size of $\mathbf{D}_l^t$ is given by $N_{t,l} = |\mathbf{D}_l^t|$.

Taken this together, the log-likelihood of $d_l$ up to the first $t$ customers as:

$$\mathcal{L}\left(\mathbf{y}, d_l; \hat{\boldsymbol{\theta}}_l^s\right) = \sum_{s=1}^t \sum_{h \in \mathbf{D}_{s,l}} \mathbf{y}_h^s \log\left(d_l \mathbf{x}_s^\top \hat{\boldsymbol{\theta}}_{S_{h,2}^s}^s\right) - d_l \mathbf{x}_s^\top \hat{\boldsymbol{\theta}}_{S_{h,2}^s}^s. \tag{2}$$

Differentiating the log-likelihood with respect to $d_l$, setting it to zero, and solving for $d_l$, results in

$$\hat{d}_l^t = \frac{\sum_{s=1}^t \sum_{h \in \mathbf{D}_{s,l}} \mathbf{y}_h^s}{\sum_{s=1}^t \sum_{h \in \mathbf{D}_{s,l}} \langle \hat{\boldsymbol{\theta}}_{S_{h,2}^s}^s, \mathbf{x}_s \rangle}. \tag{3}$$

In the online setting, $\hat{d}_l^t$ estimates $d_l$ as of customer $t$, analogous to $\hat{\boldsymbol{\theta}}_l^t$. Instead of using the most recent estimates $\{\hat{\boldsymbol{\theta}}_{S_{h,2}^t}^t\}_{s=1}^t$ as the first-stage input for estimating $\hat{d}_l^t$, we leverage $\{\hat{\boldsymbol{\theta}}_{S_{h,2}^s}^s\}_{s=1}^t$, a progressively updated estimate of the advertisement's impact across customer arrivals. This novel design is motivated by the observation that the estimation error of $\hat{\boldsymbol{\theta}}_{S_{h,2}^s}^s$ instead of $\hat{\boldsymbol{\theta}}_{S_{h,2}^t}^t$ in the direction of $\mathbf{x}_s$ is well-controlled. Further details, including the confidence region $\mathcal{D}_l^t$ (line 2 of Algorithm 2), are provided in Theorem 4.1 and its proof.

---

**Algorithm 1** Online Contextual Reinforcement Learning with Delayed Poisson Reward

---

**input** $d, T, H, \boldsymbol{b}, \boldsymbol{B}_x, \boldsymbol{B}_\theta, \boldsymbol{B}_d, \boldsymbol{B}_\mathcal{A}, \delta, \gamma, \Gamma, \underline{n_l}$

1: **for** $t = 1$ to $T$ **do**
2:     Obtain the context $\mathbf{x}_t$ for the arriving customer $t$
3:     **if** $t \le (H+1)\underline{n_l}$ **then**
4:         /* Exploration */
5:         Compute $l = \lfloor \frac{t}{n_l} \rfloor$.
6:         **if** $l = H$, set $\mathbf{a}_h^t = 0, \forall h \in [H]$; **else** set $\mathbf{a}_1^t = \mathbf{a}_{l+1}^t = \boldsymbol{B}_\mathcal{A}$, and $\mathbf{a}_h^t = 0$ for $\forall h \ne 1, l+1$
7:         **for** $h = 1$ to $H$ **do** observe $\mathbf{y}_h^t, m_h^t$; then update $\hat{\boldsymbol{\beta}}_h^t$ by Eqn. (5) and $\hat{\sigma}_h^t$ by Eqn. (6)
8:     **else**
9:         /* Exploitation */
10:        Update $\pi_t = \arg\max_\pi \max_{\tilde{\theta} \in \mathcal{C}_{t-1}} R(\pi; \tilde{\theta}, \hat{\mathcal{F}}_{t-1}^{\mathbf{x}_t})$
11:        **for** $h = 1$ to $H$ **do**
12:           Observe $S_h^t$ and take action $\mathbf{a}_h^t = \pi_t(S_h^t, \mathbf{x}_t)$
13:           Observe $\boldsymbol{o}_h^t, m_h^t$, and $\mathbf{y}_h^t$ and update $\hat{\boldsymbol{\beta}}_h^t$ by Eqn. (5) and $\hat{\sigma}_h^t$ by Eqn. (6)
14:        **end for**
15:     **end if**
16:     /* Estimation */
17:     **for** $l \in \mathcal{H}$ **do**
18:        Update dataset $\mathbf{W}_l^t, \mathbf{D}_l^t$ by Def. 3.1
19:        **if** $\mathbf{W}_{t,l}$ is nonempty, compute $\hat{\boldsymbol{\theta}}_l^t$ and $\mathcal{C}_l^t$ by Algo. 3; **else** set $\hat{\boldsymbol{\theta}}_l^t = \hat{\boldsymbol{\theta}}_l^{t-1}$ and $\mathcal{C}_l^t = \mathcal{C}_l^{t-1}$
20:        **if** $\mathbf{D}_{t,l}$ is nonempty, compute $\hat{d}_l^t$ and $\mathcal{D}_l^t$ by Algo. 2; **else** set $\hat{d}_l^t = \hat{d}_l^{t-1}$ and $\mathcal{D}_l^t = \mathcal{D}_l^{t-1}$
21:     **end for**
22:     Set $\mathcal{C}_t = \{\Theta \mid \{\boldsymbol{\theta}_l \in \mathcal{C}_l^t\} \cap \{d_l \in \mathcal{D}_l^t\}, \forall l \in \mathcal{H}\}$
23: **end for**

---

*Remark* 3.2 (Key Tuning Parameters for Algorithm 1). The input parameters $d, T$, and $H$ are structural components of the CMDP. The quantities $\boldsymbol{b}, \boldsymbol{B}_x, \boldsymbol{B}_\theta, \boldsymbol{B}_d, \boldsymbol{B}_\mathcal{A}$ are defined in Assumption 2.8. The tail probability $\delta$ determines the confidence region for $\hat{\boldsymbol{\beta}}_l$ and $\hat{d}_l$ and serves as an input for Algorithm 3 and Algorithm 2. The truncation threshold $\Gamma$, used to handle heavy-tailed distributions, is defined in Eq. (10) and appears in Algorithm 3. The parameter $\gamma$, defined in Lemma 3.3, serves as a weighted estimation error bound and is used in Algorithm 2. The exploration stage guarantees a minimum

number of observations to ensure reliable estimation. Specifically, the threshold $\underline{n}_l$ is given by $\underline{n}_l := \lceil \frac{32 \log(HT)}{e B_d B_x B_\theta b^2} \rceil$, which guarantees sufficient observations in $\mathbf{W}_l^t$ and $\mathbf{D}_l^t$, ensuring $|\mathbf{W}_l^t| \geq \underline{n}_l$ and $N_{t,l} = |\mathbf{D}_l^t| \geq \underline{n}_l$ for all $l \in \mathcal{H}$ in all subsequent episodes after the exploration stage.

The estimation of $\hat{d}_l^t$ depends on accurately estimating $\hat{\boldsymbol{\theta}}_l^t$. Using observations $\mathbf{y}_h^t$ from $\mathbf{W}_l^t$, the problem reduces to efficiently estimating $\boldsymbol{\theta}_l$ from Poisson data. To achieve this, we adopt the Confidence Region with Truncated Mean approach (see Algorithm 3 and Appendix B), introduced by Xue et al. [2024]. This method utilizes a variant of the online Newton estimator, given by:

$$\hat{\boldsymbol{\theta}}_l^t = \arg \min_{\|\boldsymbol{\theta}\|_2 \leq \boldsymbol{B}_\theta} \{ \tfrac{1}{2} \|\boldsymbol{\theta} - \hat{\boldsymbol{\theta}}_l^{t-1}\|_{\mathbf{V}_l^t}^2 + (\boldsymbol{\theta} - \hat{\boldsymbol{\theta}}_l^{t-1})^\top \nabla \tilde{l}_t(\hat{\boldsymbol{\theta}}_l^{t-1}) \}, \tag{4}$$

$\mathbf{V}_l^t = \mathbf{V}_l^{t-1} + \tfrac{1}{2} \mathbf{x}_s \mathbf{x}_s^\top$, with $\mathbf{V}_l^0$ initialized as the identity matrix $\mathbf{I}_d$. $\nabla \tilde{l}_t(\hat{\boldsymbol{\theta}}_l^{t-1}) = (-\tilde{\mathbf{y}}_h^t + (\mathbf{x}_t)^\top \hat{\boldsymbol{\theta}}_l^{t-1}) \mathbf{x}_t$. The truncated observation $\tilde{\mathbf{y}}_h^t$ is defined as $\tilde{\mathbf{y}}_h^t = \mathbf{y}_h^t \mathbb{I}_{\|\mathbf{x}_t\|_{(\mathbf{v}_l^t)^{-1}} |\mathbf{y}_l^t| \leq \Gamma}$.

This truncation mitigates the impact of extreme values in $\mathbf{y}_h^t$ by setting outliers to zero, a technique originally designed for generalized linear bandit problems with heavy-tailed data.

By Lemma 3.3, the weighted estimation error $\|\boldsymbol{\theta}_l - \hat{\boldsymbol{\theta}}_l^t\|_{\mathbf{V}_l^t}^2$ remains well-controlled with high probability.

**Lemma 3.3** (Theorem 1 in Xue et al. [2024]). *Given $l \in \mathcal{H}$, with probability at least $1 - \delta$, $\hat{\boldsymbol{\theta}}_l^t$ defined in Eqn. (4) satisfies $\|\boldsymbol{\theta}_l - \hat{\boldsymbol{\theta}}_l^t\|_{\mathbf{V}_l^t}^2 \leq \gamma, \forall t \geq 0$, where $\gamma = 896 d B_x B_\theta (1 + B_x B_\theta) \log \left( \frac{4T}{\delta} \right) \log \left( 1 + \frac{T}{2d} \right) + 2 B_x^2 B_\theta^2 + 48 d B_x B_\theta \log \left( 1 + \frac{T}{2d} \right).$*

To estimate the transition dynamics $\mathcal{F}_h$, we estimate $\boldsymbol{\beta}_h$ by Eqn. (5). Without loss of generality, we set $\lambda = 1$.

$$\hat{\boldsymbol{\beta}}_h^t = \left( \sum_{s=1}^t \mathbf{x}_s \mathbf{x}_s^\top + \lambda \mathbb{I}_d \right)^{-1} \left( \sum_{s=1}^t \mathbf{x}_s \log(m_h^s) \right). \tag{5}$$

The variability $\sigma_h$, similar to $\hat{d}_l^t$, is estimated based on the first-stage estimators $\{\hat{\boldsymbol{\beta}}_h^s\}_{s=1}^t$ (Eqn. (6)). We use $\{\hat{\boldsymbol{\beta}}_h^s\}_{s=1}^t$ instead of $\hat{\boldsymbol{\beta}}_h^t$ to control over estimation error in the direction of $\mathbf{x}_s$ (details in Appendix C.2).

$$\hat{\sigma}_h^t = \sqrt{\frac{1}{t} \sum_{s=1}^t \left( \log(m_h^s) - \mathbf{x}_s^\top \hat{\boldsymbol{\beta}}_h^s \right)^2}. \tag{6}$$

In addition to estimation, the exploration period aims to gather sufficient observations for $d_l$ to ensure quadratic tail decay of Poisson (Remark 3.2).

After exploration, the algorithm enters exploitation, selecting actions via the greedy policy $\pi_t$ to maximize rewards within the confidence region. $\pi_t$ is computed via dynamic programming with time complexity $\text{Poly}(H, |\mathcal{S}|, B_{\mathcal{A}}/\epsilon)$ when discretizing the bidding space for an $\epsilon$-optimal solution (detail in Appendix A.5). By balancing exploration and exploitation, Algorithm 1 achieves near-optimal performance, formally proven in the next section.

---

**Algorithm 2** Two-Stage Maximum Likelihood Estimation

---

**input** datasets $\mathbf{D}_l^t, \{\mathbf{x}_s\}_{s=1}^t, \{\hat{\boldsymbol{\theta}}_{S_{h,2}^s}^s\}_{s=1}^t$ and parameters $\boldsymbol{b}, \boldsymbol{B}_x, \boldsymbol{B}_\theta, \boldsymbol{B}_d, d, T, H, \delta, \gamma$

1: Update the two-stage estimator $\hat{d}_l^t$ by Eqn. (3)

2: Compute $\mathcal{D}_l^t = \left\{ d_l \in [0, \boldsymbol{B}_d] \mid |\hat{d}_l^t - d_l| \leq \frac{4 H \boldsymbol{B}_d \sqrt{d \log \left( 1 + \frac{T}{2d} \right) \gamma} + \sqrt{2e \boldsymbol{B}_d \boldsymbol{B}_x \boldsymbol{B}_\theta \log(2/\delta)}}{\boldsymbol{b} \sqrt{N_{t,l}}} \right\}$

**output** $(\hat{d}_l^t, \mathcal{D}_l^t)$

---

## 4 Analysis of Near-Optimal Regret Bound

This section demonstrates the efficiency of the TS-MLE (Theorem 4.1) and analyzes the near-optimal performance of the proposed Algorithm 1 (Theorem 4.2).

**Theorem 4.1** (Confidence Region for $\hat{d}_l^t$). *Let $\delta \geq \frac{1}{T^4 H}$ and $N_{t,l} \geq \underline{n}_l$. With probability at least $1 - \delta$, the estimation error $|\hat{d}_l^t - d_l|$, with $\hat{d}_l^t$ defined in Eqn. (3), is bounded by:*

$$\left|\hat{d}_l^t - d_l\right| \leq \frac{4H \boldsymbol{B}_d \sqrt{d \log\left(1 + \frac{T}{2d}\right)\gamma} + \sqrt{2e \boldsymbol{B}_d \boldsymbol{B}_x \boldsymbol{B}_\theta \log\left(\frac{2}{\delta}\right)}}{\boldsymbol{b} \sqrt{N_{t,l}}}. \tag{7}$$

*$\gamma$ is as defined in Lemma 3.3 and $\underline{n}_l$ defined in Remark 3.2.*

Theorem 4.1 shows that the estimation error of TS-MLE is bounded by $\tilde{\mathcal{O}}(1/\sqrt{N_{t,l}})$, where $N_{t,l}$ (Def. 3.1) is the total number of observations used to estimate $d_l$. This result demonstrates the efficiency of the estimator, achieving a near-optimal parametric convergence rate [Rao, 1992].

Proving the near-optimal convergence when data are collected from a complex CMDP with delayed observations requires precise understanding of the sources of estimation error and refined analysis to control them. As discussed in Section 3, the core idea for estimating $d_l$ is to use observations that are "purified" with respect to $d_l$. In particular, we construct $\hat{d}_l^t$ using only observations $\{\{\mathbf{y}_h^s\}_{h \in \mathbf{D}_{s,l}}\}_{s=1}^t$, where $\mathbf{y}_h^s \sim \text{Poi}(d_l \langle \boldsymbol{\theta}_{S_{h,2}^s}, \mathbf{x}_s \rangle)$. Then, to analyze how the estimation error of TS-MLE depends on the first-stage estimators $\hat{\boldsymbol{\theta}}_{S_{h,2}^s}^s$, we decompose $|\hat{d}_l^t - d_l|$ into two parts: the error by the randomness in Poisson observations (Term A in Eqn. 8) and the cumulative estimation error from the first-stage estimators (Term B in Eqn. 8). $\eta_h^s$ in Term A has sub-exponential distributions, specifically, $\text{SubE}(ed_l \mathbf{x}_s^\top \boldsymbol{\theta}_{S_{h,2}^s}, 1)$ (Lemma D.1), since $\mathbf{y}_h^s = d_l \mathbf{x}_s^\top \boldsymbol{\theta}_{S_{h,2}^s} + \eta_h^s$. To control the heavy tail of Term A, we show that $\mathbb{P}(\text{Term A} > \epsilon) \leq \delta$, where $\epsilon = \sqrt{\frac{2e \boldsymbol{B}_d \boldsymbol{B}_x \boldsymbol{B}_\theta}{\boldsymbol{b}^2 N_{t,l}} \log(\frac{2}{\delta})}$. The exploration phase of Algorithm 1 ensures $N_{t,l} \geq \underline{n}_l$, providing sufficient observations to make $\epsilon$ small enough to have faster convergence.

$$|\hat{d}_l^t - d_l| \leq \underbrace{\left| \frac{\sum_{s=1}^t \sum_{h \in \mathbf{D}_{s,l}} \eta_h^s}{\sum_{s=1}^t \sum_{h \in \mathbf{D}_{s,l}} \langle \hat{\boldsymbol{\theta}}_{S_{h,2}^s}^s, \mathbf{x}_s \rangle} \right|}_{\text{Term A}} + \underbrace{\frac{\boldsymbol{B}_d}{N_{t,l}\boldsymbol{b}} \left| \sum_{s=1}^t \sum_{h \in \mathbf{D}_{s,l}} \mathbf{x}_s^\top \left( \boldsymbol{\theta}_{S_{h,2}^s} - \hat{\boldsymbol{\theta}}_{S_{h,2}^s}^s \right) \right|}_{\text{Term B}}. \tag{8}$$

Controlling Term B is essential for bounding $|\hat{d}_l^t - d_l|$. Unlike traditional second-stage estimators, which typically plug in the most recent estimates $\hat{\boldsymbol{\theta}}_{S_{h,2}}^t$, we instead use $\hat{\boldsymbol{\theta}}_{S_{h,2}}^s$, as its estimation error is better controlled in the direction of $\mathbf{x}_s$. Specifically, let $r_h^s = |(\hat{\boldsymbol{\theta}}_{S_{h,2}^s}^s - \boldsymbol{\theta}_{S_{h,2}^s})^\top \mathbf{x}_s|$ denote this directional estimation error. By the Cauchy–Schwarz inequality, we have $r_h^s \leq \|\hat{\boldsymbol{\theta}}_{l'}^s - \boldsymbol{\theta}_{l'}\|_{\mathbf{V}_{l'}^s} \cdot \|\mathbf{x}_s\|_{(\mathbf{V}_{l'}^s)^{-1}}$, where $l' = S_{h,2}^s$. Letting $\gamma_{l'}^s = \|\hat{\boldsymbol{\theta}}_{l'}^s - \boldsymbol{\theta}_{l'}\|_{\mathbf{V}_{l'}^s}^2$, Lemma 3.3 guarantees that $\gamma_{l'}^s \leq \gamma$ holds with high probability for all $s$ and $l'$.

To analyze the growth the directional estimation error $r_h^s$ for each $l \in \mathcal{H}$ over time, we apply recounting techniques. Define the dataset $\boldsymbol{F}_{s,l'}^l := \{h | S_{h,1}^s = l, S_{h,2}^s = l', \boldsymbol{o}_h^s = 0\}$, which partitions $\mathbf{D}_{s,l}$ by $S_{h,2}^s$, the time interval between the two recent ads impression. This partitioning ensures $\sum_{l' \in \mathcal{H}_2} |\boldsymbol{F}_{s,l'}^l| = |\mathbf{D}_{s,l}|$. We then bound the total directional error as: $\sum_{s=1}^t \sum_{h \in \mathbf{D}_{s,l}} |(\hat{\boldsymbol{\theta}}_{S_{h,2}^s}^s - \boldsymbol{\theta}_{S_{h,2}^s})^\top \mathbf{x}_s| \leq \sqrt{N_{t,l}} \sqrt{\sum_{s=1}^t \sum_{l'=1}^H \sum_{h \in \boldsymbol{F}_{s,l'}^l} (r_h^s)^2}$. Let $n_{s,l'} = |\boldsymbol{F}_{s,l'}^l|$, where $n_{s,l'} \leq H$. The total count across all partitions satisfies $\sum_{s=1}^t \sum_{l' \in \mathcal{H}_2} n_{s,l'} = N_{t,l}$. We further bound the error as $\sqrt{N_{t,l}} \sqrt{\sum_{l'=1}^H \gamma \sum_{s=1}^t n_{s,l'} \min(\|\mathbf{x}_s\|_{(\mathbf{V}_{l'}^s)^{-1}}, 1)}$. Applying the Elliptical Potential Lemma [Abbasi-Yadkori et al., 2011], we obtain $2H\sqrt{d}\sqrt{N_{t,l} \log\left(1 + \frac{T}{2d}\right)\gamma}$. Finally, by applying a union bound to account for the simultaneous occurrence of Term A and Term B, we establish Theorem 4.1 (details in Appendix C.1).

Building on the efficiency of TS-MLE, we now show that the regret of Algorithm 1 is nearly optimal, as established in Theorem 4.2.

**Theorem 4.2** (Nearly Optimal Regret). *For any $\delta \geq \frac{6}{T^3}$, with probability at least $1 - \delta$, $\text{Reg}_T$ incurred by Algorithm 1 is $\mathcal{O}(dH^2 \sqrt{T \log(\frac{TH}{\delta})} \log(1 + \frac{T}{2d}))$.*

Theorem 4.2 shows that, with high probability, Algorithm 1 achieves a regret bound of $\tilde{\mathcal{O}}(\sqrt{T})$. Given the $\Omega(\sqrt{T})$ lower bound for CMDPs with even known transitions [Levy and Mansour, 2023], this confirms that Algorithm 1 is nearly optimal in $T$. As discussed earlier, the key challenge in designing an efficient reinforcement learning algorithm for CMDPs is developing an online estimation oracle that handles long-term Poisson rewards under potentially adversarial arrivals of contextual $\mathbf{x}_t$, without relying on distributional assumptions. This improves upon prior work [Modi and Tewari, 2020, Levy and Mansour, 2022, 2023, Levy et al., 2023], which is limited to instantaneous rewards and cannot capture long-term effects in CMDPs, or addresses long-term rewards but fails to account for the heterogeneous effects of $\mathbf{x}_t$ [Badanidiyuru et al., 2023]. Even with TS-MLE, establishing a high-probability regret upper bound for Algorithm 1 is nontrivial and requires careful analysis. We provide a proof sketch below, with detailed arguments in Appendix C.

The proof begins by decomposing $\text{Reg}_T$ into four terms. In Eqn. (9), Term (i) equals $\sum_{t=\tau}^{T} \text{OPT}(\Theta, \mathbf{x}_t, \mathcal{F}^{\mathbf{x}_t}) - \sum_{t=\tau}^{T} \text{OPT}(\Theta, \mathbf{x}_t, \hat{\mathcal{F}}_{t-1}^{\mathbf{x}_t})$, and Term (iv) $= \mathbb{E}(\sum_{t=\tau}^{T} R(\pi_t; \mathbf{x}_t, \Theta, \hat{\mathcal{F}}_{t-1}^{\mathbf{x}_t}) - R(\pi_t; \mathbf{x}_t, \Theta, \mathcal{F}^{\mathbf{x}_t}))$, capturing the cumulative regret due to transition error, arising from the discrepancy between $\mathcal{F}^{\mathbf{x}_t}$ and $\hat{\mathcal{F}}_{t-1}^{\mathbf{x}_t}$[5]. Term (iii) is $\sum_{t=\tau}^{T} \text{OPT}(\mathcal{C}_{t-1}, \mathbf{x}_t, \hat{\mathcal{F}}_{t-1}^{\mathbf{x}_t}) - \mathbb{E}(\sum_{t=\tau}^{T} R(\pi_t; \mathbf{x}_t, \Theta, \hat{\mathcal{F}}_{t-1}^{\mathbf{x}_t}))$, which accounts for decision error resulting from imprecise estimates of the model parameters $\Theta = \{\boldsymbol{\theta}_l\}_{l \in \mathcal{H}} \cup \{d_l\}_{l \in \mathcal{H}_1}$. Thus, accurate estimators of $\mathcal{F}^{\mathbf{x}_t}$ and $\Theta$ lead to low cumulative regret, as shown jointly by Lemma 4.3 and 4.4.

$$\text{Reg}_T \leq \Theta(\log T) + \text{Term (i)} + \text{Term (ii)} + \text{Term (iii)} + \text{Term (iv)} \tag{9}$$

**Lemma 4.3** (Term (i) Upper Bound). *For $\delta \geq 1/T^4$, with probability at least $1 - \delta$, Term (i) in Eqn. (9) is bounded above by $\mathcal{O}(dH^2\sqrt{T}\sqrt{\log((1+T)/\delta)\log(1 + T/d)})$.*

Lemma 4.3 provides a high-probability upper bound for Term (i). The proof firstly uses the simulation lemma [Kearns and Singh, 2002] to bound Term (i) by the cumulative estimation error in the transition dynamics, $\sum_{t=1}^{T} \sup_b \sup_h |\hat{\mathcal{F}}_h^{t-1}(b, \mathbf{x}_t) - \mathcal{F}_h(b, \mathbf{x}_t)|$, which depends on $|\hat{\boldsymbol{\beta}}_h^t - \boldsymbol{\beta}_h|$ and $|(\hat{\sigma}_h^t)^2 - \sigma_h^2|$. A high-probability bound is then derived for these estimations errors, with $|(\hat{\sigma}_h^t)^2 - \sigma_h^2|$ shown to follow a sub-exponential distribution. To ensure quadratic tail decay, we set $\delta \geq 1/T^4$. A similar analysis applies to Term (iv), which is bounded by $\mathcal{O}(dH^2\sqrt{T}\sqrt{\log((1+T)/\delta)\log(1 + T/d)})$ with probability at least $1 - 1/T^4$. Details are in Appendix C.2.

**Lemma 4.4** (Term (iii) Upper Bound). *With probability at least $1 - 2\delta$, where $\delta \geq \frac{1}{T^3}$, Term (iii) in Eqn. (9) is bounded by $\mathcal{O}(H^2\sqrt{dT\log(4TH/\delta)}\log(1 + T/2d))$.*

Lemma 4.4 provides a high-probability bound for Term (iii), which can be decomposed into two parts: the cumulative estimation error of $\hat{\boldsymbol{\theta}}_l^t$, which is $\sum_{t=\tau}^{T} \mathbb{E}(\sum_{h=1}^{H} |\langle \tilde{\boldsymbol{\theta}}_{S_{h,1}^t}^{t-1} - \hat{\boldsymbol{\theta}}_{S_{h,1}^t}^{t-1} + \hat{\boldsymbol{\theta}}_{S_{h,1}^t}^{t-1} - \boldsymbol{\theta}_{S_{h,1}^t}, \mathbf{x}_t \rangle|) + \sum_{t=\tau}^{T} \mathbb{E}(\sum_{h=1}^{H} |\langle \tilde{\boldsymbol{\theta}}_{S_{h,2}^t}^{t-1} - \hat{\boldsymbol{\theta}}_{S_{h,2}^t}^{t-1} + \hat{\boldsymbol{\theta}}_{S_{h,2}^t}^{t-1} - \boldsymbol{\theta}_{S_{h,2}^t}, \mathbf{x}_t \rangle|)$ and the cumulative estimation error of $\hat{d}_l$, specifically, $\sum_{t=\tau}^{T} \mathbb{E}(\sum_{h=1}^{H} |\tilde{d}_{S_{h,1}^t}^{t-1} - \hat{d}_{S_{h,1}^t}^{t-1} + \hat{d}_{S_{h,1}^t}^{t-1} - d_{S_{h,1}^t}|)$. Here, $\tilde{\Theta}_{t-1} := \arg\max_{\Theta \in \mathcal{C}_{t-1}} R(\pi_t; \mathbf{x}_t, \Theta, \hat{\mathcal{F}}_{t-1}^{\mathbf{x}_t})$. Applying Lemma 3.3 and Theorem 4.1, together with a union bound, yields the high-probability bound for Term (iii) (Appendix C.3).

Meanwhile, in Eqn. (9), Term (ii) $= \sum_{t=\tau}^{T} \text{OPT}(\Theta, \mathbf{x}_t, \hat{\mathcal{F}}_{t-1}^{\mathbf{x}_t}) - \sum_{t=\tau}^{T} \text{OPT}(\mathcal{C}_{t-1}, \mathbf{x}_t, \hat{\mathcal{F}}_{t-1}^{\mathbf{x}_t})$, measures the gap in regret between a chosen point and the optimal point within a feasible set. Lemma 4.5 proves $\Theta \in \mathcal{C}_t$ for all $t \geq \tau$ with high probability, ensuring that Term (ii) can be negative with high probability. Combining these results, we conclude that, with probability at least $1 - \frac{6}{T^3}$, the overall regret $\text{Reg}_T$ is $\mathcal{O}(dH^2\sqrt{T\log(\frac{TH}{\delta})}\log(1 + \frac{T}{2d}))$.

**Lemma 4.5** (Confidence Region of $\Theta$). *With probability at least $1 - \delta$ with $\delta \geq \frac{2}{T^3}$, $\Theta \in \mathcal{C}_t, \forall t \geq \tau$.*

# 5 Experiments

We conducted experiments, which leads to two key findings: first, our algorithm achieves near-optimal regret scaling of $\sqrt{T}$, and second it consistently outperforms several strong baselines, including *aggressive bidding*, *passive bidding*, and *random bidding* strategies, with definition shown below.

---

[5]We have $\tau = Hn_l + 1$ and regret incurred in the exploration phase is $\Theta(\log T)$.

**Environment Setup.** We simulate a second-price auction with horizon $H = 3$ (a realistic setting since advertisers typically show ads only a few times per user), context dimension $d = 2$, and $T = 20,000$ sequential customers. The first 2400 rounds are used for exploration, and the remaining rounds for exploitation. Context vectors $\mathbf{x}_t \in \mathbb{R}^2$, as well as parameters $d_l, \boldsymbol{\beta}_h, \sigma_h$, are sampled elementwise from $|\mathcal{N}(0, 1)| + 0.1$ to ensure positivity, while $\boldsymbol{\theta}_l \sim 5|\mathcal{N}(0, 1)| + 0.1$. Under this configuration, each ad impression yields an average instantaneous reward roughly five times its cost (i.e., the highest other bid). The reward also decays rapidly over time, making *aggressive bidding*—always winning the impression—close to optimal and therefore a competitive baseline.

**Estimators.** We estimate four sets of parameters: $\boldsymbol{\theta}_l$ via the online Newton method (Algorithm 3) with truncation threshold $\Gamma = 100{,}000$, bound $\boldsymbol{B}_\theta = 10$, zero initialization, and $V_0 = I$. We estimate delay impact $d_l$ via the two-stage MLE (Eq. (3)) using $\mathbf{D}_{t,l}, \hat{\boldsymbol{\theta}}$, and $\mathbf{x}_t$; $\boldsymbol{\beta}$ by ridge regression (Eq. (5), $\lambda = 1.0$); and $\sigma$ via empirical variance (Eq. (6)).

**Connection Between Bid, Outcome, and Reward.** In our setting, the bidder's action is the bid amount, but due to the second-price auction format, the reward depends only on the outcome–whether the bid exceeds the HOB-not the bid itself. Winning results in paying the HOB; losing incurs no cost. This decoupling allows optimal rewards to be computed from outcomes rather than bids. We leverage this to evaluate both oracle and policy-based rewards. With full knowledge of true parameters, we enumerate all $2^H = 8$ possible win/loss outcomes and define the oracle reward as the maximum achievable value. With estimated parameters, we compute expected rewards for all outcome sequences, select the best, and evaluate it in simulation. Regret is then the cumulative difference between oracle and realized rewards, reflecting both estimation and decision errors.

**Baseline Policies and Regret Comparison.** We compare our algorithm against three fixed policies: *Aggressive bidding*: Always bids above HOB ($\boldsymbol{o}_t = [1, 1, 1]$). *Random bidding*: Samples $\boldsymbol{o}_t$ uniformly from all 8 outcomes. *Passive bidding*: Rarely bids or wins ($\boldsymbol{o}_t = [0, 0, 0]$). All baselines are evaluated using the same oracle-based regret metric. Note that *aggressive bidding*, while competitive, still incurs $O(T)$ regret since it is non-adaptive.

We ran all algorithms over 20 independent trials. Table 1 reports the mean cumulative regret ($\pm$ 0.5 standard deviation) and the fitted regret order (via log–log regression). Our algorithm achieves an estimated regret order of 0.37, while all baselines exhibit linear regret, validating our theoretical results and demonstrating the substantial advantage of adaptive learning in ad bidding.

Table 1: Average cumulative regret ($\pm$ 0.5 standard deviation) and associated fitted regret order ($T^\alpha$).

| $t$ | **Algorithm 1** | **Aggressive Bid** | **Random Bid** | **Passive Bid** |
|---|---|---|---|---|
| 500 | 1770 [1174, 2379] | 228 [121, 336] | 1920 [1791, 2049] | 4630 [3391, 5869] |
| 5000 | 8465 [5585, 11345] | 2373 [1243, 3502] | 19285 [17873, 20697] | 46179 [33850, 58508] |
| 10000 | 8484 [5606, 11363] | 4741 [2477, 7005] | 38399 [35573, 41226] | 92468 [67735, 117201] |
| 15000 | 8505 [5628, 11382] | 7142 [3724, 10561] | 57651 [53452, 61849] | 138652 [101576, 175729] |
| 20000 | **8525 [5649, 11401]** | 9568 [4991, 14146] | 76945 [71390, 82500] | 184873 [135414, 234332] |
| $T^\alpha$ | **0.37** | **1.0** | **1.0** | **1.0** |

## 6 Discussion and Limitation

In this paper, we model ad bidding as a CMDP, presenting a unified framework to address delayed impacts, cumulative effects, and individual personalization. We designed a near-optimal algorithm using a novel two-stage maximum likelihood estimator to effectively handle delayed rewards. However, several important directions remain open for future research. One promising extension involves incorporating budget constraints, a critical practical consideration in real-world advertising. Developing constrained optimization algorithms for CMDP with delayed rewards is highly non-trivial and will be explored in future work.Although our work validates the theoretical insights through experiments, implementing and evaluating the proposed algorithms in real-world settings would provide valuable evidence of its practical applicability. In particular, such experiments could help assess two modeling choices: the adequacy of modeling the expected product conversion rate using linear functions, and the robustness of focusing only on very recent ad exposures due to recency bias. However, given the lack of publicly available online advertising and bidding datasets, we leave this empirical validation to future work as an important step toward bridging the gap between theory and practice.

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

# Appendix to *"Learning Personalized Ad Impact via Contextual Reinforcement Learning under Delayed Rewards"*

## A  Additional Discussion

### A.1  Related Work

Motivated by online advertising auctions, Weed et al. [2016] study repeated Vickrey auctions where goods with unknown values ($v_t$) are sold sequentially. Bidders receive (potentially noisy) feedback about a good's value only after purchasing it—analogous to observing product conversions after displaying an ad. They formulate this problem as online learning with bandit feedback and propose a minimax-optimal bidding strategy for bidding ($b_t$). However, a key limitation of their model is the neglect of long-term advertising effects: winning a bid at time $t$ may influence future observations and outcomes, such as conversions at time $t + 1$. Building on this line of work, Feng et al. [2018] examines learning to bid without knowing one's value in a similar bandit setting and develops the WIN-EXP algorithm, which achieves sublinear regret ($O(\sqrt{T|O|\log(B)})$). Yet, like the earlier study, it assumes that the effect of winning a bid is instantaneous, ignoring possible intertemporal dependencies.

In a related direction, Han et al. [2024] investigate online learning in repeated first-price auctions, where a bidder observes only the winning bid after each auction and must adaptively learn to maximize cumulative payoff. Facing censored feedback—since winning bidders cannot observe their competitors' bids—they design the first algorithm achieving near-optimal ($O(\sqrt{T})$) regret by exploiting structural properties of first-price auctions, including their feedback and payoff functions. They formulate the problem as partially ordered contextual bandits, incorporating graph feedback across actions, cross-learning across contexts, and a partial order over private values. Nonetheless, their framework similarly assumes independence between the value process ($v_t$) and past bidding outcomes, thereby excluding long-term effects of winning bids.

Extending to contextual settings, Zhang and Luo [2024] study online learning in contextual pay-per-click auctions. At each of $T$ rounds, the learner observes contextual information and a set of candidate ads, estimates their click-through rates (CTRs), and conducts a second-price pay-per-click auction. The objective is to minimize regret relative to an oracle that makes perfect CTR predictions. They show that a $\sqrt{T}$-regret rate is achievable (albeit with computational inefficiency) and provably optimal, as the problem reduces to the classical multi-armed bandit setting.

While these studies advance our understanding of strategic bidding under uncertainty, they share a common limitation: none account for the reinforcement or long-term impact of successful ad displays. This omission contrasts with empirical evidence such as De Haan et al. [2016], who demonstrate that online advertisements exert persistent effects on purchase conversion in a multi-channel attribution framework—though their dataset is not publicly available.

More recently, Badanidiyuru et al. [2021] address this gap by modeling the long-term causal impact of ad impressions using a Markov Decision Process (MDP) with mixed and delayed Poisson rewards. However, their approach assumes homogeneous treatment effects across users, overlooking the crucial role of personalization in determining advertising effectiveness. To address these limitations, our work jointly models the long-term, reinforcing, and personalized effects of advertisements, hoping to bridge the gap between sequential decision-making and individualized ad impact estimation.

### A.2  On the Potential to Extend the Linearity Assumption

Briefly recap of our model (Assumption 2.2), if we won the bid $o_h^t = 1$, the average product conversion rate $\mu_h^t = h_{S_{h,1}^t}(\mathbf{x}_t)$, where $\mathbf{x}_t$ is the received feature of customer, and $S_{h,1}^t$ is the state which captures the time elapsed since most recent ads impression, $h_{S_{h,1}^t}$ is the state dependent neural network which we want to estimate. If we lose the bid $o_h^t = 0$, the average product conversion rate $\mu_h^t = d_{S_{h,1}^t} h_{S_{h,2}^t}(\mathbf{x}_t)$, where $d_{S_{h,1}^t}$ is the delayed impact, and $S_{h,2}^t$ is the time interval between the two most recent impressions (Def 2.1).

To incorporate the estimation of the state-dependent neural network $h_l$ (analogous to $\boldsymbol{\theta}_l$) into our algorithm, we could do follows. Everything in Algorithm 1 remains unchanged, i.e. the exploration, exploitation, data-splitting strategy, and estimation, exception the following modifications.

- First, we replace the estimation of $\boldsymbol{\theta}_l$ (Eqn. (4)) by estimating the neural network $h_l$ by the following procedure (similar to Algorithm 1 in Jia et al. [2022]).
  - We approximate $h(x)$ by $f(x, \theta) = \sqrt{m} W_L \phi(W_{L-1}\phi(\dots\phi(W_1 x)))$, where $m$ is the network width, and $\phi(x) = ReLU(x)$, $L$ is the neural network depth.
  - We initialize $\theta_0 = (\text{vec}(W_1)\dots\text{vec}(W_L))$ by random sample entry from $\mathcal{N}(0, 4/m)$ and do online updating by gradient descent with perturbed reward using observed product conversion $\mathbf{y}_h^t$ by follows
    * Generated observation perturbation $\gamma_{s\in[t]}^t \sim \text{Poi}(\lambda)$, then generate binomial random variable $\mathbb{I}_{s\in[t]}$ to add or subtract the noise from the observation

    $$\min_\theta L(\theta) = \sum_{s=1}^{t}(f(\mathbf{x}_t, \theta) - (\mathbf{y}_h^s + \mathbb{I}_s\gamma_s^t))^2/2 + m\beta\|\theta - \theta_0\|^2/2$$

  - Then based on this first stage estimates of $h$, plugging in to estimate delayed impact factor $d$ by $\dfrac{\sum_{s=1}^{t}\sum_{h\in\mathbf{D}_{s,l}}\mathbf{y}_h^s}{\sum_{s=1}^{t}\sum_{h\in\mathbf{D}_{s,l}}f_{S_{h,2}^s}(\mathbf{x}_s, \hat{\boldsymbol{\theta}})}$ (analogous to Eqn. (3))

- Since we do not have theoretical guarantees for $h$ estimation (no confidence interval), we cannot do upper confidence bound (Line 10 of our Algorithm 1). Instead, we might act greedily based on our point estimate. Even though Algorithm in Jia et al. [2022] has theoretical guarantees, the key difference is that they assume noise of their observation is sub-Gaussian, while in our case, it is not realistic to assume that the observation $\mathbf{y}_h^t$ has sub-Gaussian noise, since product conversion has been known to have heavy tail for a long period of time. It is still a challenging open problem of developing theoretical-guaranteed neural contextual bandits with heavy tail observations.

However, our proposed framework leads to near-optimal results if we can parameterize $h_l(\mathbf{x}_t) = \boldsymbol{\theta}_l^\top \phi(\mathbf{x}_t)$, where $\phi$ can be the neural net based embedding or other known feature mapping, then our algorithm remains near optimal, satisfying $\sqrt{T}$ regret bound in this case. This construction is very common in academia and industry as we mentioned from foundational studies such as Jin et al. [2020] to the recent ones [He et al., 2022]. In addition, in real world, companies might construct these feature mapping using complicated methods but retain the overall linear structure for the ease of scalability and interpretability. For example, the expected conversion rate could be a linear function of the expected webpage stay time and the predicted total spending but the expected webpage stay time and the predicted total spending are deep neural networks of the observed customer feature $\mathbf{x}_t$.

To summarize, our proposed framework leads to near-optimal results if there exists an online estimation oracle with theoretical guarantee for the estimation of $h$ and the estimation oracle employed in our paper based on our knowledge is state-of-the-art. Our algorithm is possible to extend to a neural network if we drop the theoretical concerns. Developing theoretical guarantee for neural bandits under heavy tail observation could be a promising future direction.

### A.3 Flexibility in State Space Modeling

If we believe that all past ad exposures contribute to customer behavior, we can enrich the state variable to $S_h^t = [S_{h,1}^t, S_{h,2}^t, n_h^t]$, where $n_h^t$ denotes the total number of ads the user has seen prior to round $h$. Under this formulation, the expected conversion from showing an ad at round $h$ is given by $\langle \boldsymbol{\theta}_{S_{h,1}^t, n_h^t}, \mathbf{x}_t \rangle$, with other modeling assumptions unchanged. In other words, the effect of the current ad depends not only on the time since the last impression, $h - G_{h,1}^t$ (as shown in Figure 1), but also on the cumulative number of ad exposures the user has experienced so far.

Accordingly, Assumption 2.2 as follows:

1. When $\boldsymbol{o}_h^t = 0$, no ad is shown at round $h$, and the effect of recent ads at $G_{h,1}^t$ carries over, given by $\langle \boldsymbol{\theta}_{S_{h,2}^t, (n_h^t-1)\vee 0}, \mathbf{x}_t \rangle d_{S_{h,1}^t}$, where $a \vee b$ denotes $\max(a, b)$.

2. When $o_h^t = 1$, an ad is shown at round $h$, and the impact of the current ad is modeled as $\langle \boldsymbol{\theta}_{S_{h,1}^t, n_h^t}, \mathbf{x}_t \rangle$, influenced by all the past ads exposure history.

We emphasize that, our core algorithmic design, including the data-splitting strategy, the proposed two-stage estimator, and the exploration-exploitation algorithm, readily extends to this new model with the enriched state representation. In addition, our theoretical results continue to hold under this new model, with a new optimal regret bound of order $\widetilde{O}(dH^3\sqrt{T})$, in comparison to $\widetilde{O}(dH^2\sqrt{T})$ under the previous model. We provide supporting evidence below.

Now the enriched states is $S_h^t = [S_{h,1}^t, S_{h,2}^t, n_h^t]$. In terms of state transition, if $o_h^t = 1$, $S_{h+1}^t = [1, S_{h,1}^t, n_h^t + 1]$; else, $S_{h+1}^t = [S_{h,1}^t + 1, S_{h,2}^t, n_h^t]$. $\Theta$ now becomes $\{\boldsymbol{\theta}_{l,n}\}_{l,n \in \mathcal{H}} \cup \{d_l\}_{l \in \mathcal{H}_1}$ with $\mathcal{H} = \{(l,n) : l \in \mathcal{H}_1 \cup \mathcal{H}_2, n < l\}$, $\mathcal{H}_1 = \{\infty, 1, 2, \dots, H-1\}$ and $\mathcal{H}_2 = \{-\infty, 1, 2, \dots, H-2\}$.

For data spiting strategy in Def 3.1, $\mathbf{D}_{t,l}$ does not change and $\mathbf{W}_{t,l}$ will extend to $\mathbf{W}_{t,l,n} = \{h | S_{h,1}^t = l, n_h^t = n, o_h^t = 1\}, \forall l \in [\infty, 1, 2, \dots, H], \mathbf{W}_{t,-\infty,n} = \{h | S_{h,1}^t = -\infty, n_h^t = n, o_h^t = 0\}$. Also, $\mathbf{W}_l^t$ will extend to $\mathbf{W}_{l,n}^t = \{\mathbf{W}_{s,l,n}\}_{s=1}^t$. The estimation of $\boldsymbol{\theta}_{l,n}$ using observation in $\mathbf{W}_{l,n}^t$ still follows online newton estimator with Eqn. (4). The TS-MLE now becomes

$$\hat{d}_l^t = \frac{\sum_{s=1}^t \sum_{h \in \mathbf{D}_{s,l}} \mathbf{y}_h^s}{\sum_{s=1}^t \sum_{h \in \mathbf{D}_{s,l}} \langle \hat{\boldsymbol{\theta}}_{S_{h,2}^s, n_h^s - 1 \vee 0}^s, \mathbf{x}_s \rangle}.$$

Plugging in these new estimators in to the regret decomposition (Eqn. ((9)), Term (i) and Term (iv) does not change since new states did not affect transition error. For the estimation error, Term (iii), we just need an additional for loop to account the effect of $n_h^t$, which makes the new regret upper bound now $\tilde{\mathcal{O}}(dH^3\sqrt{T})$.

The above example suggests that our state formulation is very flexible and readily accommodates an enlarged state space. It supports a broad class of CMDP formulations for $\boldsymbol{\theta}_S$, where $S$ encodes domain knowledge about ad effects, and remains compatible with our proposed learning algorithm.

### A.4 Illustrative Example

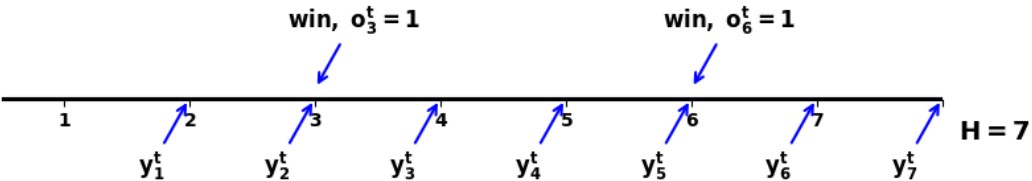

Figure 2: Illustration of Learner-Customer Interaction in CMDP

Figure 2 presents a simple example of a customer, with contextual $\mathbf{x}_t$ and no prior ad exposure, interacting with the learner $\mathcal{L}$ over $H = 7$ rounds. At the start of each round, $\mathcal{L}$ observes the state $S_h^t$, determines the bid amount $\mathbf{a}_h^t = \pi_t(S_h^t, \mathbf{x}_t)$ based on policy $\pi_t$, observes HOB $m_h^t$, and receives the bidding outcome $o_h^t$ along with the reward $R_h^t(S_h^t, \mathbf{a}_h^t, \mathbf{x}_t)$. At the end of each round, $\mathcal{L}$ observes the total product conversion $\mathbf{y}_h^t$ over the current round. As shown in Figure 2, the customer's first ad impression occurs at $h = 3$, resulting in $S_1^t = S_2^t = S_3^t = [-\infty, \infty]$ and $\mu_1^t = \mu_2^t = \langle \boldsymbol{\theta}_{-\infty}, \mathbf{x}_t \rangle$ ($\mu_h^t$ defined in Def.2.2). The first ads exposure updates the expected conversion to $\mu_3^t = \langle \boldsymbol{\theta}_\infty, \mathbf{x}_t \rangle$, capturing the immediate impact of ad exposure. The second ad impression occurs at $h = 6$. Between $h = 4$ and $h = 6$, the state evolves as $S_4^t = [1, \infty], S_5^t = [2, \infty], S_6^t = [3, \infty]$, with expected conversions $\mu_4^t = d_1\langle \boldsymbol{\theta}_\infty, \mathbf{x}_t \rangle$ and $\mu_5^t = d_2\langle \boldsymbol{\theta}_\infty, \mathbf{x}_t \rangle$, reflecting the delayed and long-term impact of the ad displayed at $h = 3$. $\mu_6^t = \langle \boldsymbol{\theta}_3, \mathbf{x}_t \rangle$, capturing the effect of repeated ad exposure. Finally, at $h = 7$, $S_7^t = [1, 3]$ and $\mu_7^t = d_1\langle \boldsymbol{\theta}_3, \mathbf{x}_t \rangle$, demonstrating the long-term effects of the second ad impression.

## A.5 Computational Complexity

For each episode of Algorithm 1, the computational complexities for updating $\hat{\boldsymbol{\theta}}_l^t$, $\hat{\boldsymbol{\beta}}_l^t$, and $\hat{d}_l^t$ are $\mathcal{O}(d^2)$ [Xue et al., 2024], $\mathcal{O}(d^2)$, and $\mathcal{O}(d)$, respectively, for each $l \in \mathcal{H}$. Notably, for a fixed policy $\pi$ and transition $\mathcal{F}^{\mathbf{x}_t}$, $R(\pi, \Theta, \mathcal{F}^{\mathbf{x}_t})$ is nondecreasing in $d_l$ and $\boldsymbol{\theta}_l$, simplifying the computation of $\max_{\tilde{\Theta} \in \mathcal{C}_{t-1}} R(\pi; \tilde{\Theta}, \mathcal{F}_{t-1}^{\mathbf{x}_t})$. The corresponding maximizer are $\tilde{d}_l^t = \hat{d}_l^t + \frac{4HB_d\sqrt{d\log\left(1+\frac{T}{2d}\right)\gamma} + \sqrt{2eB_dB_xB_\theta\log(2/\delta)}}{b\sqrt{N_{t,l}}}$ and $\tilde{\boldsymbol{\theta}}_l^t = \hat{\boldsymbol{\theta}}_l^t + \frac{\sqrt{\gamma}(\mathbf{V}_l^t)^{-1}\mathbf{x}_t}{\|\mathbf{x}_t\|_{(\mathbf{V}_l^t)^{-1}}}$, obtained via constrained linear optimization, where $\hat{\boldsymbol{\theta}}_l^t$ is from Eqn. (4) and $\gamma$ is from Lemma 3.3. The greedy policy $\pi$ is then computed via dynamic programming with time complexity $\text{Poly}(H, |\mathcal{S}|, B_{\mathcal{A}}/\epsilon)$ when discretizing the bidding space for an $\epsilon$-optimal solution [Agarwal et al., 2019].

# B Details of Omitted Algorithm

In this section, we detail the algorithm used to estimate $\boldsymbol{\theta}_l$ for $l \in \mathcal{H}$. Originally introduced by Xue et al. [2024], the Confidence Region with Truncated Mean (CRTM) algorithm (Algorithm 3) serves as an efficient estimator for generalized bandits with heavy-tailed rewards. In their framework, the stochastic observations $\mathbf{y}_t$ follow a generalized linear model:

$$\mathbb{P}(\mathbf{y}_t|\mathbf{x}_t) = \exp\left(\frac{\mathbf{y}_t\mathbf{x}_t^\top\boldsymbol{\theta}_l - m(\mathbf{x}_t^\top\boldsymbol{\theta}_l)}{g(\tau)} + h(\mathbf{y}_t, \tau)\right),$$

where $\boldsymbol{\theta}_l$ represents the inherent parameters, $\tau > 0$ is a known scale parameter, and $g(\cdot)$ and $h(\cdot)$ are normalizers. The conditional expectation of $\mathbf{y}_t$ is given by:

$$\mathbb{E}(\mathbf{y}_t|\mathbf{x}_t) = m'(\mathbf{x}_t^\top\boldsymbol{\theta}_l),$$

where $m'(\cdot)$ is the link function, denoted as $\mu(\cdot) = m'(\cdot)$. Thus, $\mathbf{y}_t$ can be expressed as:

$$\mathbf{y}_t = \mu(\mathbf{x}_t^\top\boldsymbol{\theta}_l) + \eta_t,$$

where $\eta_t$ is a random noise term satisfying $\mathbb{E}(\eta_t|\mathcal{G}_{t-1}) = 0$. Here, $\mathcal{G}_{t-1} = \{\mathbf{x}_1, \mathbf{y}_1, \ldots, \mathbf{x}_{t-1}, \mathbf{y}_{t-1}, \mathbf{x}_t\}$ denotes the $\sigma$-filtration up to time $t-1$. CRTM assumes that the link function $\mu(\cdot)$ is $L$-Lipschitz, continuously differentiable, and has a first derivative $\mu'(\cdot)$ bounded below by a positive constant $\kappa$, i.e., $\mu'(z) \geq \kappa$. Additionally, CRTM assumes that the observations $\mathbf{y}_t$ have bounded moments. Specifically, there exist positive constants $u$ and $0 < \epsilon \leq 1$ such that:

$$\mathbb{E}(|\mathbf{y}_t|^{1+\epsilon} \mid \mathcal{G}_{t-1}) \leq u.$$

In our setting, where $\mathbf{y}_t$ follows a Poisson distribution, we have $\epsilon = 1$, $u = B_xB_\theta(1 + B_xB_\theta)$, $\mu(\cdot)$ as a linear function, and $\kappa = 1$. Based on the specific parameters in our setting, the CRTM algorithm (Algorithm 3) is adapted as follows:

**Algorithm 3** Confidence Region with Truncated Mean [Xue et al., 2024]

---

**Require:** $\mathbf{W}_{t,l}, \mathbf{x}_t, \mathbf{V}_l^{t-1}, \hat{\boldsymbol{\theta}}_l^{t-1}, \Gamma, \gamma$

1: **for** $\mathbf{y}_h^t \in \mathbf{W}_{t,l}$ **do**

2:      Truncate the observed payoff: $\tilde{\mathbf{y}}_h^t = \mathbf{y}_h^t \mathbb{I}_{\|\mathbf{x}_t\|_{(\mathbf{V}_l^t)^{-1}}|\mathbf{y}_h^t| \leq \Gamma}$

3:      Compute the gradient: $\nabla \tilde{l}_t(\hat{\boldsymbol{\theta}}_l^{t-1}) = (-\tilde{\mathbf{y}}_h^t + \mathbf{x}_t^\top \hat{\boldsymbol{\theta}}_l^{t-1})\mathbf{x}_t$

4:      Update $\mathbf{V}_l^t = \mathbf{V}_l^{t-1} + \frac{1}{2}\mathbf{x}_t \mathbf{x}_t^\top$

5:      Update the estimator:

$$\hat{\boldsymbol{\theta}}_l^t = \arg\min_{\boldsymbol{\theta} \in \mathbb{R}^d} \frac{\|\boldsymbol{\theta} - \hat{\boldsymbol{\theta}}_l^{t-1}\|_{\mathbf{V}_l^t}^2}{2} + \langle \boldsymbol{\theta} - \hat{\boldsymbol{\theta}}_l^{t-1}, \nabla \tilde{l}_t(\hat{\boldsymbol{\theta}}_l^{t-1}) \rangle$$

6:      Construct the confidence region:

$$\mathcal{C}_l^t = \left\{ \boldsymbol{\theta}_l \in \mathbb{R}^d \mid \|\boldsymbol{\theta}_l - \hat{\boldsymbol{\theta}}_l^t\|_{\mathbf{V}_l^t}^2 \leq \gamma \right\}.$$

7: **end for**

8: **return** $(\hat{\boldsymbol{\theta}}_l^t, \mathcal{C}_l^t)$

---

In particular, the truncation threshold $\Gamma$ is defined by Eqn. (10).

$$\Gamma = 2\sqrt{\boldsymbol{B}_x \boldsymbol{B}_\theta (1 + \boldsymbol{B}_x \boldsymbol{B}_\theta) \log\left(\frac{4T}{\delta}\right) d \log\left(1 + \frac{T}{2d}\right)} \tag{10}$$

Moreover, the bound for the weighted estimation error, $\|\boldsymbol{\theta}_l - \hat{\boldsymbol{\theta}}_l^t\|_{\mathbf{V}_l^t}^2$, denoted by $\gamma$, is given by:

$$\gamma = 896 d \boldsymbol{B}_x \boldsymbol{B}_\theta (1 + \boldsymbol{B}_x \boldsymbol{B}_\theta) \log\left(\frac{4T}{\delta}\right) \log\left(1 + \frac{T}{2d}\right) + 2\boldsymbol{B}_x^2 \boldsymbol{B}_\theta^2 + 48 d \boldsymbol{B}_x \boldsymbol{B}_\theta \log\left(1 + \frac{T}{2d}\right).$$

Running Algorithm 3 leads to the following lemma.

*Lemma* (Lemma 3.3 restated). Given $l \in \mathcal{H}$, with probability at least $1 - \delta$, $\hat{\boldsymbol{\theta}}_l^t$ defined in Eqn. (4) satisfies $\|\boldsymbol{\theta}_l - \hat{\boldsymbol{\theta}}_l^t\|_{\mathbf{V}_l^t}^2 \leq \gamma, \forall t \geq 0$.

## C    Proof of Theorem 4.2

*Theorem* (Theorem 4.2 restated). For any $\delta \geq \frac{6}{T^3}$, with probability at least $1 - \delta$, $\text{Reg}_T$ incurred by Algorithm 1 is $\mathcal{O}(dH^2 \sqrt{T \log(\frac{TH}{\delta})} \log(1 + \frac{T}{2d}))$.

*Proof.* In this section, we present the detailed proof of Theorem 4.2. We begin by decomposing $\text{Reg}_T$ (defined in Eqn. (1)) into four terms—Term (i), Term (ii), Term (iii), and Term (iv)—as shown

in Eqn. (9). $\Theta(\log T)$ comes from regret incurred in the exploration phase with $\tau = Hn_l + 1$.

$$\text{Reg}_T = \sum_{t=1}^{T} \text{OPT}\left(\Theta, \mathbf{x}_t, \mathcal{F}^{\mathbf{x}_t}\right) - \mathbb{E}_{\pi_1, \pi_2, \ldots, \pi_T \sim \mathcal{G}}\left(\sum_{t=1}^{T} R\left(\pi_t; \mathbf{x}_t, \Theta, \mathcal{F}^{\mathbf{x}_t}\right)\right)$$

$$\leq \Theta(\log T) + \underbrace{\sum_{t=\tau}^{T} \text{OPT}\left(\Theta, \mathbf{x}_t, \mathcal{F}^{\mathbf{x}_t}\right) - \sum_{t=\tau}^{T} \text{OPT}\left(\Theta, \mathbf{x}_t, \hat{\mathcal{F}}_{t-1}^{\mathbf{x}_t}\right)}_{\text{Term (i)}} +$$

$$\underbrace{\sum_{t=\tau}^{T} \text{OPT}\left(\Theta, \mathbf{x}_t, \hat{\mathcal{F}}_{t-1}^{\mathbf{x}_t}\right) - \sum_{t=\tau}^{T} \text{OPT}\left(\mathcal{C}_{t-1}, \mathbf{x}_t, \hat{\mathcal{F}}_{t-1}^{\mathbf{x}_t}\right)}_{\text{Term (ii)}} +$$

$$\underbrace{\sum_{t=\tau}^{T} \text{OPT}\left(\mathcal{C}_{t-1}, \mathbf{x}_t, \hat{\mathcal{F}}_{t-1}^{\mathbf{x}_t}\right) - \mathbb{E}_{\pi_1, \pi_2, \ldots, \pi_T \sim \mathcal{G}}\left(\sum_{t=\tau}^{T} R\left(\pi_t; \mathbf{x}_t, \Theta, \hat{\mathcal{F}}_{t-1}^{\mathbf{x}_t}\right)\right)}_{\text{Term (iii)}} +$$

$$\underbrace{\mathbb{E}_{\pi_1, \pi_2, \ldots, \pi_T \sim \mathcal{G}}\left(\sum_{t=\tau}^{T} R\left(\pi_t; \mathbf{x}_t, \Theta, \hat{\mathcal{F}}_{t-1}^{\mathbf{x}_t}\right)\right) - \mathbb{E}_{\pi_1, \pi_2, \ldots, \pi_T \sim \mathcal{G}}\left(\sum_{t=\tau}^{T} R\left(\pi_t; \mathbf{x}_t, \Theta, \mathcal{F}^{\mathbf{x}_t}\right)\right)}_{\text{Term (iv)}}$$

At a high level, Terms (i) and (iv) account for the cumulative regret from transition error, i.e., the discrepancy between $\mathcal{F}^{\mathbf{x}_t}$ and $\hat{\mathcal{F}}^{\mathbf{x}_t}$. Term (iii) represents the decision error arising from imprecise estimates of the model parameters $\Theta = \{\boldsymbol{\theta}_l\}_{l \in \mathcal{H}} \cup \{d_l\}_{l \in [H]}$. Thus, more accurate estimators of $\mathcal{F}^{\mathbf{x}_t}$ and $\Theta$ yield lower cumulative regret. Meanwhile, Term (ii) measures the gap in regret between a chosen point and the optimal point within a feasible set. Our goal is to show $\Theta \in \mathcal{C}_t$ for all $t \geq \tau$ (after the exploration phase) with high probability, ensuring that Term (ii) can be negative with high probability. This is proved in Lemma 4.5. By Lemma 4.3, with probability at least $1 - \frac{1}{T^4}$, Term (i) in Eqn. (9) is bounded by $\widetilde{\mathcal{O}}\left(dH^2\sqrt{T}\right)$. A similar argument implies that Term (iv) is also bounded by $\widetilde{\mathcal{O}}\left(dH^2\sqrt{T}\right)$ with probability at least $1 - \frac{1}{T^4}$. Further, Lemma 4.4 shows that, with probability at least $1 - \frac{2}{T^3}$, Term (iii) is bounded by $\widetilde{\mathcal{O}}\left(H^2\sqrt{dT}\right)$. Combining these results, we conclude that, with probability at least $1 - \frac{6}{T^3}$, the overall regret $\text{Reg}_T$ is $\widetilde{\mathcal{O}}\left(dH^2\sqrt{T}\right)$. $\qquad\square$

## C.1 Proof for Theorem 4.1

*Theorem* (Theorem 4.1 restated). Let $\delta \geq \frac{1}{T^4 H}$ and $N_{t,l} \geq \underline{n}_l$. With probability at least $1 - \delta$, the estimation error $\left|\hat{d}_l^t - d_l\right|$, with $\hat{d}_l^t$ defined in Eqn. (3), is bounded by:

$$\left|\hat{d}_l^t - d_l\right| \leq \frac{4H\boldsymbol{B}_d\sqrt{d\log\left(1 + \frac{T}{2d}\right)\gamma} + \sqrt{2e\boldsymbol{B}_d\boldsymbol{B}_x\boldsymbol{B}_\theta \log\left(\frac{2}{\delta}\right)}}{\boldsymbol{b}\sqrt{N_{t,l}}}.$$

$\gamma$ is as defined in Lemma 3.3 and $\underline{n}_l$ defined in Remark 3.2.

*Proof.* The key idea in estimating $d_l$ is to isolate observations that are "purified" with respect to $d_l$. Define
$$\mathbf{D}_{t,l} := \{h \mid \boldsymbol{S}_{h,1}^t = l,\ \boldsymbol{o}_h^t = 0\},$$
which collects the episodes for user $t$ where: The second-to-last bid winning is $l$ episodes before the most recent bid winning ($\boldsymbol{S}_{h,1}^t = l$); The current bid is lost ($\boldsymbol{o}_h^t = 0$), meaning no new advertisement is shown. Under these conditions, the product-conversion observations $\{\mathbf{y}_h^t\}_{h \in \mathbf{D}_{t,l}}$ satisfy

$$\mathbf{y}_h^t \sim \text{Poi}\left(d_l \langle \boldsymbol{\theta}_{\boldsymbol{S}_{h,2}^t}, \mathbf{x}_t \rangle\right).$$

Because these observations are purified within the same $l$ that defines $d_l$, they provide reliable information for estimating the long-term advertising impact parameter $d_l$. To estimate the long-term advertising impact parameter $d_l$, we adopt a two-stage procedure. First, we construct the estimates

$\hat{\boldsymbol{\theta}}^s_{S^s_{h,2}}$ for all $s \in [t]$. Then, we substitute these estimates into the negative log-likelihood for $d_l$ (see Eqn. (11)), yielding our two-stage estimator for $d_l$.

$$\mathcal{L}(\mathbf{y}; \hat{\boldsymbol{\theta}}) = \sum_{s=1}^{t} \sum_{h \in \mathbf{D}_{s,l}} d_l \langle \hat{\boldsymbol{\theta}}^s_{S^s_{h,2}}, \mathbf{x}_s \rangle - \mathbf{y}^s_h \log \left( d_l \langle \hat{\boldsymbol{\theta}}^s_{S^s_{h,2}}, \mathbf{x}_s \rangle \right). \tag{11}$$

Taking the derivative of the negative log-likelihood with respect to $d_l$ yields:

$$\nabla_{d_l} \mathcal{L}(\mathbf{y}; \hat{\boldsymbol{\theta}}) = \sum_{s=1}^{t} \sum_{h \in \mathbf{D}_{s,l}} \langle \hat{\boldsymbol{\theta}}^s_{S^s_{h,2}}, \mathbf{x}_s \rangle - \frac{\mathbf{y}^s_h}{d_l} = 0 \tag{12}$$

This expression guides the next step of solving for $d_l$. By solving Eqn. (12), we obtain the estimator $\hat{d}^t_l$ for $d_l$ as follows:

$$\hat{d}^t_l = \frac{\sum_{s=1}^{t} \sum_{h \in \mathbf{D}_{s,l}} \mathbf{y}^s_h}{\sum_{s=1}^{t} \sum_{h \in \mathbf{D}_{s,l}} \langle \hat{\boldsymbol{\theta}}^s_{S^s_{h,2}}, \mathbf{x}_s \rangle}.$$

Here, $\hat{\boldsymbol{\theta}}^s_{S^s_{h,2}}$ denotes the updated estimate of the advertisement's impact for user $s$. Next, we bound the estimation error of $\hat{d}^t_l$. We express $\mathbf{y}^s_h$ as

$$\mathbf{y}^s_h = d_l \mathbf{x}^\top_s \boldsymbol{\theta}_{S^s_{h,2}} + \eta^s_h,$$

where $\eta^s_h$ is sub-exponential with parameters $(\nu^2, \alpha) = \left( ed_l \mathbf{x}^\top_s \boldsymbol{\theta}_{S^s_{h,2}}, 1 \right)$ (see Lemma D.1). Based on this formulation, the estimation error $\left| \hat{d}^t_l - d_l \right|$ can be decomposed as follows.

$$
\begin{aligned}
\left| \hat{d}^t_l - d_l \right| &= \left| \frac{\sum_{s=1}^{t} \sum_{h \in \mathbf{D}_{s,l}} \mathbf{y}^s_h}{\sum_{s=1}^{t} \sum_{h \in \mathbf{D}_{s,l}} \langle \hat{\boldsymbol{\theta}}^s_{S^s_{h,2}}, \mathbf{x}_s \rangle} - d_l \right| \\
&= \left| \frac{\sum_{s=1}^{t} \sum_{h \in \mathbf{D}_{s,l}} \left( d_l \mathbf{x}^\top_s \boldsymbol{\theta}_{S^s_{h,2}} + \eta^s_h \right)}{\sum_{s=1}^{t} \sum_{h \in \mathbf{D}_{s,l}} \langle \hat{\boldsymbol{\theta}}^s_{S^s_{h,2}}, \mathbf{x}_s \rangle} - d_l \right| \\
&= \left| \frac{\sum_{s=1}^{t} \sum_{h \in \mathbf{D}_{s,l}} \eta^s_h}{\sum_{s=1}^{t} \sum_{h \in \mathbf{D}_{s,l}} \langle \hat{\boldsymbol{\theta}}^s_{S^s_{h,2}}, \mathbf{x}_s \rangle} + d_l + \frac{d_l \sum_{s=1}^{t} \sum_{h \in \mathbf{D}_{s,l}} \mathbf{x}^\top_s \left( \boldsymbol{\theta}_{S^s_{h,2}} - \hat{\boldsymbol{\theta}}^s_{S^s_{h,2}} \right)}{\sum_{s=1}^{t} \sum_{h \in \mathbf{D}_{s,l}} \langle \hat{\boldsymbol{\theta}}^s_{S^s_{h,2}}, \mathbf{x}_s \rangle} - d_l \right| \\
&\leq \left| \frac{\sum_{s=1}^{t} \sum_{h \in \mathbf{D}_{s,l}} \eta^s_h}{\sum_{s=1}^{t} \sum_{h \in \mathbf{D}_{s,l}} \langle \hat{\boldsymbol{\theta}}^s_{S^s_{h,2}}, \mathbf{x}_s \rangle} \right| + d_l \left| \frac{\sum_{s=1}^{t} \sum_{h \in \mathbf{D}_{s,l}} \mathbf{x}^\top_s \left( \boldsymbol{\theta}_{S^s_{h,2}} - \hat{\boldsymbol{\theta}}^s_{S^s_{h,2}} \right)}{\sum_{s=1}^{t} \sum_{h \in \mathbf{D}_{s,l}} \langle \hat{\boldsymbol{\theta}}^s_{S^s_{h,2}}, \mathbf{x}_s \rangle} \right| \\
&\leq \underbrace{\left| \frac{\sum_{s=1}^{t} \sum_{h \in \mathbf{D}_{s,l}} \eta^s_h}{\sum_{s=1}^{t} \sum_{h \in \mathbf{D}_{s,l}} \langle \hat{\boldsymbol{\theta}}^s_{S^s_{h,2}}, \mathbf{x}_s \rangle} \right|}_{\text{\color{red}Term (i)}} + \underbrace{\frac{\boldsymbol{B}_d}{N_{t,l} \boldsymbol{b}} \left| \sum_{s=1}^{t} \sum_{h \in \mathbf{D}_{s,l}} \mathbf{x}^\top_s \left( \boldsymbol{\theta}_{S^s_{h,2}} - \hat{\boldsymbol{\theta}}^s_{S^s_{h,2}} \right) \right|}_{\text{\color{red}Term (ii)}}
\end{aligned}
$$

$$\tag{13}$$

To upper bound the estimation error (see Eqn. (13)), it suffices to bound Term (i) and Term (ii) in Eqn. (13) respectively. We can use sub-exponential concentration bound for Term (i), shown as follows.

$$
\begin{aligned}
\mathbb{P} \left( \left| \frac{\sum_{s=1}^{t} \sum_{h \in \mathbf{D}_{s,l}} \eta^s_h}{\sum_{s=1}^{t} \sum_{h \in \mathbf{D}_{s,l}} \langle \hat{\boldsymbol{\theta}}^s_{S^s_{h,2}}, \mathbf{x}_s \rangle} \right| > \epsilon \right) &\leq \mathbb{P} \left( \left| \frac{\sum_{s=1}^{t} \sum_{h \in \mathbf{D}_{s,l}} \eta^s_h}{N_{t,l} \boldsymbol{b}} \right| > \epsilon \right), \eta^s_h \sim \text{SubE} \left( ed_l \mathbf{x}^\top_s \boldsymbol{\theta}_{S^s_{h,2}}, 1 \right) \\
&\leq \mathbb{P} \left( \frac{1}{\boldsymbol{b}} \left| \frac{1}{N_{t,l}} \sum_{i=1}^{N_{t,l}} X_i \right| > \epsilon \right), X_i \sim \text{SubE}(e \boldsymbol{B}_d \boldsymbol{B}_x \boldsymbol{B}_\theta, 1) \\
&\leq \delta.
\end{aligned}
$$

$$\tag{14}$$

We derive Eqn. (14) by applying the tail bounds for sub-exponential random variables (Lemmas D.4, D.5, and D.6) and setting

$$\epsilon = \sqrt{\frac{2\,e\,\boldsymbol{B}_d\,\boldsymbol{B}_x\,\boldsymbol{B}_\theta}{\boldsymbol{b}^2\,N_{t,l}}\,\log(\tfrac{2}{\delta})}.$$

We also use the accelerated convergence rate for sub-exponential variables under the conditions $N_{t,l} > \frac{32\,\log(HT)}{e\,\boldsymbol{B}_d\,\boldsymbol{B}_x\,\boldsymbol{B}_\theta\,\boldsymbol{b}^2}$ and $\delta \geq \frac{1}{2HT^4}$, which ensure $0 \leq \epsilon \leq e\,\boldsymbol{B}_d\,\boldsymbol{B}_x\,\boldsymbol{B}_\theta$. Therefore, with probability at least $1-\delta$, we have Term (i) $\leq \sqrt{\frac{2\,e\,\boldsymbol{B}_d\,\boldsymbol{B}_x\,\boldsymbol{B}_\theta}{\boldsymbol{b}^2\,N_{t,l}}\,\log(\tfrac{2}{\delta})}$. Then it remains to bound Term (ii) in Eqn. (13). Define the dataset $\boldsymbol{F}^l_{s,l'} := \{\, h \mid S^s_{h,1} = l,\ S^s_{h,2} = l',\ \boldsymbol{o}^s_h = 0 \,\}$. Observe that

$$\sum_{l'=1}^{H} |\boldsymbol{F}^l_{s,l'}| = |\mathbf{D}_{s,l}|,$$

meaning $\boldsymbol{F}^l_{s,l'}$ partitions $\mathbf{D}_{s,l}$ according to $S^s_{h,2}$, the second-to-last winning state. Let $n_{s,l'} := |\boldsymbol{F}^l_{s,l'}|$. Then

$$\sum_{s=1}^{t}\sum_{l'=1}^{H} n_{s,l'} = N_{t,l}.$$

Since $\|\theta_l\|_2 \leq \boldsymbol{B}_\theta$ for all $l$, and $\|\mathbf{x}_s\|_2 \leq \boldsymbol{B}_x$ for each $s \in [T]$, define

$$r^s_h = \left| \left(\hat{\boldsymbol{\theta}}^s_{S^s_{h,2}} - \boldsymbol{\theta}_{S^s_{h,2}}\right)^\top \mathbf{x}_s \right| \leq \|\hat{\boldsymbol{\theta}}^s_{S^s_{h,2}} - \boldsymbol{\theta}_{S^s_{h,2}}\|_{\mathbf{V}^s_{l'}}\,\|\mathbf{x}_s\|_{(\mathbf{V}^s_{l'})^{-1}},$$

where $l' = S^s_{h,2}$. For convenience, let $\sqrt{\gamma^s_{l'}} = \|\hat{\boldsymbol{\theta}}^s_{S^s_{h,2}} - \boldsymbol{\theta}_{S^s_{h,2}}\|_{\mathbf{V}^s_{l'}}$. Then, with probability at least $1-\delta$, we have

$$\sum_{s=1}^{t}\sum_{h\in\mathbf{D}_{s,l}} \left| \left(\hat{\boldsymbol{\theta}}^s_{S^s_{h,2}} - \boldsymbol{\theta}_{S^s_{h,2}}\right)^\top \mathbf{x}_s \right| = \sum_{s=1}^{t}\sum_{h\in\mathbf{D}_{s,l}} r^s_h$$

$$\leq \sqrt{N_{t,l}}\sqrt{\sum_{s=1}^{t}\sum_{h\in\mathbf{D}_{s,l}} (r^s_h)^2}$$

$$= \sqrt{N_{t,l}}\sqrt{\sum_{s=1}^{t}\sum_{l'=1}^{H}\sum_{h\in\boldsymbol{F}^l_{s,l'}} (r^s_h)^2}$$

$$= \sqrt{N_{t,l}}\sqrt{\sum_{s=1}^{t}\sum_{l'=1}^{H} n_{s,l'}\gamma \min\left(\|\mathbf{x}_s\|_{(\mathbf{V}^s_{l'})^{-1}}, 1\right)}$$

$$= \sqrt{N_{t,l}}\sqrt{\sum_{l'=1}^{H}\sum_{s=1}^{t} n_{s,l'}\gamma \min\left(\|\mathbf{x}_s\|_{(\mathbf{V}^s_{l'})^{-1}}, 1\right)} \qquad (15)$$

$$\leq \sqrt{N_{t,l}}\sqrt{\sum_{l'=1}^{H}\gamma\sum_{s=1}^{t} n_{s,l'} \min\left(\|\mathbf{x}_s\|_{(\mathbf{V}^s_{l'})^{-1}}, 1\right)}$$

$$\leq \sqrt{N_{t,l}H}\sqrt{\sum_{l'=1}^{H}\gamma\sum_{s=1}^{t} \min\left(\|\mathbf{x}_s\|_{(\mathbf{V}^s_{l'})^{-1}}, 1\right)}$$

$$\leq H\sqrt{N_{t,l}}\sqrt{\gamma\max_{l'\in[H]}\sum_{s=1}^{t} \min\left(\|\mathbf{x}_s\|_{(\mathbf{V}^s_{l'})^{-1}}, 1\right)}$$

$$\leq 2H\sqrt{d}\sqrt{N_{t,l}\log\left(1+\frac{T}{2d}\right)\gamma}.$$

Therefore, with probability at least $1 - \delta$, $\delta \geq \frac{1}{HT^4}$, we have

$$|\hat{d}_l^t - d_l| \leq \frac{4H\boldsymbol{B}_d\sqrt{d\log\left(1 + \frac{T}{2d}\right)\gamma} + \sqrt{2e\boldsymbol{B}_d\boldsymbol{B}_x\boldsymbol{B}_\theta \log(2/\delta)}}{b} \frac{1}{\sqrt{N_{t,l}}}.$$

$\square$

## C.2 Proof of Lemma 4.3

*Lemma* (Lemma 4.3 restated). *For $\delta \geq 1/T^4$, with probability at least $1 - \delta$, Term (i) in Eqn. (9) is bounded above by $\mathcal{O}(dH^2\sqrt{T}\sqrt{\log((1+T)/\delta)\log(1+T/d)})$.*

*Proof.* In essence, the proof of Lemma 4.3 relies on two main steps. First, we show that Term (i) can be bounded by the cumulative estimation error in transition, expressed as

$$\sum_{t=1}^{T} \sup_b \sup_h \left|\hat{\mathcal{F}}_h^{t-1}(b, \mathbf{x}_t) - \mathcal{F}_h(b, \mathbf{x}_t)\right|.$$

Second, we demonstrate that, when $\hat{\mathcal{F}}_h^{t-1}(b, \mathbf{x}_t)$ is estimated using Algorithm 1, this cumulative estimation error remains small. Combined, these two steps yield the desired upper bound on Term (i).

**Step 1: Upper bound Term (i) by the cumulative estimation error in transition.**

Term (i) captures the regret induced by transition error, as shown in Eqn. (16). The final inequality in Eqn. (16) holds because $R(\pi_t^*; \Theta, \hat{\mathcal{F}}_{t-1}^{\mathbf{x}_t}) - \text{OPT}(\Theta, \mathbf{x}_t, \hat{\mathcal{F}}_{t-1}^{\mathbf{x}_t}) \leq 0$, due to the definition of $\text{OPT}(\Theta, \mathbf{x}_t, \hat{\mathcal{F}}_{t-1}^{\mathbf{x}_t})$. We then apply the Simulation Lemma (Lemma C.1) to bound $R(\pi_t^*; \Theta, \mathcal{F}^{\mathbf{x}_t}) - R(\pi_t^*; \Theta, \hat{\mathcal{F}}_{t-1}^{\mathbf{x}_t})$.

$$\text{Term (i)} = \sum_{t=\tau}^{T} \text{OPT}(\Theta, \mathbf{x}_t, \mathcal{F}^{\mathbf{x}_t}) - \sum_{t=\tau}^{T} \text{OPT}\left(\Theta, \mathbf{x}_t, \hat{\mathcal{F}}_{t-1}^{\mathbf{x}_t}\right)$$

$$= \sum_{t=\tau}^{T} R(\pi_t^*; \Theta, \mathcal{F}^{\mathbf{x}_t}) - R(\pi_t^*; \Theta, \hat{\mathcal{F}}_{t-1}^{\mathbf{x}_t}) + R(\pi_t^*; \Theta, \hat{\mathcal{F}}_{t-1}^{\mathbf{x}_t}) - \text{OPT}\left(\Theta, \mathbf{x}_t, \hat{\mathcal{F}}_{t-1}^{\mathbf{x}_t}\right) \quad (16)$$

$$\leq \sum_{t=\tau}^{T} R(\pi_t^*; \Theta, \mathcal{F}^{\mathbf{x}_t}) - R(\pi_t^*; \Theta, \hat{\mathcal{F}}_{t-1}^{\mathbf{x}_t}).$$

**Lemma C.1** (Simulation Lemma [Kearns and Singh, 2002]). *For all $s \in \mathcal{S}$ and $a \in \mathcal{A}$, if $\sum_{s'\in\mathcal{S}}\left|\mathbb{P}(s' \mid s, a) - \mathbb{P}'(s' \mid s, a)\right| \leq \epsilon_1$ and $\forall h \in [H]$, $\left|r_h(s, a) - r_h'(s, a)\right| \leq \epsilon_2$, then*

$$\left|\mathbb{E}_{\{s_h,a_h\}_{h=1}^H \sim \mathcal{M}, \pi}\left[\sum_{h\in[H]} r_h(s_h, a_h)\right] - \mathbb{E}_{\{s_h,a_h\}_{h=1}^H \sim \mathcal{M}', \pi}\left[\sum_{h\in[H]} r_h'(s_h, a_h)\right]\right| \leq \frac{H(H-1)\epsilon_1}{2} + H\epsilon_2.$$

To find $\epsilon_1$ in Lemma C.1, we compare two MDPs:

$$\mathcal{M}_t, \text{ induced by policy } \pi_t^*, \theta, \text{ and HOB distribution } \mathcal{F}^{\mathbf{x}_t},$$

and

$$\hat{\mathcal{M}}_t, \text{ induced by policy } \pi_t^*, \theta, \text{ and HOB distribution } \hat{\mathcal{F}}_{t-1}^{\mathbf{x}_t}.$$

Given state $s = (s_1, s_2)$, we have:

$$\sum_{s'\in S}\left|\mathbb{P}_h(s'|s,a) - \hat{\mathbb{P}}_h(s'|s,a)\right| = \left|\mathbb{P}_h(s' = (s_1 + 1, s_2)|s, a) - \hat{\mathbb{P}}_h(s' = (s_1 + 1, s_2)|s, a)\right|$$

$$+ \left|\mathbb{P}_h(s' = (1, s_1)|s, a) - \hat{\mathbb{P}}_h(s' = (1, s_1)|s, a)\right| \quad (17)$$

$$= 2\left|\hat{\mathcal{F}}_h^{t-1}(a, \mathbf{x}_t) - \mathcal{F}_h(a, \mathbf{x}_t)\right|$$

$$\leq 2\sup_b \sup_h \left|\hat{\mathcal{F}}_h^{t-1}(b, \mathbf{x}_t) - \mathcal{F}_h(b, \mathbf{x}_t)\right|.$$

Define

$$\epsilon_t = \sup_b \sup_h \left| \hat{\mathcal{F}}_h^{t-1}(b, \mathbf{x}_t) - \mathcal{F}_h(b, \mathbf{x}_t) \right|.$$

From Eqn. (17), it follows that the parameter $\epsilon_1$ in Lemma C.1 can be taken as $2\epsilon_t$. In the following, we bound the difference between $R_h^t$ and $R_h^{t'}$ to find $\epsilon_2$ in Lemma C.1. By definition, $R_h^t(S_h^t, A_h^t, \mathbf{x}_t) = d_{S_{h,1}^t}\langle \boldsymbol{\theta}_{S_{h,2}^t}, \mathbf{x}_t\rangle\left(1 - \mathcal{F}_h(A_h^t, \mathbf{x}_t)\right) + \left(\langle \boldsymbol{\theta}_{S_{h,1}^t}, \mathbf{x}_t\rangle - p_h(A_h^t, \mathbf{x}_t)\right)\mathcal{F}_h(A_h^t, \mathbf{x}_t)$. Here, $p_h(A_h^t, \mathbf{x}_t)$ is the second highest price conditioned on winning. Since $p_h^{\mathbf{x}_t}(b) = b - \frac{1}{\mathcal{F}_h(b, \mathbf{x}_t)}\int_0^b \mathcal{F}_h(v, \mathbf{x}_t)\,dv$, we obtain $\hat{p}_h(A_h^t, \mathbf{x}_t)\hat{\mathcal{F}}_h^{t-1}(A_h^t, \mathbf{x}_t) - p_h(A_h^t, \mathbf{x}_t)\mathcal{F}_h(A_h^t, \mathbf{x}_t) = \int_0^{A_h^t}\mathcal{F}_h(v, \mathbf{x}_t)\,dv - \int_0^{A_h^t}\hat{\mathcal{F}}_h^{t-1}(v, \mathbf{x}_t)\,dv$. In addition, we can show

$$\int_0^b f(v)dv - \int_0^b \hat{f}(v)dv \leq \int_0^b \sup_x |f(x) - \hat{f}(x)|dv$$

$$= \sup_x |f(x) - \hat{f}(x)| \int_0^b 1 dv$$

$$= b \sup_x |f(x) - \hat{f}(x)|.$$

Therefore, we can bound the reward difference $|R_h^t(s, a) - R_h^{t'}(s, a)|$ by Eqn. (18).

$$\left| R_h^t - R_h^{'t} \right| \leq \left( d_{S_{h,1}^t}\langle \boldsymbol{\theta}_{S_{h,2}^t}, \mathbf{x}_t\rangle + \langle \boldsymbol{\theta}_{S_{h,1}^t}, \mathbf{x}_t\rangle + \boldsymbol{B}_{\mathcal{A}} \right) \sup_{b \in [0, \boldsymbol{B}_{\mathcal{A}}]} \sup_h \left| \hat{\mathcal{F}}_h^{t-1}(b, \mathbf{x}_t) - \mathcal{F}_h(b, \mathbf{x}_t) \right|$$

$$\leq ((\boldsymbol{B}_d + 1)\boldsymbol{B}_x\boldsymbol{B}_\theta + \boldsymbol{B}_{\mathcal{A}}) \sup_{b \in [0, \boldsymbol{B}_{\mathcal{A}}]} \sup_h \left| \hat{\mathcal{F}}_h^{t-1}(b, \mathbf{x}_t) - \mathcal{F}_h(b, \mathbf{x}_t) \right| = ((\boldsymbol{B}_d + 1)\boldsymbol{B}_x\boldsymbol{B}_\theta + \boldsymbol{B}_{\mathcal{A}})\epsilon_t, \tag{18}$$

Combining Eqn. (17) and Eqn. (18), Lemma C.1 implies

$$R(\pi_t^*; \Theta, \mathcal{F}^{\mathbf{x}_t}) - R(\pi_t^*; \Theta, \hat{\mathcal{F}}_{t-1}^{\mathbf{x}_t}) \leq H(H-1)\epsilon_t + H((\boldsymbol{B}_d + 1)\boldsymbol{B}_x\boldsymbol{B}_\theta + \boldsymbol{B}_{\mathcal{A}})\epsilon_t$$

$$= H^2\epsilon_t + H\left((\boldsymbol{B}_d + 1)\boldsymbol{B}_x\boldsymbol{B}_\theta + \boldsymbol{B}_{\mathcal{A}} - 1\right)\epsilon_t. \tag{19}$$

Therefore, by Eqn. (19), bounding Term (i) reduces to bounding

$$\sum_{t=\tau}^T \left( H^2\,\epsilon_t + H\left((\boldsymbol{B}_d + 1)\,\boldsymbol{B}_x\,\boldsymbol{B}_\theta + \boldsymbol{B}_{\mathcal{A}} - 1\right)\epsilon_t \right).$$

**Step 2: Construct high probability bound for cumulative estimation error in transition**

The next step is to derive a high-probability bound for $\sum_{t=\tau}^T \epsilon_t$. By Assumption 2.6, the Highest Other Bids (HOB) distribution satisfies $\log(m_h^t) \sim \mathcal{N}\left(\langle \mathbf{x}_t, \boldsymbol{\beta}_h\rangle, \sigma_h^2\right)$, suggesting that the transition estimation error takes the form

$$\left| \hat{\mathcal{F}}_h^{t-1}(b, \mathbf{x}_t) - \mathcal{F}_h(b, \mathbf{x}_t) \right| = \left| \Phi\left( \frac{\log(b) - \langle \mathbf{x}_t, \hat{\boldsymbol{\beta}}_h^{t-1}\rangle}{\hat{\sigma}_h^{t-1}} \right) - \Phi\left( \frac{\log(b) - \langle \mathbf{x}_t, \boldsymbol{\beta}_h\rangle}{\sigma_h} \right) \right|. \tag{20}$$

Using a first-order Taylor expansion of the standard normal CDF $\Phi(\cdot)$, we obtain $\left| \Phi(a) - \Phi(a + \Delta) \right| \leq |\Delta|\phi(a)$, $\phi$ is the standard normal pdf. This inequality allows us to further bound the estimation error by $\left| \hat{\mathcal{F}}_h^{t-1}(b, \mathbf{x}_t) - \mathcal{F}_h(b, \mathbf{x}_t) \right| \leq |\Delta_h^t|\phi\left( \frac{\log(b) - \langle \mathbf{x}_t, \hat{\boldsymbol{\beta}}_h^{t-1}\rangle}{\hat{\sigma}_h^{t-1}} \right)$, where $|\Delta_h^t|$ is defined

and bounded by follows.

$$
\begin{aligned}
|\Delta_h^t| &= \left| \frac{\log(b) - \langle \mathbf{x}_t, \hat{\boldsymbol{\beta}}_h^{t-1} \rangle}{\hat{\sigma}_h^{t-1}} - \frac{\log(b) - \langle \mathbf{x}_t, \boldsymbol{\beta}_h \rangle}{\sigma_h} \right| \\
&= \left| \frac{\log(b) - \langle \mathbf{x}_t, \hat{\boldsymbol{\beta}}_h^{t-1} \rangle}{\hat{\sigma}_h^{t-1}} - \frac{\log(b) - \langle \mathbf{x}_t, \hat{\boldsymbol{\beta}}_h^{t-1} \rangle}{\sigma_h} + \frac{\log(b) - \langle \mathbf{x}_t, \hat{\boldsymbol{\beta}}_h^{t-1} \rangle}{\sigma_h} - \frac{\log(b) - \langle \mathbf{x}_t, \boldsymbol{\beta}_h \rangle}{\sigma_h} \right| \\
&\leq \frac{1}{\sigma_h} \left| \mathbf{x}_t^\top \left( \hat{\boldsymbol{\beta}}_h^{t-1} - \boldsymbol{\beta}_h \right) \right| + \left| \log(b) - \langle \mathbf{x}_t, \hat{\boldsymbol{\beta}}_h^{t-1} \rangle \right| \left| \frac{1}{\hat{\sigma}_h^{t-1}} - \frac{1}{\sigma_h} \right| \\
&\leq \frac{1}{\sigma_h} \left| \mathbf{x}_t^\top \left( \hat{\boldsymbol{\beta}}_h^{t-1} - \boldsymbol{\beta}_h \right) \right| + \frac{(\log(\boldsymbol{B}_{\mathcal{A}}) + \boldsymbol{B}_x \boldsymbol{B}_\beta)(\sigma_h + \hat{\sigma}_h^{t-1})}{\left( \sigma_h \hat{\sigma}_h^{t-1} \right)^3} \left| \sigma_h^2 - \left( \hat{\sigma}_h^{t-1} \right)^2 \right|.
\end{aligned}
$$

$$(21)$$

Note that we estimate the variance $\sigma_h^2$ using second-stage empirical estimates. Concretely,

$$
\left( \hat{\sigma}_h^{t-1} \right)^2 \;=\; \frac{1}{t-1} \sum_{s=1}^{t-1} \left( \log(m_h^s) - \mathbf{x}_s^\top \hat{\boldsymbol{\beta}}_h^s \right)^2.
$$

Meanwhile, by definition,

$$
\sigma_h^2 \;=\; \frac{1}{t-1} \sum_{s=1}^{t-1} \mathbb{E} \left[ \log(m_h^s) - \mathbf{x}_s^\top \boldsymbol{\beta}_h \right]^2.
$$

To complete the argument, it suffices to show that these two quantities are close with high probability.

$$
\begin{aligned}
\left| \sigma_h^2 - \left( \hat{\sigma}_h^{t-1} \right)^2 \right| &= \left| \frac{1}{t-1} \sum_{s=1}^{t-1} \left[ \left( \log(m_h^s) - \mathbf{x}_s^\top \hat{\boldsymbol{\beta}}_h^s \right)^2 - \mathbb{E} \left( \log(m_h^s) - \mathbf{x}_s^\top \boldsymbol{\beta}_h \right)^2 \right] \right| \\
&= \left| \frac{1}{t-1} \sum_{s=1}^{t-1} \left[ \left( \log(m_h^s) - \mathbf{x}_s^\top \boldsymbol{\beta}_h + \mathbf{x}_s^\top \boldsymbol{\beta}_h - \mathbf{x}_s^\top \hat{\boldsymbol{\beta}}_h^s \right)^2 - \mathbb{E} \left( \log(m_h^s) - \mathbf{x}_s^\top \boldsymbol{\beta}_h \right)^2 \right] \right| \\
&= \left| \frac{1}{t-1} \sum_{s=1}^{t-1} \left[ \left( \log(m_h^s) - \mathbf{x}_s^\top \boldsymbol{\beta}_h \right)^2 - \mathbb{E} \left( \log(m_h^s) - \mathbf{x}_s^\top \boldsymbol{\beta}_h \right)^2 \right] \right| \\
&\quad + \left| \frac{1}{t-1} \sum_{s=1}^{t-1} \left( \mathbf{x}_s^\top \boldsymbol{\beta}_h - \mathbf{x}_s^\top \hat{\boldsymbol{\beta}}_h^s \right)^2 \right| \\
&\quad + 2 \left| \frac{1}{t-1} \sum_{s=1}^{t-1} \left( \log(m_h^s) - \mathbf{x}_s^\top \boldsymbol{\beta}_h \right) \left( \mathbf{x}_s^\top \boldsymbol{\beta}_h - \mathbf{x}_s^\top \hat{\boldsymbol{\beta}}_h^s \right) \right|.
\end{aligned}
$$

Using Eqns. (20), (21), and (6), we derive the following equations:

$$
\begin{aligned}
\sum_{t=1}^T \epsilon_t \leq& \underbrace{\sum_{t=\tau}^T \sup_{h \in H} \frac{1}{\sigma_h} \left| \mathbf{x}_t^\top \left( \hat{\boldsymbol{\beta}}_h^{t-1} - \boldsymbol{\beta}_h \right) \right|}_{\text{Term A}} \\
&+ 2 (\log(\boldsymbol{B}_{\mathcal{A}}) + \boldsymbol{B}_x \boldsymbol{B}_\theta)^2 \underbrace{\sum_{t=1}^T \sup_{h \in H} C_h \left| \frac{1}{t-1} \sum_{s=1}^{t-1} \mathbf{x}_s^\top \left( \hat{\boldsymbol{\beta}}_h^s - \boldsymbol{\beta}_h \right) \right|}_{\text{Term B}} \\
&+ (\log(\boldsymbol{B}_{\mathcal{A}}) + \boldsymbol{B}_x \boldsymbol{B}_\theta) \underbrace{\sum_{t=1}^T \sup_{h \in H} \left| \frac{1}{t-1} \sum_{s=1}^{t-1} \left( \mathbf{x}_s^\top \boldsymbol{\beta}_h - \mathbf{x}_s^\top \hat{\boldsymbol{\beta}}_h^s \right)^2 \right|}_{\text{Term C}} \\
&+ (\log(\boldsymbol{B}_{\mathcal{A}}) + \boldsymbol{B}_x \boldsymbol{B}_\theta) \underbrace{\sum_{t=1}^T \left| \frac{1}{t-1} \sum_{s=1}^{t-1} \left[ \left( \log(m_h^s) - \mathbf{x}_s^\top \boldsymbol{\beta}_h \right)^2 - \mathbb{E} \left( \log(m_h^s) - \mathbf{x}_s^\top \boldsymbol{\beta}_h \right)^2 \right] \right|}_{\text{Term D}},
\end{aligned}
$$

$$(22)$$

where $C_h = \frac{\sigma_h + \hat{\sigma}_h^{t-1}}{(\sigma_h \hat{\sigma}_h^{t-1})^3} \leq \frac{2\bar{\sigma}}{\underline{\sigma}^6}$, by Assumption 2.8. In what follows, we bound each term one by one. To bound Term A, we show that for all $h$, with probability at least $1 - \delta$, the following holds:

$$\sum_{t=1}^{T} \left| \mathbf{x}_t^\top \left( \hat{\boldsymbol{\beta}}_h^{t-1} - \boldsymbol{\beta}_h \right) \right| \leq \sqrt{T \sum_{t=1}^{T} \left| \mathbf{x}_t^\top \left( \hat{\boldsymbol{\beta}}_h^{t-1} - \boldsymbol{\beta}_h \right) \right|^2}$$

$$\leq \sqrt{T \sum_{t=1}^{T} \min \left\{ 2 \boldsymbol{B}_\beta \boldsymbol{B}_x, \|\mathbf{x}_t\|_{\left(\Sigma_h^{t-1}\right)^{-1}}^2 \|\hat{\boldsymbol{\beta}}_h^{t-1} - \boldsymbol{\beta}_h\|_{\Sigma_h^{t-1}}^2 \right\}}$$

$$\leq \sqrt{T \sum_{t=1}^{T} \|\hat{\boldsymbol{\beta}}_h^{t-1} - \boldsymbol{\beta}_h\|_{\Sigma_h^{t-1}}^2 \min \left\{ \frac{2 \boldsymbol{B}_\beta \boldsymbol{B}_x}{\|\hat{\boldsymbol{\beta}}_h^{t-1} - \boldsymbol{\beta}_h\|_{\Sigma_h^{t-1}}^2}, \|\mathbf{x}_t\|_{\left(\Sigma_h^{t-1}\right)^{-1}}^2 \right\}}$$

$$\leq \left( \sigma_h^2 \sqrt{d \log \left( \frac{1 + T \boldsymbol{B}_x^2 / \lambda}{\delta} \right)} + \sqrt{\lambda} \boldsymbol{B}_\beta \right) \sqrt{T \sum_{t=1}^{T} \min \left\{ 1, \|\mathbf{x}_t\|_{\left(\Sigma_h^{t-1}\right)^{-1}}^2 \right\}}, \text{ by Lemma D.2}$$

$$\leq \left( \sigma_h^2 \sqrt{d \log \left( \frac{1 + T \boldsymbol{B}_x^2 / \lambda}{\delta} \right)} + \sqrt{\lambda} \boldsymbol{B}_\beta \right) \sqrt{T d \log \left( 1 + \frac{T \boldsymbol{B}_x}{\lambda d} \right)}$$

$$\leq d\sqrt{T} \sup_{h \in H} \left( \sigma_h^2 \sqrt{\log \left( \frac{1 + T \boldsymbol{B}_x^2 / \lambda}{\delta} \right)} + \sqrt{\lambda} \boldsymbol{B}_\beta \right) \sqrt{\log \left( 1 + \frac{T \boldsymbol{B}_x}{\lambda d} \right)}.$$

Therefore,

$$\sum_{t=1}^{T} \sup_{h \in H} \frac{1}{\sigma_h} |\mathbf{x}_t^\top (\hat{\boldsymbol{\beta}}_h^{t-1} - \boldsymbol{\beta}_h)| \leq d\sqrt{T} \left( \frac{\bar{\sigma}^2}{\underline{\sigma}} \sqrt{\log \left( \frac{1 + \frac{T\boldsymbol{B}_x^2}{\lambda}}{\delta} \right)} + \sqrt{\lambda} \boldsymbol{B}_\beta \right) \sqrt{\log \left( 1 + \frac{T\boldsymbol{B}_x}{\lambda d} \right)}.$$

This shows that Term C is on the order of $\mathcal{O}(\log T)$. For Term B, a similar argument implies that, with probability at least $1 - \delta$,

$$\left| \frac{1}{t-1} \sum_{s=1}^{t-1} \mathbf{x}_s^\top (\hat{\boldsymbol{\beta}}_h^s - \boldsymbol{\beta}_h) \right| \leq \frac{d}{\sqrt{t-1}} \left( \frac{\bar{\sigma}^2}{\underline{\sigma}} \sqrt{\log \left( \frac{1 + \frac{T\boldsymbol{B}_x^2}{\lambda}}{\delta} \right)} + \sqrt{\lambda} \boldsymbol{B}_\beta \right) \sqrt{\log \left( 1 + \frac{T\boldsymbol{B}_x}{\lambda d} \right)}.$$

Therefore, we conclude the following bound.

$$\sum_{t=\tau}^{T} \sup_{h \in H} C_h \left| \frac{1}{t-1} \sum_{s=1}^{t-1} \mathbf{x}_s^\top \left( \hat{\boldsymbol{\beta}}_h^s - \boldsymbol{\beta}_h \right) \right| \leq \frac{2\bar{\sigma}}{\underline{\sigma}^6} d\sqrt{T} \left( \frac{\bar{\sigma}^2}{\underline{\sigma}} \sqrt{\log \left( \frac{1 + T\boldsymbol{B}_x^2 / \lambda}{\delta} \right)} + \sqrt{\lambda} \boldsymbol{B}_\beta \right) \sqrt{\log \left( 1 + \frac{T\boldsymbol{B}_x}{\lambda d} \right)}.$$

It remains to bound Term D. First, observe that

$$\left[ \left( \log(m_h^s) - \mathbf{x}_s^\top \boldsymbol{\beta}_h \right)^2 - \mathbb{E} \left( \log(m_h^s) - \mathbf{x}_s^\top \boldsymbol{\beta}_h \right)^2 \right] = \sigma_h^2 \left( \frac{\left( \log(m_h^s) - \mathbf{x}_s^\top \boldsymbol{\beta}_h \right)^2}{\sigma_h^2} - 1 \right) = \sigma_h^2 \left( \chi_1^2 - 1 \right).$$

Since $\chi_1^2 \sim \text{SubE}(4, 4)$, by Lemma C.2 and a union bound argument, we can show that, with probability at least $1 - \delta$, Term D is bounded above by

$$\sqrt{T} \sqrt{8 \log \left( \frac{2T}{\delta} \right)} + \mathcal{O} \left( \log(T) \right).$$

The $\mathcal{O} \left( \log(T) \right)$ term arises from "burning in" the first $64 \log(T)$ which guarantees $\delta \geq 1/4T^4$ and ensure quadratic decay of sub-exponential bound.

**Lemma C.2** (Example 2.11 in Wainwright [2019]). *A chi-squared ($\chi^2$) random variable with $n$ degrees of freedom, denoted by $Y \sim \chi_n^2$, can be represented as $Y = \sum_{k=1}^{n} Z_k^2$, where $Z_k \sim N(0, 1)$ are i.i.d. variables. Since each $Z_k^2$ is sub-exponential with parameters $(\nu^2, \alpha) = (2, 4)$, we have $\mathbb{P} \left( \left| \frac{1}{n} \sum_{k=1}^{n} Z_k^2 - 1 \right| \geq \sqrt{\frac{8}{n} \log \frac{2}{\delta}} \right) \leq \delta$, as long as $\delta \geq 2 \exp \left( -\frac{n}{8} \right)$.*

Bringing all these results together and applying union bound, we conclude that there exists a constant $C$, depending on $\boldsymbol{B}_x, \boldsymbol{B}_\theta, \boldsymbol{B}_\beta, \bar{\sigma}, \underline{\sigma}, \boldsymbol{B}_d$, and $\boldsymbol{B}_\mathcal{A}$, such that with probability at least $1 - \delta$ (where $\delta \geq \frac{1}{T^4}$), we have

$$\text{Term (i)} \ \leq \ C\, d\, H^2\, \sqrt{T}\, \sqrt{\log\Big(\frac{1+T}{\lambda\,\delta}\Big)\, \log\Big(1 + \frac{T}{\lambda\, d}\Big)}.$$

By choosing $\lambda = 1$, we proves Lemma 4.3, we states that for $\delta \geq 1/T^4$, with probability at least $1 - \delta$, Term (i) in Eqn. (9) is bounded above by $\mathcal{O}(dH^2\sqrt{T}\sqrt{\log((1+T)/\delta)\log(1+T/d)})$. $\qquad \square$

### C.3    Proof for Lemma 4.4

*Lemma* (Lemma 4.4 restated). With probability at least $1 - 2\delta$, where $\delta \geq \frac{1}{T^3}$, Term (iii) in Eqn. (9) is bounded by $\mathcal{O}\Big(H^2\sqrt{dT\log\big(\frac{4TH}{\delta}\big)}\,\log\big(1 + \frac{T}{2d}\big)\Big)$.

*Proof.* Term (iii) reflects the portion of regret attributable to inaccuracies in estimating the true parameter $\Theta$. Let $\tilde{\Theta}_{t-1} := \arg\max_{\Theta\in\mathcal{C}_{t-1}} R(\pi_t; \mathbf{x}_t, \Theta, \hat{\mathcal{F}}_{t-1}^{\mathbf{x}_t})$. By definition,

$$R(\pi_t; \mathbf{x}_t, \tilde{\Theta}_{t-1}, \hat{\mathcal{F}}_{t-1}^{\mathbf{x}_t}) = \text{OPT}\big(\mathcal{C}_{t-1}, \mathbf{x}_t, \hat{\mathcal{F}}_{t-1}^{\mathbf{x}_t}\big).$$

By Eqn. (23), we demonstrate that Term (iii) in (9) can be split into two parts corresponding to the cumulative estimation error of $\hat{\theta}_l^t$ (see Terms 1 and 2 in (23)) and the cumulative estimation error of $\hat{d}_l$ (see Term 3 in (23)). Recall that $\hat{\theta}_l^t$ is a variant of the online Newton estimator (see Algorithm 3), defined by

$$\hat{\theta}_l^t = \arg\min_{\|\theta\|_2\in\boldsymbol{B}_\theta}\Big\{\tfrac{1}{2}\|\theta - \hat{\theta}_l^{t-1}\|_{\mathbf{V}_l^t}^2 + \langle\theta - \hat{\theta}_l^{t-1}, \nabla\tilde{l}_t(\hat{\theta}_l^{t-1})\rangle\Big\},$$

where $\nabla\tilde{l}_t(\hat{\theta}_l^{t-1}) = (-\tilde{\mathbf{y}}_h^t + \mathbf{x}_t^\top\hat{\theta}_l^{t-1})\mathbf{x}_t$ and $\mathbf{V}_l^t = \mathbf{V}_l^{t-1} + \frac{1}{2}\mathbf{x}_t\mathbf{x}_t^\top$. $\tilde{\mathbf{y}}_h^t$ is the truncated observation of $\mathbf{y}_h^t$. Because $\mathbf{y}_h^t$ follows a Poisson distribution and is linear in $\theta_l$, it is a special case of a heavy-tailed distribution with bounded second moment. Consequently, Lemma 3.3 can be used to construct a high-confidence bound on $\hat{\theta}_l^t$, which in turn controls Terms 1 and 2 in Eqn. (23).

$$\text{Term (iii)} = \sum_{t=\tau}^{T} \text{OPT}\left(\mathcal{C}_{t-1}, \mathbf{x}_t, \hat{\mathcal{F}}_{t-1}^{\mathbf{x}_t}\right) - \mathbb{E}_{\pi_1,\pi_2,\ldots,\pi_T \sim \mathcal{G}}\left(\sum_{t=\tau}^{T} R\left(\pi_t; \mathbf{x}_t, \Theta, \hat{\mathcal{F}}_{t-1}^{\mathbf{x}_t}\right)\right)$$

$$= \mathbb{E}_{\pi_1,\pi_2,\ldots,\pi_T \sim \mathcal{G}}\left[\sum_{t=\tau}^{T} R\left(\pi_t; \mathbf{x}_t, \tilde{\Theta}_{t-1}, \hat{\mathcal{F}}_{t-1}^{\mathbf{x}_t}\right) - R\left(\pi_t; \mathbf{x}_t, \Theta, \hat{\mathcal{F}}_{t-1}^{\mathbf{x}_t}\right)\right]$$

$$= \mathbb{E}_{\pi_1,\pi_2,\ldots,\pi_T \sim \mathcal{G}}\left[\sum_{t=\tau}^{T} \mathbb{E}_{S_h^t \sim \hat{P}_{t-1}^{\mathbf{x}_t}}\left(\sum_{h=1}^{H} R_h^t(S_h^t, A_h^t, \mathbf{x}_t)\right) - \sum_{t=\tau}^{T} R\left(\pi_t; \mathbf{x}_t, \Theta, \hat{\mathcal{F}}_{t-1}^{\mathbf{x}_t}\right)\right]$$

$$= \mathbb{E}\left(\sum_{t=\tau}^{T} \mathbb{E}\left(\sum_{h=1}^{H} \tilde{d}_{S_{h,1}^t}^{t-1}\langle\tilde{\boldsymbol{\theta}}_{S_{h,2}^t}^{t-1}, \mathbf{x}_t\rangle(1 - \hat{\mathcal{F}}_h^{t-1}(A_h^t, \mathbf{x}_t)) + (\langle\tilde{\boldsymbol{\theta}}_{S_{h,1}^t}^{t-1}, \mathbf{x}_t\rangle - \hat{p}_h(A_h^t, \mathbf{x}_{t-1}))\hat{\mathcal{F}}_h^{t-1}(A_h^t, \mathbf{x}_t))\right)\right)$$

$$- \mathbb{E}_{\pi_1,\pi_2,\ldots,\pi_T \sim \mathcal{G}}\left(\sum_{t=\tau}^{T} R\left(\pi_t; \mathbf{x}_t, \Theta, \hat{\mathcal{F}}_{t-1}^{\mathbf{x}_t}\right)\right), \text{where } A_h^t = \pi_t(S_h^t, \mathbf{x}_t)$$

$$\leq \sum_{t=\tau}^{T} \mathbb{E}_{S_h^t \sim \hat{P}_{t-1}^{\mathbf{x}_t}}\left(\sum_{h=1}^{H} \left|\langle\tilde{\boldsymbol{\theta}}_{S_{h,1}^t}^{t-1} - \boldsymbol{\theta}_{S_{h,1}^t}, \mathbf{x}_t\rangle\right| + \left|d_{S_{h,1}^t}\langle\boldsymbol{\theta}_{S_{h,2}^t}, \mathbf{x}_t\rangle - \tilde{d}_{S_{h,1}^t}^{t-1}\langle\tilde{\boldsymbol{\theta}}_{S_{h,2}^t}^{t-1}, \mathbf{x}_t\rangle\right|\right)$$

$$\leq \sum_{t=\tau}^{T} \mathbb{E}_{S_h^t \sim \hat{P}_{t-1}^{\mathbf{x}_t}}\left(\sum_{h=1}^{H} \left\{\left|\langle\tilde{\boldsymbol{\theta}}_{S_{h,1}^t}^{t-1} - \boldsymbol{\theta}_{S_{h,1}^t}, \mathbf{x}_t\rangle\right| + \boldsymbol{B}_\mathcal{A}\left|\langle\tilde{\boldsymbol{\theta}}_{S_{h,2}^t}^{t-1} - \boldsymbol{\theta}_{S_{h,2}^t}, \mathbf{x}_t\rangle\right| + \left|\tilde{d}_{S_{h,1}^t}^{t-1} - d_{S_{h,1}^t}\right|\boldsymbol{B}_\theta\boldsymbol{B}_x\right\}\right)$$

$$= \underbrace{\sum_{t=\tau}^{T} \mathbb{E}\left(\sum_{h=1}^{H} \left|\langle\tilde{\boldsymbol{\theta}}_{S_{h,1}^t}^{t-1} - \hat{\boldsymbol{\theta}}_{S_{h,1}^t}^{t-1} + \hat{\boldsymbol{\theta}}_{S_{h,1}^t}^{t-1} - \boldsymbol{\theta}_{S_{h,1}^t}, \mathbf{x}_t\rangle\right|\right)}_{\text{Term 1}}$$

$$+ \boldsymbol{B}_\mathcal{A}\underbrace{\sum_{t=\tau}^{T} \mathbb{E}\left(\sum_{h=1}^{H} \left|\langle\tilde{\boldsymbol{\theta}}_{S_{h,2}^t}^{t-1} - \hat{\boldsymbol{\theta}}_{S_{h,2}^t}^{t-1} + \hat{\boldsymbol{\theta}}_{S_{h,2}^t}^{t-1} - \boldsymbol{\theta}_{S_{h,2}^t}, \mathbf{x}_t\rangle\right|\right)}_{\text{Term 2}} +$$

$$+ \boldsymbol{B}_\theta\boldsymbol{B}_x\underbrace{\sum_{t=\tau}^{T} \mathbb{E}\left(\sum_{h=1}^{H} \left|\tilde{d}_{S_{h,1}^t}^{t-1} - \hat{d}_{S_{h,1}^t}^{t-1} + \hat{d}_{S_{h,1}^t}^{t-1} - d_{S_{h,1}^t}\right|\right)}_{\text{Term 3}}.$$

$$(23)$$

To bound Term 1 in Eqn. (23), we first introduce $n_t^l$, where

$$n_t^l := \left|\{h \in [H] : S_{h,1}^t = l\}\right|; n_t^l \leq H.$$

Then we have the following inequality:

$$\sum_{t=\tau}^{T} \mathbb{E}\left(\sum_{h=1}^{H}\left|\langle\tilde{\boldsymbol{\theta}}_{S_{h,1}^{t}}^{t-1} - \hat{\boldsymbol{\theta}}_{S_{h,1}^{t}}^{t-1} + \hat{\boldsymbol{\theta}}_{S_{h,1}^{t}}^{t-1} - \boldsymbol{\theta}_{S_{h,1}^{t}}, \mathbf{x}_t\rangle\right|\right) = \mathbb{E}\left(\sum_{t=\tau}^{T}\sum_{h=1}^{H}\left|\langle\tilde{\boldsymbol{\theta}}_{S_{h,1}^{t}}^{t-1} - \hat{\boldsymbol{\theta}}_{S_{h,1}^{t}}^{t-1} + \hat{\boldsymbol{\theta}}_{S_{h,1}^{t}}^{t-1} - \boldsymbol{\theta}_{S_{h,1}^{t}}, \mathbf{x}_t\rangle\right|\right)$$

$$\leq 2\mathbb{E}\left(\sum_{t=\tau}^{T}\sum_{l\in\mathcal{H}} n_t^l \sqrt{\gamma_l^{t-1}}\|\mathbf{x}_t\|_{(\mathbf{V}_l^{t-1})^{-1}}\right) \tag{24}$$

$$\leq 2H\sum_{t=\tau}^{T}\sum_{l\in\mathcal{H}}\sqrt{\gamma_l^{t-1}}\|\mathbf{x}_t\|_{(\mathbf{V}_l^{t-1})^{-1}} \tag{25}$$

$$\leq 2H\sum_{l\in\mathcal{H}}\sqrt{\sum_{t=\tau}^{T}\gamma_l^{t-1}}\sqrt{\sum_{t=\tau}^{T}\|\mathbf{x}_t\|_{(\mathbf{V}_l^{t-1})^{-1}}^2} \tag{26}$$

$$\leq \sqrt{d}H^2\mathcal{O}\left(\sqrt{T\log\left(\frac{4TH}{\delta}\right)}\log\left(1+\frac{T}{2d}\right)\right) \tag{27}$$

Let us denote $\sqrt{\gamma_{l'}^s} := \left\|\hat{\boldsymbol{\theta}}_{S_{h,2}^s}^s - \boldsymbol{\theta}_{S_{h,2}^s}\right\|_{\mathbf{V}_{l'}^s}$. We obtain Eqn. (24) by rearranging terms in the expression involving $\boldsymbol{\theta}_l$ with same $l$. We use the fact that $n_t^l \leq H$ to get Eqn. (25). Next, we apply the Cauchy–Schwarz inequality to derive Eqn. (26). By Lemma 3.3, we know that, with probability at least $1 - \delta$, $\gamma_{l'}^s \leq \gamma$ for all $t \geq 1$, with $\gamma$ defined in Lemma 3.3. In addition, we use Lemma 11 of Abbasi-Yadkori et al. [2011] to bound $\sum_{t=1}^{T}\|\mathbf{x}_t\|_{(\mathbf{V}_l^t)^{-1}}^2$, yielding

$$\sum_{t=1}^{T}\|\mathbf{x}_t\|_{(\mathbf{V}_l^t)^{-1}}^2 \leq 4\log\left(\frac{\det(\mathbf{V}_{T+1})}{\det(\mathbf{V}_1)}\right) \leq 4d\log\left(1+\frac{T}{2d}\right).$$

Combining these results and applying the union bound gives us Eqn. (27). Similarly, by an analogous argument, we can show that, with probability at least $1 - \delta$, Term 2 in Eqn. (23) satisfies a similar bound, as shown below.

$$\sum_{t=\tau}^{T}\mathbb{E}\left(\sum_{h=1}^{H}\left|\langle\tilde{\boldsymbol{\theta}}_{S_{h,2}^{t}}^{t-1} - \hat{\boldsymbol{\theta}}_{S_{h,2}^{t}}^{t-1} + \hat{\boldsymbol{\theta}}_{S_{h,2}^{t}}^{t-1} - \boldsymbol{\theta}_{S_{h,2}^{t}}, \mathbf{x}_t\rangle\right|\right) \leq \sqrt{d}H^2\mathcal{O}\left(\sqrt{T\log\left(\frac{4TH}{\delta}\right)}\log\left(1+\frac{T}{2d}\right)\right).$$

The principal difficulty in designing Algorithm 1 lies in estimating $\hat{d}_l^t$, which captures the long-term impact of advertising on product conversion. Since $d_l$ always appears alongside $\boldsymbol{\theta}_{l'}$ in the model $\mathbf{y}_h^t \sim \text{Poi}(d_l\,\boldsymbol{\theta}_{l'}^\top\mathbf{x}_t)$, the main technical challenge is establishing a high-probability bound on $|\hat{d}_l^t - d_l|$ that leverages the estimation error of $\hat{\boldsymbol{\theta}}_{l'}^t$. Theorem 4.1 summarizes our results in this direction. By Theorem 4.1, we know that given $t \geq H\underline{n}_l$, and $l \in \mathcal{H}_1$, with probability $1 - \delta, \delta \geq \frac{1}{T^4 H}$, $d_l \in \mathcal{D}_l^t$, where

$$\mathcal{D}_l^t = \left\{d \in [0, \boldsymbol{B}_d] \mid |\hat{d}_l^t - d| \leq \frac{4H\,\boldsymbol{B}_d\sqrt{d\log\left(1+\frac{T}{2d}\right)\gamma} + \sqrt{2\,e\,\boldsymbol{B}_d\,\boldsymbol{B}_x\,\boldsymbol{B}_\theta\,\log\left(\frac{2}{\delta}\right)}}{b}\frac{1}{\sqrt{N_{t,l}}}\right\}.$$

Therefore, by applying the union bound to make results valid $\forall t \in [T]$ and $t \geq H\underline{n}_l, \forall l \in \mathcal{H}_1$, we have that for $\delta \geq \frac{1}{T^3}$, we conclude that with probability at least $1 - \delta$, Term (3) in Eqn. (23) can be bounded above by

$$\sum_{t=\tau}^{T} \mathbb{E}\left(\sum_{h=1}^{H} \left| \tilde{d}_{S_{h,1}^t}^{t-1} - \hat{d}_{S_{h,1}^t}^{t-1} + \hat{d}_{S_{h,1}^t}^{t-1} - d_{S_{h,1}^t} \right| \right)$$

$$\leq \frac{8H\, \boldsymbol{B}_d \sqrt{d \, \log\!\left(1 + \frac{T}{2d}\right)\gamma} + \sqrt{2\,e\,\boldsymbol{B}_d\,\boldsymbol{B}_x\,\boldsymbol{B}_\theta\,\log\!\left(\frac{2}{\delta}\right)}}{b} \mathbb{E}\left(\sum_{l=1}^{H}\sum_{t=1}^{N_{T,l}} \frac{1}{\sqrt{t}}\right) + \mathcal{O}(\log(HT)) \quad (28)$$

$$\leq \mathcal{O}\left(H^2\sqrt{dT}\log\left(1 + \frac{T}{2d}\right)\sqrt{\log\left(\frac{4HT}{\delta}\right)\log\left(\frac{2H}{\delta}\right)}\right) \quad (29)$$

Equation (28) follows by gathering all terms involving the same $l$ of $d_l$. Next, we obtain Equation (29) by noting that $\sum_{l=1}^{H} N_{T,l} = T\,H$. Combining the above results, we conclude that, with probability at least $1 - 2\delta$ (where $\delta \geq \frac{1}{T^3}$), Term (iii) in Eqn. (9) is bounded by

$$\mathcal{O}\!\left(H^2\sqrt{d\,T\,\log(\tfrac{4\,T\,H}{\delta})}\,\log\!\left(1 + \tfrac{T}{2\,d}\right)\right).$$

$\square$

## C.4 Proof of Lemma 4.5

*Lemma* (Lemma 4.5 restated). With probability at least $1 - \delta$ with $\delta \geq \frac{2}{T^3}$, $\Theta \in \mathcal{C}_t, \forall t \geq \tau$.

*Proof.*

$$\begin{aligned}
\mathbb{P}(\Theta \in \mathcal{C}_t, \forall t \geq \tau) &= \mathbb{P}(\{\forall t \geq \tau, \forall l \in \mathcal{H}, \boldsymbol{\theta}_l \in \mathcal{C}_l^t\} \cap \{\forall t \geq \tau, \forall l \in \mathcal{H}_1, d_l \in \mathcal{D}_l^t\}) \\
&= 1 - \mathbb{P}\left\{ \left(\{\forall t \geq \tau, \forall l \in \mathcal{H}, \boldsymbol{\theta}_l \in \mathcal{C}_l^t\} \cap \{\forall t \geq \tau, \forall l \in \mathcal{H}_1, d_l \in \mathcal{D}_l^t\}\right)^c \right\} \\
&= 1 - \mathbb{P}\left\{ \left(\{\forall t \geq \tau, \forall l \in \mathcal{H}, \boldsymbol{\theta}_l \in \mathcal{C}_l^t\}\right)^c \cup \left(\{\forall t \geq \tau, \forall l \in \mathcal{H}_1, d_l \in \mathcal{D}_l^t\}\right)^c \right\} \\
&\geq 1 - \mathbb{P}\left(\left(\{\forall t \geq \tau, \forall l \in \mathcal{H}, \boldsymbol{\theta}_l \in \mathcal{C}_l^t\}\right)^c\right) - \mathbb{P}\left(\left(\{\forall t \geq \tau, \forall l \in \mathcal{H}_1, d_l \in \mathcal{D}_l^t\}\right)^c\right)
\end{aligned}$$

By Lemma 3.3, we have

$$\mathbb{P}\left(\left(\{\forall t \geq \tau, \forall l \in \mathcal{H}, \boldsymbol{\theta}_l \in \mathcal{C}_l^t\}\right)^c\right) \leq \sum_{l \in \mathcal{H}} \mathbb{P}(\exists t \geq \tau \text{ s.t. } \boldsymbol{\theta}_l \notin \mathcal{C}_l^t) \leq \delta.$$

Moreover, by Theorem 4.1, with $\delta \geq \frac{1}{HT^4}$ we have

$$\mathbb{P}\left(\left(\{\forall t \geq \tau, \forall l \in \mathcal{H}, d_l \in \mathcal{D}_l^t\}\right)^c\right) \leq \sum_{l \in \mathcal{H}_1}\sum_{t \in [T]} \mathbb{P}(d_l^t \notin \mathbf{D}_l^t) \leq HT\delta.$$

Combining these results together, we have with probability at least $1 - \delta$ with $\delta \geq \frac{2}{T^3}$, $\Theta \in \mathcal{C}_t, \forall t$. $\square$

# D  Technical Lemmas

**Lemma D.1** (Sub-Exponentiality of Poisson Random Variable). *If $X \sim \text{Poi}(\mu)$, then the centered random variable $X - \mu$ is $\text{SubE}(e\mu, 1)$. Equivalently, there exist constants $\nu^2 = e\mu$ and $\alpha = 1$ such that, for all $|t| < 1/\alpha = 1$, $\mathbb{E}\left[e^{t\,(X-\mu)}\right] \leq \exp\!\left(\nu^2\,t^2\right) = \exp\!\left(e\,\mu\,t^2\right).$*

*Proof.* Below is the proof showing that a $\text{Poisson}(\mu)$ random variable is sub-exponential with parameters $\left(e\mu,\, 1\right)$. The proof hinges on bounding the moment-generating function (MGF) of the centered random variable $X - \mu$. Recall that if $X \sim \text{Poi}(\mu)$, its moment-generating function (MGF) is

$$\mathbb{E}\left[e^{tX}\right] = \exp\!\left(\mu(e^t - 1)\right).$$

For the centered random variable $X - \mu$,

$$\mathbb{E}\left[e^{t(X-\mu)}\right] = e^{-t\mu}\,\mathbb{E}\left[e^{tX}\right] = \exp\!\left(\mu\left(e^t - 1 - t\right)\right).$$

We claim that, for all $|t| \leq 1$,
$$e^t - 1 - t \ \leq \ e\,t^2.$$
Indeed, expanding $e^t$ in its Taylor series about $t = 0$,
$$e^t \ = \ 1 + t + \frac{t^2}{2} + \frac{t^3}{6} + \ldots \quad \implies \quad e^t - 1 - t \ = \ \frac{t^2}{2} + \frac{t^3}{6} + \ldots$$
For $|t| \leq 1$, this sum of higher-order terms is bounded above by a constant times $t^2$. A convenient choice is $e$, giving
$$e^t - 1 - t \ \leq \ e\,t^2.$$
Combining the two steps, for $|t| \leq 1$,
$$\mathbb{E}\big[e^{t(X-\mu)}\big] \ = \ \exp\big(\mu(e^t - 1 - t)\big) \ \leq \ \exp\big(\mu \cdot e\,t^2\big).$$
Therefore, $X - \mu$ satisfies
$$\mathbb{E}\big[e^{t(X-\mu)}\big] \ \leq \ \exp\big(e\,\mu\,t^2\big) \quad \text{for all } |t| < 1.$$
By definition, this means $X - \mu$ is $\mathrm{SubE}\big(e\mu, 1\big)$. $\qquad\square$

**Lemma D.2** (Theorem 2 in Abbasi-Yadkori et al. [2011]). *Assume the same in Theorem D.3, let $\Sigma_0 = \lambda \mathbb{I}_d, \lambda > 0$. Define $\mathbf{y}_t = \mathbf{x}_t^\top \boldsymbol{\beta} + \eta_t$, with $\eta_t$ defined in Lemma D.3, and assume that $\|\boldsymbol{\beta}\|_2 \leq \boldsymbol{B}_\beta, \|\mathbf{x}_t\|_2 \leq \boldsymbol{B}_x$. Then, for any $\delta > 0$, with probability at least $1 - \delta$, for all $t > 0$, $\beta$ lies in the set*
$$\boldsymbol{C}_t = \left\{ \boldsymbol{\beta} \in \mathbb{R}^d : \|\hat{\boldsymbol{\beta}}^t - \boldsymbol{\beta}\|_{\Sigma_t} \leq \sigma^2 \sqrt{d \log\left(\frac{1 + t\boldsymbol{B}_x^2/\lambda}{\delta}\right)} + \sqrt{\lambda}\boldsymbol{B}_\beta \right\},$$
*where $\hat{\boldsymbol{\beta}}^t := \left(\sum_{s=1}^{t} \mathbf{x}_s \mathbf{x}_s^\top + \lambda \mathbb{I}_d\right)^{-1} \left(\sum_{s=1}^{t} \mathbf{x}_s \mathbf{y}_s\right)$.*

**Lemma D.3** (Theorem 1 in Abbasi-Yadkori et al. [2011]). *Let $\{\mathcal{F}_t\}_{t=0}^\infty$ be a filtration. Let $\{\eta_t\}_{t=1}^\infty$ be a real-valued stochastic process such that $\eta_t$ is $\mathcal{F}_t$-measurable and $\eta_t$ is conditionally $\sigma^2$-sub-Gaussian for some $\sigma^2 \geq 0$. Let $\{\mathbf{x}_t\}_{t=1}^\infty$ be an $\mathbb{R}^d$-valued stochastic process such that $\mathbf{x}_t$ is $\mathcal{F}_{t-1}$ measurable. Assume that $\Sigma_0$ is a $d \times d$ positive definite matrix. For any $t \geq 0$, define $\Sigma_t = \Sigma_{t-1} + \mathbf{x}_t \mathbf{x}_t^\top$ and $S_t = \sum_{s=1}^t \eta_s \mathbf{x}_s$. Then for any $\delta > 0$, with probability at least $1 - \delta$, for all $t \geq 0$, we have $\|S_t\|_{\Sigma_t^{-1}}^2 \leq 2\sigma^4 \log\left(\sqrt{\frac{\det \Sigma_t}{\det \Sigma_0}}/\delta\right)$.*

**Lemma D.4** (Sub-exponential tail bound). *Suppose $X$ is sub-exponential with parameters $(\nu^2, \alpha)$. Then*
$$\mathbb{P}(|X - \mu| > t) \leq \begin{cases} 2\exp(-\frac{t^2}{2\nu^2}), & \text{if } 0 \leq t \leq \frac{\nu^2}{\alpha} \\ 2\exp(-\frac{t}{2\alpha}). & \text{for } t > \frac{\nu^2}{\alpha} \end{cases}$$

**Lemma D.5** (Lemma 3 in Hao et al. [2021]). *Consider a random variable $X_i \sim SE(\nu^2, \alpha)$ and $\beta$ is a non-zero scalar, then $\beta X_i \sim SE(\beta^2 \nu^2, |\beta|\alpha)$*

**Lemma D.6** (Lemma 4 in Hao et al. [2021]). *Consider independent random variables $X_i \sim SE(\nu_i^2, \alpha_i)$ for $i = 1, \ldots, n$, then $X = \sum_{i=1}^n X_i$ follows $SE\left(\sum_{i=1}^n \nu_i^2, \max_i \alpha_i\right)$.*

### D.1 Proof of Fact 2.7

*Proof.* In this section, we prove that the tuple $(\mathcal{X}, \mathcal{S}, \mathcal{A}, \mathcal{P}^{\mathbf{x}_t}, \{R_h^t(S_h^t, \mathbf{a}_h^t, \mathbf{x}_t)\}_{h=1}^H, s_1, H)$ constitutes a Contextual Markov decision process (CMDP) by demonstrating that it satisfies the Markov property.

If we lose the bid, then we have
$$\mathbb{P}\Big( R_{h+1}^t = d_{S_{h+1,1}^t} \langle \boldsymbol{\theta}_{S_{h+1,2}^t}, \mathbf{x}_t \rangle (1 - \mathcal{F}_{h+1}(\mathbf{a}_{h+1}^t, \mathbf{x}_t)) + \Big( \langle \boldsymbol{\theta}_{S_{h+1,1}^t}, \mathbf{x}_t \rangle - p_{h+1}(\mathbf{a}_{h+1}^t, \mathbf{x}_t) \Big) \mathcal{F}_{h+1}(\mathbf{a}_{h+1}^t, \mathbf{x}_t),$$
$$S_{h+1}^t = (S_{h,1}^t + 1, S_{h,2}^t) | S_0^t, \mathbf{a}_0^t, R_1^t, \ldots, S_h^t, \mathbf{a}_h^t \Big) = 1 - \mathcal{F}_h(\mathbf{a}_h^t, \mathbf{x}_t).$$

If we win the bid, then we have

$$\mathbb{P}\Big(R_{h+1}^t = d_{S_{h+1,1}^t}\langle\boldsymbol{\theta}_{S_{h+1,2}^t}, \mathbf{x}_t\rangle(1 - \mathcal{F}_{h+1}(\mathbf{a}_{h+1}^t, \mathbf{x}_t)) + \Big(\langle\boldsymbol{\theta}_{S_{h+1,1}^t}, \mathbf{x}_t\rangle - p_{h+1}(\mathbf{a}_{h+1}^t, \mathbf{x}_t)\Big)\mathcal{F}_{h+1}(\mathbf{a}_{h+1}^t, \mathbf{x}_t),$$

$$S_{h+1}^t = (1, S_{h,1}^t)|S_0^t, \mathbf{a}_0^t, R_1^t, \ldots, S_h^t, \mathbf{a}_h^t\Big) = \mathcal{F}_h(\mathbf{a}_h^t, \mathbf{x}_t).$$

From the above equation, we notice that

$$\mathbb{P}\left(R_{h+1}^t, S_{h+1}^t|S_0^t, \mathbf{a}_0^t, \boldsymbol{r}_1^t, \ldots, S_h^t, \mathbf{a}_h^t\right) = \mathbb{P}\left(R_{h+1}^t, S_{h+1}^t|S_h^t, \mathbf{a}_h^t\right).$$

Therefore, the Markov property holds and the model is a CMDP. □

