# OpenReview forum: "Learning Personalized Ad Impact via Contextual Reinforcement Learning under Delayed Rewards"
_NeurIPS.cc/2025/Conference — NeurIPS 2025 poster_

### Official Review · Reviewer_iRrk · 2025-07-01

**Clarity:** 3
**Significance:** 2
**Originality:** 2
**Rating:** 3
**Confidence:** 3

**Summary:**

This paper presents a novel approach to personalized ad bidding that simultaneously addresses delayed rewards, cumulative ad effects, and customer heterogeneity. The authors model this complex environment as a CMDP with delayed Poisson rewards, where context is used for personalization and states capture the history of ad exposure. A core contribution is the development of a two-stage maximum likelihood estimator (TS-MLE) combined with a data-splitting strategy, designed to efficiently and accurately estimate the delayed impacts of advertising. Based on this estimation framework, the paper proposes a reinforcement learning algorithm that learns personalized bidding strategies and is proven to achieve a near-optimal regret bound.

**Questions:**

1. The inner product often contains additional parenthesis and sometimes missing comma like in line 161

**Ethical Concerns:**

["NO or VERY MINOR ethics concerns only"]

**Final Justification:**

I have read authors' rebuttal and will remain my initial evaluation

**Quality:**

3

**Strengths And Weaknesses:**

Strength
1. The paper studied an important and interesting question in real-world ads bidding. All the three components have been extensively studied in the context of Ads CTR prediction task but not well studied under the auto bidding setting.
2. The authors provide solid theoretical guarantees on the regret bound and proved that the proposed algorithm achieves near optimal regret.

Weakness
1. The proposed model is not realistic compared to real-world Ads applications. A fixed and simple linear structure is assumed to model cumulative effects while in CTR modeling sequence models are used. Moreover, the heterogeneity not only comes from user and context but also comes from different types of ads.
2. The algorithm and the theoretical analysis relies on potentially strong assumptions that may be hard to generalize for example the state definition, highest competing bids follow a lognormal distribution, the specific reward function formula.
3. The paper is entirely theoretical and lacks any experimental results to validate the proposed algorithms. Without experiments, even on synthetic data, it is difficult to assess the algorithm's practical performance, scalability, and robustness to assumption violations.

---

> ### Author Rebuttal · Authors · 2025-07-31
>
> We sincerely appreciate the reviewer’s appreciation of our work, especially about its theoretical strength. We also appreciate the reviewer for pointing out some of the weaknesses of our work. We will address them one by one as follows and hope the reviewer will reconsider our rating.
>
> - We acknowledge that linear models may be overly simplistic for some real-world applications. However, more complex models, such as deep sequence models, often lack theoretical guarantees. To the best of our knowledge, our work is the first to provide a theoretical framework that jointly accounts for long-term effects, cumulative delayed impacts, and customer heterogeneity—three critical aspects that have not been addressed together in prior literature. Indeed, this point is also noted in your comment. Our current model can incorporate the ad effect. In particular, for the HOB modeling, we still just use $x_t$, since we do not know what types of ad the other bidders will show. But we can use a feature map say, $u_t=\phi(x_t, ad_t)$, to model the demand, via $u^{\top}_t \theta$. The feature map can be as simple as $(x_t, ad_t)$ a concatenation. Extending our framework to move beyond linear structures is an important direction for future work.
>
> - Our state space modeling is highly flexible and can be easily extended to incorporate domain knowledge (see Appendix A.1 for details). We assume the highest competing bids follow a lognormal distribution, which is a standard choice in the literature. This assumption can be generalized to any flexible parametric distribution such as Gamma distribution. Regarding the reward structure, our formulation aligns with the common assumption in the literature that rewards are generated via a known linear feature mapping. Our model can also be seamlessly extended from a linear model to a GLM model, i.e. $\psi(x^{\top}_t \theta)$, where $\psi$ is a known link function such as logistic or exponential function.
>
> - Following your recommendation, we conducted a set of synthetic experiments that support two key points:
> Our algorithm empirically achieves near-optimal performance, with regret scaling as $\sqrt{T}$, where $T$ is the number of customers.
> It significantly outperforms several baseline policies, including an aggressive bidding strategy (which is close to the optimal strategy under our simulation setting), a passive bidding strategy, and a random bidding strategy.
>
> **Environment Setup**: We simulate a second-price auction with horizon $H = 3$ (a reasonable horizon length since in reality advertisers show their ads only a few times to each Internet user), context dimension $D = 2$, and $T = 20,000$ sequential customers. The first 2,400 rounds are for exploration; the rest for exploitation. Context vectors $x_t \in \mathbb{R}^2, d_l, \beta_h, \sigma_h$ are sampled elementwise from $|\mathcal{N}(0,1)| + 0.1$, ensuring positivity. Parameters $\theta_l  \sim 5 \cdot |\mathcal{N}(0,1)| + 0.1$. Note that under our current simulation setting, an impression brings on average an instantaneous reward that is around 5 times than its cost (i.e. HOB price). Moreover, the reward of the impression decays quickly with time. These suggest that an aggressive bidding strategy (i.e. try to always win the bid and get the impression) is actually close to the optimal bidding strategy and thus serves as a competitive baseline policy.
>
> **Estimators**: We estimate four categories of parameters. $\theta_l$ is learned using an online Newton method (Algorithm 3) with truncation threshold $\tau = 100{,}000$, bound $B_\theta = 10$, zero initialization, and $V_0 = I$. The delay impact $d_l$ is estimated via a two-stage MLE (Eq. 3) using $D_{t,l}$, $\hat{\theta}_l$, and $x_t$. $\beta_h$ is estimated by ridge regression (Eq. 5, $\lambda = 1.0$), and $\sigma_h$ via empirical variance (Eq. 6). As $\theta_l$ and $d_l$ are the most critical, we provide the corresponding code in the appendix.
>
> **Connection Between Bid Amount, Auction Outcome, and Realized Reward**: In our setting, the bidder's action is the bid amount, but due to the second-price auction format, the reward depends only on the outcome–whether the bid exceeds the highest other bid (HOB)-not the bid itself. Winning results in paying the HOB; losing incurs no cost. This decoupling allows optimal rewards to be computed from outcomes rather than bids.
> We leverage this to evaluate both oracle and policy-based rewards. With full knowledge of true parameters, we enumerate all $2^H = 8$ possible win/loss outcomes across $H = 3$ rounds and define the **oracle reward** as the maximum achievable. With estimated parameters, we compute the expected reward for each outcome and choose the best sequence (Algorithm 1), which is then run through the simulator to obtain the realized reward. Regret is defined as the cumulative gap between oracle and actual rewards, reflecting errors from estimation and suboptimal bidding.
>
> **Simulation and Regret**
> 1. Sample ground-truth parameters $(\theta_l, d_l, \beta_h, \sigma_h)$, which remain fixed. Initialize estimates $(\hat{\theta}_l, \hat{d}_l, \hat{\beta}_h, \hat{\sigma}_h)$ using default values (e.g., zeros or identity).
> 2. Simulate $T = 20{,}000$ customers over two phases: exploration (first 2,400 rounds) and exploitation (remaining 17,600 rounds).
> 3. Exploration Phase: Uniformly sample each of the $2^H = 8$ bidding outcome sequences across 2,400 rounds (300 per outcome). For each round: Sample context $x_t$; simulate reward and HOB under the chosen bidding outcome; Compute and record the oracle reward; Store observations for estimation; Apply data splitting to update parameter estimates.
> 4. Exploitation Phase: For each round: Observe $x_t$; compute estimated rewards for all 8 bidding sequences using current estimates; Select the best sequence; simulate reward; compute oracle reward and reward gap; Update estimates via **data splitting**.
> 5. Return cumulative regret as $np.cumsum(reward_{diff})$.
>
> **Baseline Policies**: We compare against three baselines. **Aggressive  bidding:** This simulates the strategy that the firm actively engages in winning ad impressions by consistently offering the highest bidding price (and pays second best price). In other words, the firm always wins by bidding above the HOB $(o_t = [1, 1, 1])$; **Bidding at random:** Sample $o_t$ uniformly from all 8 bidding outcomes and set bids accordingly; **Passive bidding:** This simulates the strategy that the firm rarely engages in the ad impression bidding war, where it gives very low bidding or simply does not bid $(o_t = [0, 0, 0])$.
>
> Regret for each is computed using the same oracle-based metric, serving as benchmarks for our algorithm.
> We remark here again that due to our simulation setting, the aggressive bidding strategy is close to the optimal strategy. Therefore, it serves as a competitive benchmark, though it obviously suffers a O(T) regret since it is a fixed policy without any learning.
>
> We run our proposed algorithm alongside the three baseline policies over 20 independent trials. The table below reports the average cumulative regret with ±0.5 standard deviation for all four algorithms. The last row shows the estimated regret order, obtained by applying a log-log transformation followed by linear regression. As shown, our algorithm achieves a fitted regret order of 0.37, while all three baselines exhibit linear regret. These results are consistent with our theoretical analysis and demonstrate the superior performance of our approach and the importance of learning.
> | Round   | Our Algorithm                 | Aggressive Bid              | Random Bid                   | Passive Bid                    |
> |---------|----------------------------|-----------------------------|------------------------------|-------------------------------|
> | 500     | 1770 [1174, 2379]          | 228 [121, 336]              | 1920 [1791, 2049]            | 4630 [3391, 5869]             |
> | 5000    | 8465 [5585, 11345]         | 2373 [1243, 3502]           | 19285 [17873, 20697]         | 46179 [33850, 58508]          |
> | 10000   | 8484 [5606, 11363]         | 4741 [2477, 7005]           | 38399 [35573, 41226]         | 92468 [67735, 117201]         |
> | 15000   | 8505 [5628, 11382]         | 7142 [3724, 10561]          | 57651 [53452, 61849]         | 138652 [101576, 175729]       |
> | 20000   | 8525 [5649, 11401]         | 9568 [4991, 14146]          | 76945 [71390, 82500]         | 184873 [135414, 234332]       |
> | **Fitted Order** | **0.37**                 | **1.0**                    | **1.0**                      | **1.0**                        |
>
>
> ---
> **Appendix**
>
> For Class ThetaEstimator, Class DelayImpactEstimator and data_splitting function, please refers to rebuttal for **Reviewer uxuX** Due to space constraints, additional codes and results are available upon request.

---

### Official Review · Reviewer_Xfce · 2025-07-02

**Clarity:** 3
**Significance:** 3
**Originality:** 3
**Rating:** 5
**Confidence:** 3

**Summary:**

Motivated by the widely documented phenomenon of *recency bias*, the authors propose a reinforcement learning algorithm for efficient auto-bidding/ad placement which takes into consideration cumulative and delayed effects of prior advertising on user behaviour, rather than focusing solely on short-term engagement signals. Specifically, the auto-bidding decision process is modelled as a contextual MDP with delayed rewards represented through a Poisson distribution. The contextual factor allows for personalisation, accounting for preferences, seasonal factors etc., whilst the state captures the combined, long-term effect of multiple ads on the user over time. The authors note that prior work by Badanidiyuru et al. [2023] explores modeling the long-term effects of ads on users also via MDPs with Poisson rewards, however they assume homogenous treatment effects across the user-base, effectively ignoring personalisation as a factor in the outcome. This represents the first theoretical work on the design of bidding algorithms which addresses this factor alongside modeling delayed and cumulative ad impacts on users.

A three-stage process is formulated for solving these MDPs. Firstly, an *exploration* stage, where ridge regression is used with variance estimators to estimate the transition dynamics. Secondly, an *exploitation* stage takes place which involves estimation of the advertisement effects. Finally, a two-stage likelihood estimator is developed by the authors in order to estimate the delayed advertisement effects. The latter approach is required due to two fundamental challenges, the non-concave nature of the log-likelihood, and the lack of closed form solutions to the scoring equations.

The contributions of the paper are theoretical, with the authors presenting proof of a near optimal regret bound for their algorithm. Evaluation on real-world data is left to future work, owing to the lack of publicly available online bidding datasets.

**Questions:**

**Q1** - Regarding the point raised in *Weaknesses* above regarding the background literature, could the authors please extend the discussion of prior work in the area in either the main text or appendix through additional text or a table briefly detailing competing approaches and their properties?

**Q2** - As also mentioned prior, could the authors expand slightly on the impact of the decision on how the state should be formulated, and if possible, how much benefit extending the state from one to two or more recent ad exposures may have on performance? I understand here that this cannot yet be assessed empirically, however if there is other prior work which may point to an answer to this question, that would be useful detail to include.

These minor points would certainly improve the paper and I encourage the authors to address these points, however I feel that the contribution is strong regardless, and therefore recommend acceptance.

**Ethical Concerns:**

["NO or VERY MINOR ethics concerns only"]

**Final Justification:**

I keep my original score unchanged, as whilst the clarifications and results provided by the authors in the rebuttal would improve the paper, they do not change my overall assessment of the work.

**Limitations:**

Yes

**Quality:**

4

**Strengths And Weaknesses:**

**Strengths**
* Overall, the theoretical analysis carried out by the authors is thorough, and results in a near-optimal regret bound for online contextual RL with delayed Poisson rewards, which is a high quality and novel contribution to the literature. This contribution lays the groundwork for future work to explore the practical implementation of such an algorithm in real-world settings with online bidding datasets, likely by organisations which have access to such data on a large scale, as well as for additional theoretical contributions on considerations relevant to large-scale deployment such as budget constraints.
* The authors present and introduce the background and problem setting clearly. Early in the paper, the work is placed clearly in the context of prior work in the area, with an intuitive explanation given of where this work offers a novel contribution compared to this prior work.
* Whilst it’s difficult to categorically assign a very high level of significance to an approach which has been analysed theoretically before it has been evaluated in a real-world deployment setting, this approach does seem to have the potential for considerable impact. As the authors note themselves, this problem of learning to bid is highly challenging for a number of reasons, such as the heterogeneity of preferences across user bases on online services, the delayed nature of any feedback on such services, and the cumulative nature by which advertising impacts the behaviour of end users. The development of an effective approach to bidding which simultaneously tackles all of these concerns is a considerable step forward.

**Weaknesses**
* Whilst the discussion of background and prior work in the Introduction is clearly outlined, it is somewhat brief, limited to effectively around a single paragraph. Whilst of course space is always a limitation, I feel it would be helpful to include (or at least point to in the appendix), a slightly extended discussion or table listing the other referenced approaches to learning to bid, including brief details about the core of their approach and any obvious advantages/disadvantages relative to the proposed method.
* One question which arose whilst reading the section discussing the state formulation was that whilst I fully understand the motivation for using only very recent ad exposures due to recency bias, I wondered whether there is any support/discussion in the literature surrounding only focusing on the single most recent advertising event. This is presented as the default choice for the algorithm in the paper, so although we’re not able to empirically assess the algorithm to check, if possible it would be useful to have some further discussion of how much useful information is gained by for example, extending the state formulation to the previous two ad exposures versus just the most recent one.

---

> ### Author Rebuttal · Authors · 2025-07-31
>
> We sincerely appreciate the reviewer’s positive feedback on the originality of our work and the contribution of our theoretical development. We address the reviewer’s weakness and questions as below.
>
> **Adding a Discussion of Related Work in “Learning-to-Bid”**
>
> We will add additional literature reviews following the suggestions. Due to space constraints in the rebuttal, we will include the additional references and discussion in the appendix of the revised manuscript.
>
> **On recency bias and impact of state formulation**
>
> As we noted in the paper (Line 122), recency-based attribution is pretty much the default method in the industry for evaluating ad effectiveness, such as Google Auto Bidding. This idea is also supported by several psychology studies (Lines 118–120) that show people tend to be more influenced by recent experiences when making decisions. To the best of our knowledge, “Incrementality Bidding via Reinforcement Learning under Mixed and Delayed Rewards” is the only published work in “Learning-to-Bid” literature that incorporates a notion of “state” when modeling ad impact. However, that work assumes customer homogeneity and does not account for heterogeneity—one of the key limitations that our paper seeks to address.
>
> We provide a detailed discussion in Appendix 1 on how the state formulation can be extended beyond the most recent ad exposure to incorporate domain knowledge. For instance, the state can include $n^t_h$, the total number of ads a customer has seen before round $h$. This opens the door to modeling richer exposure histories, such as the impact of last $k$ ad exposures. However, we acknowledge that directly modeling ad impact based on the past $k$ exposures is highly nontrivial. It introduces additional complexity due to mixture and delayed rewards as well as customer heterogeneity, making estimation significantly more difficult, and will leave it for future work.
>
> **Experiments**
>
> As suggested by other reviewers, over this rebuttal period, we conducted a simulation to empirically verify our proposed algorithm, which we show that (1) our algorithm achieves near-optimal regret scaling of $\sqrt{T}$, and (2) it substantially outperforms several baseline policies such as aggressive  bidding, passive bidding, and a bidding at random strategy. We include the results here for your reference.
>
> **Environment Setup**: We simulate a second-price auction with horizon $H = 3$ (a reasonable horizon length since in reality advertisers show their ads only a few times to each Internet user), context dimension $D = 2$, and $T = 20,000$ sequential customers. The first 2,400 rounds are for exploration; the rest for exploitation. Context vectors $x_t \in \mathbb{R}^2, d_l, \beta_h, \sigma_h$ are sampled elementwise from $|\mathcal{N}(0,1)| + 0.1$, ensuring positivity. Parameters $\theta_l  \sim 5 \cdot |\mathcal{N}(0,1)| + 0.1$. Note that under our current simulation setting, an impression brings on average an instantaneous reward that is around 5 times than its cost (i.e. HOB price). Moreover, the reward of the impression decays quickly with time. These suggest that an aggressive bidding strategy (i.e. try to always win the bid and get the impression) is actually close to the optimal bidding strategy and thus serves as a competitive baseline policy.
>
> **Estimators**: We estimate four categories of parameters. $\theta_l$ is learned using an online Newton method (Algorithm 3) with truncation threshold $\tau = 100{,}000$, bound $B_\theta = 10$, zero initialization, and $V_0 = I$. The delay impact $d_l$ is estimated via a two-stage MLE (Eq. 3) using $D_{t,l}$, $\hat{\theta}_l$, and $x_t$. $\beta_h$ is estimated by ridge regression (Eq. 5, $\lambda = 1.0$), and $\sigma_h$ via empirical variance (Eq. 6). As $\theta_l$ and $d_l$ are the most critical, we provide the corresponding code in the appendix.
>
> **Connection Between Bid Amount, Auction Outcome, and Realized Reward**: In our setting, the bidder's action is the bid amount, but due to the second-price auction format, the reward depends only on the outcome–whether the bid exceeds the highest other bid (HOB)-not the bid itself. Winning results in paying the HOB; losing incurs no cost. This decoupling allows optimal rewards to be computed from outcomes rather than bids.
> We leverage this to evaluate both oracle and policy-based rewards. With full knowledge of true parameters, we enumerate all $2^H = 8$ possible win/loss outcomes across $H = 3$ rounds and define the **oracle reward** as the maximum achievable. With estimated parameters, we compute the expected reward for each outcome and choose the best sequence (Algorithm 1), which is then run through the simulator to obtain the realized reward. Regret is defined as the cumulative gap between oracle and actual rewards, reflecting errors from estimation and suboptimal bidding.
>
> **Simulation and Regret**
> 1. Sample ground-truth parameters $(\theta_l, d_l, \beta_h, \sigma_h)$, which remain fixed. Initialize estimates $(\hat{\theta}_l, \hat{d}_l, \hat{\beta}_h, \hat{\sigma}_h)$ using default values (e.g., zeros or identity).
> 2. Simulate $T = 20{,}000$ customers over two phases: exploration (first 2,400 rounds) and exploitation (remaining 17,600 rounds).
> 3. Exploration Phase: Uniformly sample each of the $2^H = 8$ bidding outcome sequences across 2,400 rounds (300 per outcome). For each round: Sample context $x_t$; simulate reward and HOB under the chosen bidding outcome; Compute and record the oracle reward; Store observations for estimation; Apply data splitting to update parameter estimates.
> 4. Exploitation Phase: For each round: Observe $x_t$; compute estimated rewards for all 8 bidding sequences using current estimates; Select the best sequence; simulate reward; compute oracle reward and reward gap; Update estimates via **data splitting**.
> 5. Return cumulative regret as $np.cumsum(reward_{diff})$.
>
> **Baseline Policies**: We compare against three baselines. **Aggressive  bidding:** This simulates the strategy that the firm actively engages in winning ad impressions by consistently offering the highest bidding price (and pays second best price). In other words, the firm always wins by bidding above the HOB $(o_t = [1, 1, 1])$; **Bidding at random:** Sample $o_t$ uniformly from all 8 bidding outcomes and set bids accordingly; **Passive bidding:** This simulates the strategy that the firm rarely engages in the ad impression bidding war, where it gives very low bidding or simply does not bid $(o_t = [0, 0, 0])$.
>
> Regret for each is computed using the same oracle-based metric, serving as benchmarks for our algorithm.
> We remark here again that due to our simulation setting, the aggressive bidding strategy is close to the optimal strategy. Therefore, it serves as a competitive benchmark, though it obviously suffers a O(T) regret since it is a fixed policy without any learning.
>
> We run our proposed algorithm alongside the three baseline policies over 20 independent trials. The table below reports the average cumulative regret with ±0.5 standard deviation for all four algorithms. The last row shows the estimated regret order, obtained by applying a log-log transformation followed by linear regression. As shown, our algorithm achieves a fitted regret order of 0.37, while all three baselines exhibit linear regret. These results are consistent with our theoretical analysis and demonstrate the superior performance of our approach and the importance of learning.
> | Round   | Our Algorithm                 | Aggressive Bid              | Random Bid                   | Passive Bid                    |
> |---------|----------------------------|-----------------------------|------------------------------|-------------------------------|
> | 500     | 1770 [1174, 2379]          | 228 [121, 336]              | 1920 [1791, 2049]            | 4630 [3391, 5869]             |
> | 5000    | 8465 [5585, 11345]         | 2373 [1243, 3502]           | 19285 [17873, 20697]         | 46179 [33850, 58508]          |
> | 10000   | 8484 [5606, 11363]         | 4741 [2477, 7005]           | 38399 [35573, 41226]         | 92468 [67735, 117201]         |
> | 15000   | 8505 [5628, 11382]         | 7142 [3724, 10561]          | 57651 [53452, 61849]         | 138652 [101576, 175729]       |
> | 20000   | 8525 [5649, 11401]         | 9568 [4991, 14146]          | 76945 [71390, 82500]         | 184873 [135414, 234332]       |
> | **Fitted Order** | **0.37**                 | **1.0**                    | **1.0**                      | **1.0**                        |
>
>
> ---
> **Appendix**
>
> For Class ThetaEstimator, Class DelayImpactEstimator and data_splitting function, please refers to rebuttal for **Reviewer uxuX** Due to space constraints, additional codes and results are available upon request.

---

> > ### Comment · Reviewer_Xfce · 2025-08-04
> > **Response to rebuttal**
> >
> > I thank the authors for their extremely thorough and considered response to my questions. I feel that these details provide very helpful additional context and would be very useful to include in the paper, especially the experimental results provided which have been requested by other reviewers. I keep my original score unchanged, as whilst these additions improve the paper, they do not change my overall assessment of the work.

---

> > > ### Author Response · Authors · 2025-08-04
> > > **Thank you!**
> > >
> > > We sincerely appreciate your supportive feedback and we welcome any further questions or discussions.
> > >
> > > Warm Regards,
> > > The Authors

---

### Official Review · Reviewer_uxuX · 2025-07-03

**Clarity:** 3
**Significance:** 2
**Originality:** 3
**Rating:** 4
**Confidence:** 2

**Summary:**

This paper tackles the problem of optimizing online ad bidding strategies by creating a model that, unlike previous work, simultaneously accounts for three real-world complexities: 1) the delayed impact of ads on customer purchases, 2) the cumulative effects of repeated ad exposure, and 3) customer heterogeneity. To achieve this, the authors make the following key contributions: They model the ad bidding problem as a CMDP with delayed Poisson rewards. The "contextual" aspect allows for personalized strategies, while the MDP structure captures cumulative effects over time. They introduce a novel and efficient two-stage maximum likelihood estimator with a data-splitting strategy. This allows them to learn the parameters of their complex model online while carefully controlling for estimation errors caused by the delayed rewards. Based on this estimation method, they design a reinforcement learning algorithm that learns personalized bidding policies and prove that it achieves a near-optimal theoretical regret bound.

**Questions:**

1. The paper is entirely theoretical. Could you please justify this choice? While the theoretical results are strong, the absence of simulations makes it difficult to gauge the practical performance of the proposed algorithm. My evaluation would improve significantly if you could provide even a small-scale simulation study on synthetic data.

2. The model relies on several specific parametric assumptions. How sensitive do you expect your theoretical results and your proposed algorithm to be if these assumptions are violated, as they likely would be in a real-world setting?

**Ethical Concerns:**

["NO or VERY MINOR ethics concerns only"]

**Final Justification:**

The author provided simulation results in their rebuttal and show the performance compared with other baselines.

**Limitations:**

yes

**Quality:**

2

**Strengths And Weaknesses:**

Strengths
1. The paper tackles a highly significant and practical problem in computational advertising. Its primary contribution is the novel and unified CMDP framework that, for the first time, jointly models the crucial dynamics of delayed rewards, cumulative ad effects, and user heterogeneity.

2. The technical quality of the paper is exceptionally high. The main theoretical contribution—the development of the two-stage MLE for online estimation under delayed Poisson rewards—is sophisticated and rigorously analyzed.

3. The paper is well-written, clearly structured, and does an excellent job of motivating the problem by highlighting the limitations of existing approaches.

Weakness

The most significant weakness of the paper is the complete absence of empirical results. The work is purely theoretical, with no simulations or experiments on real or synthetic data. While the theoretical guarantees are strong, this omission makes it impossible to assess the practical performance of the algorithm and the real-world magnitude of the constants hidden in the regret bound and the robustness of the method to violations of its underlying assumptions.

---

> ### Author Rebuttal · Authors · 2025-07-30
>
> Thank you for the positive feedback and the suggestion to include synthetic simulations. In response, we conducted experiments during the rebuttal period with two main findings: (1) our algorithm achieves near-optimal regret scaling of $\sqrt{T}$; (2) it substantially outperforms several baseline policies such as aggressive bidding, passive bidding, and a bidding at random strategy. We hope this additional empirical evidence could mitigate the reviewer's concerns and encourage the reviewer to re-consider our paper’s rating evaluation.
>
> **Environment Setup**: We simulate a second-price auction with horizon $H = 3$ (a reasonable horizon length since in reality advertisers show their ads only a few times to each Internet user), context dimension $D = 2$, and $T = 20,000$ sequential customers. The first 2,400 rounds are for exploration; the rest for exploitation. Context vectors $x_t \in \mathbb{R}^2, d_l, \beta_h, \sigma_h$ are sampled elementwise from $|\mathcal{N}(0,1)| + 0.1$, ensuring positivity. Parameters $\theta_l  \sim 5 \cdot |\mathcal{N}(0,1)| + 0.1$. Note that under our current simulation setting, an impression brings on average an instantaneous reward that is around 5 times than its cost (i.e. HOB price). Moreover, the reward of the impression decays quickly with time. These suggest that an aggressive bidding strategy (i.e. try to always win the bid and get the impression) is actually close to the optimal bidding strategy and thus serves as a competitive baseline policy.
>
> **Estimators**: We estimate four categories of parameters. $\theta_l$ is learned using an online Newton method (Algorithm 3) with truncation threshold $\tau = 100{,}000$, bound $B_\theta = 10$, zero initialization, and $V_0 = I$. The delay impact $d_l$ is estimated via a two-stage MLE (Eq. 3) using $D_{t,l}$, $\hat{\theta}_l$, and $x_t$. $\beta_h$ is estimated by ridge regression (Eq. 5, $\lambda = 1.0$), and $\sigma_h$ via empirical variance (Eq. 6). As $\theta_l$ and $d_l$ are the most critical, we provide the corresponding code in the appendix.
>
> **Connection Between Bid Amount, Auction Outcome, and Realized Reward**: In our setting, the bidder's action is the bid amount, but due to the second-price auction format, the reward depends only on the outcome–whether the bid exceeds the highest other bid (HOB)-not the bid itself. Winning results in paying the HOB; losing incurs no cost. This decoupling allows optimal rewards to be computed from outcomes rather than bids.
> We leverage this to evaluate both oracle and policy-based rewards. With full knowledge of true parameters, we enumerate all $2^H = 8$ possible win/loss outcomes across $H = 3$ rounds and define the **oracle reward** as the maximum achievable. With estimated parameters, we compute the expected reward for each outcome and choose the best sequence (Algorithm 1), which is then run through the simulator to obtain the realized reward. Regret is defined as the cumulative gap between oracle and actual rewards, reflecting errors from estimation and suboptimal bidding.
>
> **Simulation and Regret**
> 1. Sample ground-truth parameters $(\theta_l, d_l, \beta_h, \sigma_h)$, which remain fixed. Initialize estimates $(\hat{\theta}_l, \hat{d}_l, \hat{\beta}_h, \hat{\sigma}_h)$ using default values (e.g., zeros or identity).
> 2. Simulate $T = 20{,}000$ customers over two phases: exploration (first 2,400 rounds) and exploitation (remaining 17,600 rounds).
> 3. Exploration Phase: Uniformly sample each of the $2^H = 8$ bidding outcome sequences across 2,400 rounds (300 per outcome). For each round: Sample context $x_t$; simulate reward and HOB under the chosen bidding outcome; Compute and record the oracle reward; Store observations for estimation; Apply data splitting to update parameter estimates.
> 4. Exploitation Phase: For each round: Observe $x_t$; compute estimated rewards for all 8 bidding sequences using current estimates; Select the best sequence; simulate reward; compute oracle reward and reward gap; Update estimates via **data splitting**.
> 5. Return cumulative regret as $np.cumsum(reward_{diff})$.
>
> **Baseline Policies**: We compare against three baselines. **Aggressive  bidding:** This simulates the strategy that the firm actively engages in winning ad impressions by consistently offering the highest bidding price (and pays second best price). In other words, the firm always wins by bidding above the HOB $(o_t = [1, 1, 1])$; **Bidding at random:** Sample $o_t$ uniformly from all 8 bidding outcomes and set bids accordingly; **Passive bidding:** This simulates the strategy that the firm rarely engages in the ad impression bidding war, where it gives very low bidding or simply does not bid $(o_t = [0, 0, 0])$.
>
> Regret for each is computed using the same oracle-based metric, serving as benchmarks for our algorithm.
> We remark here again that due to our simulation setting, the aggressive bidding strategy is close to the optimal strategy. Therefore, it serves as a competitive benchmark, though it obviously suffers a O(T) regret since it is a fixed policy without any learning.
>
> We run our proposed algorithm alongside the three baseline policies over 20 independent trials. The table below reports the average cumulative regret with ±0.5 standard deviation for all four algorithms. The last row shows the estimated regret order, obtained by applying a log-log transformation followed by linear regression. As shown, our algorithm achieves a fitted regret order of 0.37, while all three baselines exhibit linear regret. These results are consistent with our theoretical analysis and demonstrate the superior performance of our approach and the importance of learning.
> | Round   | Our Algorithm                 | Aggressive Bid              | Random Bid                   | Passive Bid                    |
> |---------|----------------------------|-----------------------------|------------------------------|-------------------------------|
> | 500     | 1770 [1174, 2379]          | 228 [121, 336]              | 1920 [1791, 2049]            | 4630 [3391, 5869]             |
> | 5000    | 8465 [5585, 11345]         | 2373 [1243, 3502]           | 19285 [17873, 20697]         | 46179 [33850, 58508]          |
> | 10000   | 8484 [5606, 11363]         | 4741 [2477, 7005]           | 38399 [35573, 41226]         | 92468 [67735, 117201]         |
> | 15000   | 8505 [5628, 11382]         | 7142 [3724, 10561]          | 57651 [53452, 61849]         | 138652 [101576, 175729]       |
> | 20000   | 8525 [5649, 11401]         | 9568 [4991, 14146]          | 76945 [71390, 82500]         | 184873 [135414, 234332]       |
> | **Fitted Order** | **0.37**                 | **1.0**                    | **1.0**                      | **1.0**                        |
>
>
> ---
> **Appendix**
>
> Due to space constraints, additional codes and results are available upon request.
>
> ```python
> def data_splitting(results):
>     W_t_l = {}; D_t_l = {}
>     for result in results:
>          h = result['round']; s1 = result['S1']; s2 = result['S2'];o_t = result['o_t'];y = result['y_t']
>          if o_t == 1:
>             l = s1;W_t_l.setdefault(l, []).append((h, y))
>          else:
>             if s1 == 'inf':
>                W_t_l.setdefault('-inf', []).append((h, y))  # For θ_{-inf}
>             else:
>                l = s1;D_t_l.setdefault(l, []).append((h, s2, y))
>     return W_t_l, D_t_l
> class ThetaEstimator:
>     def truncate_payoff(self, y, x, V, tau):
>         norm_x_V = np.sqrt(x.T @ V @ x); return y if norm_x_V * abs(y) < tau else 0.0
>     def gradient_term(self, y, x, theta):
>         return (-y + x @ theta) * x
>     def constrained_quad_solver(self, theta0, V, k, b, tol=1e-6, max_iter=100):
>         d = len(theta0)
>         def phi(lambda_):
>             A = V + 2 * lambda_ * np.eye(d); rhs = -k - 2 * lambda_ * theta0; return np.linalg.solve(A, rhs)
>         def constraint(lambda_):
>             phi_val = phi(lambda_); theta = theta0 + phi_val; return norm(theta) - b
>         try:
>             phi0 = np.linalg.solve(V, -k); theta_unconstrained = theta0 + phi0
>             if norm(theta_unconstrained) <= b: return theta_unconstrained
>         except: pass
>         l, r = 0.0, 1e6
>         for _ in range(max_iter):
>             m = (l + r) / 2
>             if constraint(m) > 0: l = m
>             else: r = m
>             if r - l < tol: break
>         return theta0 + phi(r)
>     def run_estimation_by_state(self, W_t_l, true_theta_l, X_t_l, theta_l_hat, V_l, tau=10, radius=1):
>         error_l = {l: [] for l in W_t_l}
>         for l, observations in W_t_l.items():
>             for (round_idx, y_raw) in observations:
>                 x_t = X_t_l[l][round_idx]; y_t = self.truncate_payoff(y_raw, x_t, V_l[l], tau)
>                 k_t = self.gradient_term(y_t, x_t, theta_l_hat[l]); theta_next = self.constrained_quad_solver(theta_l_hat[l], V_l[l], k_t, radius)
>                 V_l[l] += np.outer(x_t, x_t); theta_l_hat[l] = theta_next
>                 error = norm(theta_next - true_theta_l[l]) ** 2; error_l[l].append(error)
>         return V_l, theta_l_hat, error_l
> class DelayImpactEstimator:
>     def __init__(self, d_true):
>         self.d_true = d_true; self.num = {l: 0.0 for l in d_true}; self.denom = {l: 0.0 for l in d_true};
>         self.d_hat = {l: [] for l in d_true}; self.errors = {l: [] for l in d_true};
>         self.latest_d_hat = {l: np.nan for l in d_true}
>     def run_update(self, D_t_l, x, theta_hat):
>         self.theta_hat = theta_hat
>         for l in D_t_l:
>             for h, S2, y in D_t_l[l]:
>                 key = str(S2) if str(S2) in self.theta_hat else int(S2)
>                 theta = self.theta_hat[key]
>                 self.num[l] += y; self.denom[l] += np.dot(theta, x)
>             d_hat = self.num[l] / self.denom[l] if self.denom[l] != 0 else np.nan
>             self.d_hat[l].append(d_hat); self.latest_d_hat[l] = d_hat
>             error = (d_hat - self.d_true[l]) ** 2 if not np.isnan(d_hat) else np.nan
>             self.errors[l].append(error)
>         return self.latest_d_hat

---

> > ### Comment · Reviewer_uxuX · 2025-08-04
> >
> > Thanks for your response. You have resolved most of my concern. I will raise my score to 4.

---

> > > ### Author Response · Authors · 2025-08-04
> > > **Thank you!**
> > >
> > > We sincerely appreciate your positive support for our efforts in conducting simulations during the rebuttal period. We will continue working to make the experiments more comprehensive and welcome any further questions or discussions.
> > >
> > > Warm Regards,
> > > The Authors

---

### Official Review · Reviewer_1MCA · 2025-07-04

**Clarity:** 3
**Significance:** 3
**Originality:** 3
**Rating:** 3
**Confidence:** 3

**Summary:**

The paper addresses the challenge of modeling and optimizing online advertising bids by integrating delayed rewards, cumulative ad impacts, and customer heterogeneity. It formulates ad bidding as a Contextual Markov Decision Process (CMDP) with delayed Poisson rewards and proposes a two-stage maximum likelihood estimator combined with a reinforcement learning algorithm to derive personalized bidding strategies.

**Questions:**

NA

**Ethical Concerns:**

["NO or VERY MINOR ethics concerns only"]

**Final Justification:**

I shall keep my score.

**Paper Formatting Concerns:**

No formatting issue.

**Quality:**

3

**Strengths And Weaknesses:**

Strengths:
1. It is interesting to modeling the online advertising problem through Contextual MDP.
2. Concrete theoretical proofs for the regret bound.

Weaknesses:
1. As the motivation of this work is the online advertising problem, the empirical experiments seem to be necessary. Basically, it is hard to determine whether the assumption about the conversion rate is reasonable.

---

> ### Author Rebuttal · Authors · 2025-07-30
>
> We appreciate the reviewer’s support for our theoretical analysis. Due to the sensitivity and non-disclosure of real-world bidding data, such data is unfortunately rarely available—explaining why many prior works lack real-world evaluations. Nonetheless, following the reviewer's comment,  we have conducted synthetic study during the rebuttal period: (1) our algorithm achieves near-optimal regret scaling of $\sqrt{T}$ in empirical studies, and (2) it substantially outperforms baseline policies such as aggressive bidding, passive bidding, and a random bidding policy. We hope this additional empirical evidence could mitigate the reviewer's concerns and encourage the reviewer to re-consider our paper’s rating evaluation accordingly.
>
> **Environment Setup**: We simulate a second-price auction with horizon $H = 3$ (a reasonable horizon length since in reality advertisers show their ads only a few times to each Internet user), context dimension $D = 2$, and $T = 20,000$ sequential customers. The first 2,400 rounds are for exploration; the rest for exploitation. Context vectors $x_t \in \mathbb{R}^2, d_l, \beta_h, \sigma_h$ are sampled elementwise from $|\mathcal{N}(0,1)| + 0.1$, ensuring positivity. Parameters $\theta_l  \sim 5 \cdot |\mathcal{N}(0,1)| + 0.1$. Note that under our current simulation setting, an impression brings on average an instantaneous reward that is around 5 times than its cost (i.e. HOB price). Moreover, the reward of the impression decays quickly with time. These suggest that an aggressive bidding strategy (i.e. try to always win the bid and get the impression) is actually close to the optimal bidding strategy and thus serves as a competitive baseline policy.
>
> **Estimators**: We estimate four categories of parameters. $\theta_l$ is learned using an online Newton method (Algorithm 3) with truncation threshold $\tau = 100{,}000$, bound $B_\theta = 10$, zero initialization, and $V_0 = I$. The delay impact $d_l$ is estimated via a two-stage MLE (Eq. 3) using $D_{t,l}$, $\hat{\theta}_l$, and $x_t$. $\beta_h$ is estimated by ridge regression (Eq. 5, $\lambda = 1.0$), and $\sigma_h$ via empirical variance (Eq. 6). As $\theta_l$ and $d_l$ are the most critical, we provide the corresponding code in the appendix.
>
> **Connection Between Bid Amount, Auction Outcome, and Realized Reward**: In our setting, the bidder's action is the bid amount, but due to the second-price auction format, the reward depends only on the outcome–whether the bid exceeds the highest other bid (HOB)-not the bid itself. Winning results in paying the HOB; losing incurs no cost. This decoupling allows optimal rewards to be computed from outcomes rather than bids.
> We leverage this to evaluate both oracle and policy-based rewards. With full knowledge of true parameters, we enumerate all $2^H = 8$ possible win/loss outcomes across $H = 3$ rounds and define the **oracle reward** as the maximum achievable. With estimated parameters, we compute the expected reward for each outcome and choose the best sequence (Algorithm 1), which is then run through the simulator to obtain the realized reward. Regret is defined as the cumulative gap between oracle and actual rewards, reflecting errors from estimation and suboptimal bidding.
>
> **Simulation and Regret**
> 1. Sample ground-truth parameters $(\theta_l, d_l, \beta_h, \sigma_h)$, which remain fixed. Initialize estimates $(\hat{\theta}_l, \hat{d}_l, \hat{\beta}_h, \hat{\sigma}_h)$ using default values (e.g., zeros or identity).
> 2. Simulate $T = 20{,}000$ customers over two phases: exploration (first 2,400 rounds) and exploitation (remaining 17,600 rounds).
> 3. Exploration Phase: Uniformly sample each of the $2^H = 8$ bidding outcome sequences across 2,400 rounds (300 per outcome). For each round: Sample context $x_t$; simulate reward and HOB under the chosen bidding outcome; Compute and record the oracle reward; Store observations for estimation; Apply data splitting to update parameter estimates.
> 4. Exploitation Phase: For each round: Observe $x_t$; compute estimated rewards for all 8 bidding sequences using current estimates; Select the best sequence; simulate reward; compute oracle reward and reward gap; Update estimates via data splitting.
> 5. Return cumulative regret as $np.cumsum(reward_{diff})$.
>
> **Baseline Policies**: We compare against three baselines. **Aggressive  bidding:** This simulates the strategy that the firm actively engages in winning ad impressions by consistently offering the highest bidding price (and pays second best price). In other words, the firm always wins by bidding above the HOB $(o_t = [1, 1, 1])$; **Bidding at random:** Sample $o_t$ uniformly from all 8 bidding outcomes and set bids accordingly; **Passive bidding:** This simulates the strategy that the firm rarely engages in the ad impression bidding war, where it gives very low bidding or simply does not bid $(o_t = [0, 0, 0])$.
>
> Regret for each is computed using the same oracle-based metric, serving as benchmarks for our algorithm.
> We remark here again that due to our simulation setting, the aggressive bidding strategy is close to the optimal strategy. Therefore, it serves as a competitive benchmark, though it obviously suffers a O(T) regret since it is a fixed policy without any learning.
>
> We run our proposed algorithm alongside the three baseline policies over 20 independent trials. The table below reports the average cumulative regret with ±0.5 standard deviation for all four algorithms. The last row shows the estimated regret order, obtained by applying a log-log transformation followed by linear regression. As shown, our algorithm achieves a fitted regret order of 0.37, while all three baselines exhibit linear regret. These results are consistent with our theoretical analysis and demonstrate the superior performance of our approach and the importance of learning.
> | Round   | Our Algorithm                 | Aggressive Bid              | Random Bid                   | Passive Bid                    |
> |---------|----------------------------|-----------------------------|------------------------------|-------------------------------|
> | 500     | 1770 [1174, 2379]          | 228 [121, 336]              | 1920 [1791, 2049]            | 4630 [3391, 5869]             |
> | 5000    | 8465 [5585, 11345]         | 2373 [1243, 3502]           | 19285 [17873, 20697]         | 46179 [33850, 58508]          |
> | 10000   | 8484 [5606, 11363]         | 4741 [2477, 7005]           | 38399 [35573, 41226]         | 92468 [67735, 117201]         |
> | 15000   | 8505 [5628, 11382]         | 7142 [3724, 10561]          | 57651 [53452, 61849]         | 138652 [101576, 175729]       |
> | 20000   | 8525 [5649, 11401]         | 9568 [4991, 14146]          | 76945 [71390, 82500]         | 184873 [135414, 234332]       |
> | **Fitted Order** | **0.37**                 | **1.0**                    | **1.0**                      | **1.0**                        |
>
>
> **Appendix**
>
> Due to space constraints, additional codes and results are available upon request.
>
> ```python
> class ThetaEstimator:
>     def truncate_payoff(self, y, x, V, tau):
>         norm_x_V = np.sqrt(x.T @ V @ x); return y if norm_x_V * abs(y) < tau else 0.0
>     def gradient_term(self, y, x, theta):
>         return (-y + x @ theta) * x
>     def constrained_quad_solver(self, theta0, V, k, b, tol=1e-6, max_iter=100):
>         d = len(theta0)
>         def phi(lambda_):
>             A = V + 2 * lambda_ * np.eye(d); rhs = -k - 2 * lambda_ * theta0; return np.linalg.solve(A, rhs)
>         def constraint(lambda_):
>             phi_val = phi(lambda_); theta = theta0 + phi_val; return norm(theta) - b
>         try:
>             phi0 = np.linalg.solve(V, -k); theta_unconstrained = theta0 + phi0
>             if norm(theta_unconstrained) <= b: return theta_unconstrained
>         except: pass
>         l, r = 0.0, 1e6
>         for _ in range(max_iter):
>             m = (l + r) / 2
>             if constraint(m) > 0: l = m
>             else: r = m
>             if r - l < tol: break
>         return theta0 + phi(r)
>     def run_estimation_by_state(self, W_t_l, true_theta_l, X_t_l, theta_l_hat, V_l, tau=10, radius=1):
>         error_l = {l: [] for l in W_t_l}
>         for l, observations in W_t_l.items():
>             for (round_idx, y_raw) in observations:
>                 x_t = X_t_l[l][round_idx]; y_t = self.truncate_payoff(y_raw, x_t, V_l[l], tau)
>                 k_t = self.gradient_term(y_t, x_t, theta_l_hat[l]); theta_next = self.constrained_quad_solver(theta_l_hat[l], V_l[l], k_t, radius)
>                 V_l[l] += np.outer(x_t, x_t); theta_l_hat[l] = theta_next
>                 error = norm(theta_next - true_theta_l[l]) ** 2; error_l[l].append(error)
>         return V_l, theta_l_hat, error_l
> class DelayImpactEstimator:
>     def __init__(self, d_true):
>         self.d_true = d_true; self.num = {l: 0.0 for l in d_true}; self.denom = {l: 0.0 for l in d_true};
>         self.d_hat = {l: [] for l in d_true}; self.errors = {l: [] for l in d_true};
>         self.latest_d_hat = {l: np.nan for l in d_true}
>     def run_update(self, D_t_l, x, theta_hat):
>         self.theta_hat = theta_hat
>         for l in D_t_l:
>             for h, S2, y in D_t_l[l]:
>                 key = str(S2) if str(S2) in self.theta_hat else int(S2)
>                 theta = self.theta_hat[key]
>                 self.num[l] += y; self.denom[l] += np.dot(theta, x)
>             d_hat = self.num[l] / self.denom[l] if self.denom[l] != 0 else np.nan
>             self.d_hat[l].append(d_hat); self.latest_d_hat[l] = d_hat
>             error = (d_hat - self.d_true[l]) ** 2 if not np.isnan(d_hat) else np.nan
>             self.errors[l].append(error)
>         return self.latest_d_hat

---

> ### Comment · Reviewer_1MCA · 2025-08-01
>
> Thanks for your repsonse. However, I am sorry about some mislead in previous reivew. Let me make it clear: in Assumption 2.2, the conversion rate is assumed to be the form that is linear to the context. This is common in the field. for example contextual bandit, since it is helpful in proving concrete results. My concern is whether such a form is able to capture the conversion rate in real online ads platform, which leads to the necessarity of real-world empirical experiments.

---

> > ### Author Response · Authors · 2025-08-01
> > **Responses**
> >
> > We sincerely appreciate the reviewer's clarification. We believe linearity assumption is general to capture the real-world conversion rates for online advertising platforms for the following reasons.
> >
> > - Our model is essentially equivalent to linear representation, specifically the average conversion rate is modeleld $\mu_t = \theta^{\top}_l\phi(x_t)$, where $\phi(x_t) \in R^d$ is a potentially complex, known feature mapping. Such feature mappings are capable of representing diverse and sophisticated functions, including kernel methods, deep neural network embeddings, and these linear representations are even found and studied in large language models (LLMs) and transformer-based architectures, for example in papers [1, 2]. This shows that a broad class of nonlinear functions $f(x_t)$ can indeed be effectively represented as a linear function by feature $\phi(x_t)$.
> >
> > - Moreover, this linearity assumption on feature mappings is widely used in reinforcement learning literature, from foundational studies such as [3] to the most recent ones [4]. Our assumption follows standard practice and further builds upon the existing theoretical frameworks.
> >
> > - In real-world applications, the feature mapping $\phi$ may come from domain expertise or learned from historical data. However, as highlighted in our initial rebuttal, real-world ad-bidding data is highly sensitive and thus generally not publicly accessible, which has resulted in the learning-to-bid literature remaining largely theoretical. Despite these practical constraints, recent studies such as [5] demonstrate that well-designed simulation studies alone can offer valuable insights that merit publication. Given the importance of simulation and followed by other reviewer’s suggestion, we add the non-trivial simulations over the rebuttal period and show that under the linear mapping assumption, our algorithm is provably optimal.
> > We hope this finding could open new directions for future empirical research and adds value to the learning-to-bid community.
> >
> >
> > We hope this response mitigates the reviewer's concerns and we are open to further discussion and questions.
> >
> > **References**
> >
> > [1] The Linear Representation Hypothesis and the Geometry of Large Language Models
> >
> > [2] Linear Representations of Political Perspective Emerge in Large Language Models
> >
> > [3] Provably Efficient Reinforcement Learning with Linear Function Approximation
> >
> > [4] Nearly Optimal Algorithms for Contextual Dueling Bandits from Adversarial Feedback
> >
> > [5] Non-uniform bid-scaling and equilibria for different auctions: An empirical study

---

> > > ### Comment · Reviewer_1MCA · 2025-08-07
> > >
> > > Thanks for your response.

---

> > > > ### Author Response · Authors · 2025-08-08
> > > > **Further Feedback Requested**
> > > >
> > > > We sincerely appreciate the Reviewer IMCA's response. Please let the authors know whether the reviewer has further concerns. We are willing to address them as best as we could.
> > > >
> > > > Warm Regards,
> > > >
> > > > The authors

---

### Comment · Area_Chair_iov6 · 2025-08-08
**Question for the authors re linear assumption**

Hi,

Thanks for engaging in the discussion with the reviewers!

I have a quick question re the linear assumption: as you mentioned, it is a fairly common assumption in the literature and it allows deriving a "simple enough" algorithm that you are able to derive strong theoretical guarantees. If I'm willing to give up on the theoretical guarantees, would I still be able to take the blueprint of the proposed algorithm and adapt it to, say, a neural-net based algorithm? If yes, could you please sketch what are the needed changes to the algorithm?

Thanks!

---

> ### Author Response · Authors · 2025-08-09
> **Response**
>
> We sincerely thank the AC for the discussion and we can incorporate neural-net as follows.
>
> Just a brief recap of our model, if we win the bid, $o^t_h = 1$, the average product conversion rate $\mu^t_h = h_{S^t_{h, 1}}(x_t)$, where $x_t$ is the received feature of customer $t$, and $S^t_{h, 1}$ is the state which captures the time elapsed since most recent ads impression, $h_{S^t_{h, 1}}$ is the state dependent neural network which we want to estimate. If we lose the bid $o^t_h = 0$, the average product conversion rate $\mu^t_h = d_{S^t_{h,1}}h_{S^t_{h, 2}}(x_t)$, where $d_{S^t_{h,1}}$ is the delayed impact, and $S^t_{h, 2}$ is the time interval between the two most recent impressions (details see Def 2.1).
>
> To incorporate the estimation of the state-dependent neural network $h_l$ (analogous to $\theta_l$) into our algorithm, we do the following. In essence, everything in Algorithm 1 remains unchanged, i.e. the exploration, exploitation, data-splitting strategy, and estimation, except the following modifications.
>
> - First, we replace the estimation of $\theta_l$ (Eqn. (4)) by estimating the neural network $h_l$ by the following procedure (pertubed reward + gradient descent, similar to Algorithm 1 in [1])
>   - We approximate $h(x)$ by $f(x, \theta) = \sqrt{m}W_L\phi(W_{L-1}\phi(\ldots \phi(W_1x)))$, where $m$ is the network width, and $\phi(x) = ReLU(x)$, $L$ is the neural network depth
>   - We initialize $\theta_0 = (vec(W_1) \ldots vec(W_L))$ by random sample entry from N(0, 4/m) and do online updating by gradient descent with perturbed reward using observed product conversion $y^t_h$ by follows
>     - Generated observation perturbation $\gamma_{s \in [t]}$ from Poisson $(\lambda)$, then generate binomial random variable $1_{s \in [t]}$ to add / subtract the noise from the observation
>     - $\min_{\theta} L(\theta) $ = $\frac{1}{2} \sum^t_{s=1} (f(x_t, \theta) - (y^s_h + 1_{s} \gamma_s) )^2 + m \beta |\theta - \theta_0|^2$
>   - Then based on this first stage estimates of $h$, plugging in to estimate delayed impact factor $d_l$ by $\frac{\sum^t_{s=1}\sum_{h \in D_{s, l}} y^s_h }{\sum^t_{s=1}\sum_{h \in D_{s, l}}f_{S^s_{h,2}}(x_s, \hat \theta)}$ (analogous to Eqn. (3))
> - Since we do not have theoretical guarantees for $h$ estimation (no confidence interval), we cannot do UCB (Line 10 of our Algorithm 1). Instead, we might act greedily based on our point estimate. Even though Algorithm in [1] has theoretical guarantees, the key difference is that they assume noise of their observation is subGaussian, while in our case, it is not realistic to assume that the observation $y^t_h$ has sub-Gaussian noise, since product conversion has been known to have heavy tail for a long period of time. It is still a challenging open problem of developing theoretical-guaranteed neural contextual bandits with heavy tail observation.
>
> Instead, our proposed framework leads to near-optimal results if we can parameterize $h_l(x_t) = \theta^{\top}_l \phi(x_t)$, where $\phi$ can be the neural net based embedding or other known feature mapping, then our algorithm remains near optimal, satisfying $\sqrt{T}$ regret bound in this case. This construction is very common in academia and industry as we mentioned from foundational studies such as [2] to the most recent ones [3]. During discussion with our industry friends over the rebuttal period, in the real-world, companies might construct these feature mapping using complicated methods but retain the overall linear structure for the ease of scalability and interpretability. For example, the expected conversion rate could be a linear function of the expected webpage stay time and the predicted total spending but the expected webpage stay time and the predicted total spending are deep neural networks of the observed customer feature $x_t$.
>
> To summarize, our proposed framework leads to near-optimal results if there exists an online estimation oracle with theoretical guarantee for the estimation of $h$ and the estimation oracle employed in our paper based on our knowledge is SOTA. Our algorithm is possible to extend to a neural network if we drop the theoretical concerns. Developing theoretical guarantee for neural bandits under heavy tail observation is a promising future direction.
>
> We hope our response clarifies the questions and we are open and willing to further discussions.
>
> **References:**
>
> [1] Learning Neural Contextual Bandits through Perturbed Rewards
>
> [2] Provably Efficient Reinforcement Learning with Linear Function Approximation
>
> [3] Nearly Optimal Algorithms for Contextual Dueling Bandits from Adversarial Feedback

---

### Note · Authors · 2025-08-14

Dear AC and Reviewers:

We sincerely appreciate the time and effort all reviewers and AC spent to evaluating our work, with special thanks to AC iov6 for engaging in the discussion. We thank the constructive feedback and are happy that all reviewers think our problem as both interesting and critical to important real-world applications, as well as recognizing our solid theoretical contribution (IMCA, uxuX, Xfce, iRrk).

A common concern among the reviewers (IMCA, uxuX, Xfce, iRrk) was the experimental validation of our algorithm on synthetic data. In response, we conducted simulations over the rebuttal period, showing that the results closely aligned with the predictions of our theorem and outperformed strong baseline policies. We sincerely happy to see that these results convinced reviewer uxuX to raise their rating to 4.

Reviewers IMCA and iRrk questioned whether the linearity assumption was adequate for modeling advertisement conversion rates. To address this, we demonstrated the richness of linear feature mappings, highlight their popularity in academia, and explain why linearity is also favored in industry for its interpretability and scalability. Furthermore, we showed how our algorithms could be adapted to neural network–based conversion rates modeling if one is willing to give up theoretical guarantees. Unfortunately, reviewers IMCA and iRrk did not provide follow-up suggestions, and reviewer iRrk remained silent throughout the rebuttal period.

We also appreciate the supportive comments from reviewer Xfce and will incorporate the learning-to-bid literature review in the appendix as suggested.

We hope that the additional experiments, clarifications, and improvements provided during the rebuttal phase have addressed the reviewers’ concerns and will contribute to a favorable decision. Once again, we extend our sincere gratitude for your invaluable contributions to our work.

Warm Regards,

Authors

---

### Decision · Program_Chairs · 2025-09-17

**Decision:**

Accept (poster)

**Comment:**

The paper studies an online advertising setting focusing on three key factors: delayed and long-term effects, cumulative ad impacts such as reinforcement or fatigue, and customer heterogeneity. The authors formalize the problem as a specific instance of contextual MDPs and introduces an algorithm with strong theoretical guarantees. The algorithm is then validated on small-scale synthetic experiments.

After the initial round of reviews, the rebuttal, and the discussion, I'm proposing acceptance for the paper.

The paper is primarily theoretical in nature. Overall the technical contribution and the theoretical developments are borderline, as the results are technically sound, but there is no major surprising aspect in the analysis or in the algorithmic design. The other aspect that then enters into the decision is on whether the modeling part is significant and the paper is a rigorous study of how to design algorithms for the advertising scenario. On this aspect, the authors have made a significant effort in justifying most of the modeling choices and they are indeed pushing the boundary of previous theoretical models by considering different aspects of the problem that are significant in practice (the "delayed and long-term effects, cumulative ad impacts such as reinforcement or fatigue, and customer heterogeneity" mentioned in the abstract). As pointed out by several reviewers, there are indeed still residual assumptions that are not realistic (e.g., linear) and some elements that are not integrated yet (e.g., budget). On the former, I think it is important to discriminate between assumptions that are introduced to derive the theory or are intrinsic in the algorithm itself to the extent that we would not know how to use it beyond the theoretical assumptions. As discussed during the rebuttal, the authors have quite realistic ideas on how to extend the proposed algorithm to eg, using neural networks. On the second aspect (missing elements), I think the current submission is going in the right direction, although I would still encourage the authors to expand the discussion on the limitations of the current formalization and what are the next aspects of the real-world problem that would need to be studied to close the gap even further.

Finally, most reviewers were concerned with the scope of the empirical validation. Unfortunately there is no dataset or simulator commonly used to test online learning algorithms in these domains, so it's always very challenging to add any meaningful study, unless authors have direct access to production platforms. In this sense, what the authors reported in the rebuttal is probably the best that we can expect and I strongly encourage them to add it to the paper to provide a significant empirical validation of the setting and of the theoretical results.